# Understanding the Generalization of Adam in Learning Neural Networks with Proper Regularization

## Abstract

Adaptive gradient methods such as Adam have gained increasing popularity in deep learning optimization. However, it has been observed in many deep learning applications such as image classification, Adam can converge to a different solution with a worse test error compared to (stochastic) gradient descent, even with a fine-tuned regularization. In this paper, we provide a theoretical explanation for this phenomenon: we show that in the nonconvex setting of learning over-parameterized two-layer convolutional neural networks starting from the same random initialization, for a class of data distributions (inspired from image data), Adam and gradient descent (GD) can converge to different global solutions of the training objective with provably different generalization errors, even with weight decay regularization. In contrast, we show that if the training objective is convex, and the weight decay regularization is employed, any optimization algorithms including Adam and GD will converge to the same solution if the training is successful. This suggests that the generalization gap between Adam and SGD in the presence of explicit regularization is fundamentally tied to the nonconvex landscape of deep learning optimization, which cannot be covered by the recent neural tangent kernel (NTK) based analysis.

## 1 Introduction

Adaptive gradient methods (Duchi et al., 2011; Hinton et al., 2012; Kingma & Ba, 2015; Reddi et al., 2018) such as Adam are very popular optimizers for training deep neural networks. By adjusting the learning ratethis coordinate-wisely based on historical gradient information, they are known to be able to automatically choose appropriate learning rates to achieve fast convergence in training. Because of this advantage, Adam and its variants are widely used in deep learning. Despite their fast convergence, adaptive gradient methods have been observed to achieve worse generalization performance compared with gradient descent and stochastic gradient descent (SGD) (Wilson et al., 2017; Luo et al., 2019; Chen et al., 2020; Zhou et al., 2020) in many deep learning tasks such as image classification (we have done some simple deep learning experiments to justify this, the results are reported in Table 1). Even with proper regularization, achieving good test error with adaptive gradient methods seems to be challenging.

Several recent works provided theoretical explanations of this generalization gap between Adam and GD. Wilson et al. (2017); Agarwal et al. (2019) considered a setting of linear regression, and showed that Adam can fail when learning an overparameterized linear model on certain specifically designed data, while SGD can learn the linear model to achieve zero test error. This ex-

| Models | AlexNet | VGG-16 | ResNet-18 |
|--------|---------|--------|-----------|
| SGD    | 75.22   | 93.25  | 94.62     |
| Adam   | 73.08   | 92.19  | 92.93     |

Table 1: Test accuracy (%) comparison between Adam and SGD on the CIFAR-10 dataset.

ample in linear regression offers valuable insights into the difference between SGD and Adam. However, it is under a convex optimization setting, and as we will show in this paper (Theorem 4.2), the performance difference between Adam and GD can be easily avoided by adding an arbitrarily small regularization term, because the regularized training loss function is strongly convex and all algorithms will converge to the same unique global optimum. For this reason, we argue that the example in the convex setting cannot capture the fundamental differences between GD and Adam. More recently, Zhou et al. (2020) studied the expected escaping time of Adam and SGD from a local

basin, and utilized this to explain the difference between SGD and Adam. However, their results do not take NN architecture into consideration, and do not provide an analysis of test errors either.

In this paper, we aim at answering the following question

*Why is there a generalization gap between Adam and gradient descent in learning neural networks, even with proper regularization?*

Specifically, we study Adam and GD for training neural networks with weight decay regularization on an image-like data model, and demonstrate the difference between Adam and GD from a feature learning perspective. We consider a model where the data are generated as a combination of feature and noise patches under certain sparsity conditions, and analyze the convergence and generalization of Adam and GD for training a two-layer convolutional neural network (CNN). The contributions of this paper are summarized as follows.

- We establish global convergence guarantees for Adam and GD with proper weight decay regularization. We show that, starting at the same random initialization, Adam and GD can both train a two-layer convolutional neural network to achieve zero training error after polynomially many iterations, despite the nonconvex optimization landscape.
- We further show that GD and Adam in fact converge to different global solutions with different generalization performance: GD can achieve nearly zero test error, while the generalization performance of the model found by Adam is no better than a random guess. In particular, we show that the reason for this gap is due to the different training behaviors of Adam and GD: Adam is more likely to fit noises in the data and output a model that is largely contributed by the noise patches of the training data; GD prefers to fit training data based on their feature patch and finds a solution that is mainly composed by the true features. We also illustrate such different training processes in Figure 1, where it can be seen that the model trained by Adam is clearly more "noisy" than that trained by SGD.
- We also show that for convex settings with weight decay regularization, both Adam and gradient descent converge to the exact same solution and therefore have no test error difference. This suggests that the difference between Adam and GD cannot be fully explained by linear models or neural networks trained in the "almost convex" neural tangent kernel (NTK) regime Jacot et al. (2018); Allen-Zhu et al. (2019b); Du et al. (2019a); Zou et al. (2019); Allen-Zhu et al. (2019a); Arora et al. (2019a;b); Cao & Gu (2019); Ji & Telgarsky (2020); Chen et al. (2021). It also demonstrates that the inferior generalization performance of Adam is fundamentally tied to the nonconvex landscape of deep learning optimization, and cannot be solved by adding regularization.

**Notation.** For a scalar $x$, we use $[x]_+$ to denote $\max\{x, 0\}$. For a vector $\mathbf{v} = (v_1, \ldots, v_d)^\top$, we denote by $\|\mathbf{v}\|_2 := \left(\sum_{j=1}^d v_j^2\right)^{1/2}$ its $\ell_2$-norm, and use $\mathrm{supp}(\mathbf{v}) := \{j : v_j \neq 0\}$ to denote its support.

## 2 RELATED WORK

In this section, we discuss the works that are mostly related to our paper.

**Generalization gap between Adam and (stochastic) gradient descent.** The worse generalization of Adam compared with SGD has also been observed by some recent works and has motivated new variants of neural network training algorithms. Keskar & Socher (2017) proposed to switch between Adam and SGD to achieve better generalization. Merity et al. (2018) proposed a variant of the averaged stochastic gradient method to achieve good generalization performance for LSTM language models. Luo et al. (2019) proposed to use dynamic bounds on learning rates to achieve a smooth transition from adaptive methods to SGD to improve generalization. Our theoretical results for GD and Adam can also provide theoretical insights into the effectiveness of these empirical studies.

**Optimization and generalization guarantees in deep learning.** Our work is also closely related to the recent line of work studying the optimization and generalization guarantees of neural networks. A series of results have shown the convergence (Jacot et al., 2018; Li & Liang, 2018; Du et al., 2019b; Allen-Zhu et al., 2019b; Du et al., 2019a; Zou et al., 2019) and generalization (Allen-Zhu et al., 2019c;a; Arora et al., 2019a;b; Cao & Gu, 2019; Ji & Telgarsky, 2020; Chen et al., 2021) guarantees in the so-called "neural tangent kernel" (NTK) regime, where the neural network function is approximately linear in its parameters. Allen-Zhu & Li (2019); Bai & Lee (2019); Allen-Zhu & Li (2020a); Li et al. (2020) studied the learning of neural networks beyond the NTK regime. Our analysis in this paper is also beyond NTK, and gives a detailed comparison between GD and Adam.

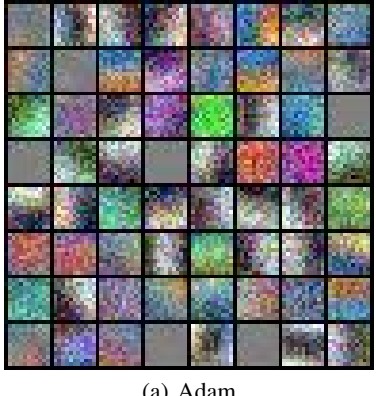 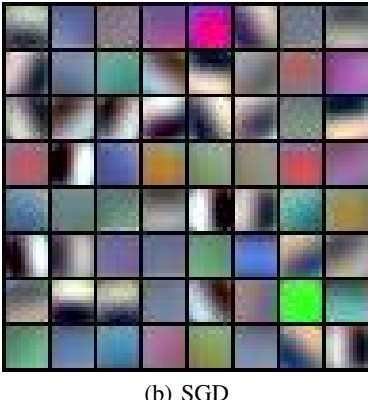

(a) Adam                                   (b) SGD

Figure 1: Visualization of the first layer of AlexNet trained by Adam and SGD on the CIFAR-10 dataset. Both algorithms are run for 100 epochs with weight decay regularization and standard data augmentations, but without batch normalization. Clearly, the model learned by Adam is more "noisy" than that learned by SGD, implying that Adam is more likely to overfit the noise in the training data.

**Feature learning by neural networks.** This paper is also closely related to several recent works that studied how neural networks can learn features. Allen-Zhu & Li (2020b) showed that adversarial training purifies the learned features by removing certain "dense mixtures" in the hidden layer weights of the network. Allen-Zhu & Li (2020c) studied how ensemble and knowledge distillation work in deep learning when the data have "multi-view" features. This paper studies a different aspect of feature learning by Adam and GD, and shows that GD can learn the features while Adam may fail even with proper regularization.

## 3 PROBLEM SETUP AND PRELIMINARIES

We consider learning a CNN with Adam and GD based on $n$ independent training examples $\{(\mathbf{x}_i, y_i)\}_{i=1}^n$ generated from a data model $\mathcal{D}$. In the following. we first introduce our data model $\mathcal{D}$, and then explain our neural network model and the details of the training algorithms.

**Data model.** We consider a data model where the data inputs consist of feature and noise patches. Such a data model is motivated by image classification problems where the label of an image usually only depends on part of an image, and the other parts of the image showing random objects, or features that belong to other classes, can be considered as noises. When using CNN to fit the data, the convolution operation is applied to each patch of the data input separately. We claim that our data model is more practical than those considered in Wilson et al. (2017); Reddi et al. (2018), which are handcrafted for showing the failure of Adam in term of either convergence or generalization. For simplicity, we only consider the case where the data consists of one feature patch and one noise patch. However, our result can be easily extended to cover the setting where there are multiple feature/noise patches. The detailed definition of our data model is given in Definition 3.1 as follows.

**Definition 3.1.** Let $d = \Omega(n^4)$, each data $(\mathbf{x}, y)$ with $\mathbf{x} \in \mathbb{R}^{2d}$ and $y \in \{-1, 1\}$ is generated as follows,

$$\mathbf{x} = [\mathbf{x}_1^\top, \mathbf{x}_2^\top]^\top,$$

where one of $\mathbf{x}_1$ and $\mathbf{x}_2$ denotes the feature patch that consists of a feature vector $y \cdot \mathbf{v}$, which is assumed to be 1-sparse, and the other one denotes the noise patch and consists of a noise vector $\boldsymbol{\xi}$. Without loss of generality, we assume $\mathbf{v} = [1, 0, \dots, 0]^\top$. The noise vector $\boldsymbol{\xi}$ is generated according to the following process:

1. Randomly select $s$ coordinates from $[d] \setminus \{1\}$ with equal probabilities, which is denoted as a vector $\mathbf{s} \in \{0, 1\}^d$.

2. Generate $\boldsymbol{\xi}$ from distribution $\mathcal{N}(\mathbf{0}, \sigma_p^2 \mathbf{I})$, and then mask off the first coordinate and other $d - s - 1$ coordinates, i.e., $\boldsymbol{\xi} = \boldsymbol{\xi} \odot \mathbf{s}$.

3. Add feature noise to $\boldsymbol{\xi}$, i.e., $\boldsymbol{\xi} = \boldsymbol{\xi} - \alpha y \mathbf{v}$, where $0 < \alpha < 1$ is the strength of the feature noise.

In particular, throughout this paper we set $s = \Theta\left(\frac{d^{1/2}}{n^2}\right)$, $\sigma_p^2 = \Theta\left(\frac{1}{s \cdot \text{polylog}(n)}\right)$ and $\alpha = \Theta\left(\sigma_p \cdot \text{polylog}(n)\right)$.

The most natural way to think of our data model is to treat $\mathbf{x}$ as the output of some intermediate layer of a CNN. In literature, Papyan et al. (2017) pointed out that the outputs of an intermediate layer of a CNN are usually sparse. Yang (2019) also discussed the setting where the hidden nodes in such an intermediate layer are sampled independently. This motivates us to study sparse features and entry-wisely independent noises in our model. In this paper, we focus on the case where the feature vector $\mathbf{v}$ is 1-sparse and the noise vector is $s$-sparse for simplicity. However, these sparsity assumptions can be generalized to the settings where the feature and the noises are denser.

Moreover, we would like to clarify that the data distribution considered in our paper is an **extreme case** where we assume there is only one feature vector and all data has a feature noise, since we believe this is the simplest model that captures the fundamental difference between Adam and SGD. With this data model, we aim to show why Adam and SGD perform differently. Our theoretical results and analysis techniques can also be extended to more practical settings where there are multiple feature vectors and multiple patches, each data can either contain a single feature or multiple features, together with pure random noise or feature noise.

**Two-layer CNN model.** We consider a two-layer CNN model $F$ using truncated polynomial activation function $\sigma(z) = (\max\{0, z\})^q$ and fixed the weights of second layer to be all 1's, where $q \geq 3$. Mathematically, given the data $(\mathbf{x}, y)$, the $j$-th output of the CNN can be formulated as

$$F_j(\mathbf{W}, \mathbf{x}) = \sum_{r=1}^m \left[\sigma(\langle \mathbf{w}_{j,r}, \mathbf{x}_1 \rangle) + \sigma(\langle \mathbf{w}_{j,r}, \mathbf{x}_2 \rangle)\right] = \sum_{r=1}^m \left[\sigma(\langle \mathbf{w}_{j,r}, y \cdot \mathbf{v} \rangle) + \sigma(\langle \mathbf{w}_{j,r}, \boldsymbol{\xi} \rangle)\right], \quad (3.1)$$

where $m$ is the width of the network, $\mathbf{w}_{j,r} \in \mathbb{R}^d$ denotes the weight at the $r$-th neuron, and $\mathbf{W}$ is the collection of model weights. Here the use of the polynomial ReLU activation function is for simplifying our analysis. It can be replaced by a smoothed ReLU activation function (e.g., the activation function used in Allen-Zhu & Li (2020c)). If we assume the input data distribution is Gaussian, we can also deal with ReLU activation function (Li et al., 2020). Moreover, we would like to emphasize that $\mathbf{x}_1$ and $\mathbf{x}_2$ denote two data patches, which are randomly assigned with feature vector or noise vector independently for each data point. The leaner has no knowledge about which one is the feature patch (or noise patch).

In this paper we assume the width of the network is polylogarithmic in the training sample size, i.e., $m = \text{polylog}(n)$. We assume $j \in \{-1, 1\}$ in order to make the logit index be consistent with the data label. Moreover, we assume that the each weight is initialized from a random draw of Gaussian random variable $\sim N(0, \sigma_0^2)$ with $\sigma_0 = \Theta\left(d^{-1/4}\right)$.

**Training objective.** Given the training data $\{(\mathbf{x}_i, y_i)\}_{i=1,\ldots,n}$, we consider to learn the model parameter $\mathbf{W}$ by optimizing the empirical loss function with weight decay regularization

$$L(\mathbf{W}) = \frac{1}{n} \sum_{i=1}^n L_i(\mathbf{W}) + \frac{\lambda}{2} \|\mathbf{W}\|_F^2, \quad (3.2)$$

where $L_i(\mathbf{W}) = -\log \frac{e^{F_{y_i}(\mathbf{W}, \mathbf{x}_i)}}{\sum_{j \in \{-1, 1\}} e^{F_j(\mathbf{W}, \mathbf{x}_i)}}$ denotes the individual loss for the data $(\mathbf{x}_i, y_i)$ and $\lambda \geq 0$ is the regularization parameter. In particular, the regularization parameter can be arbitrary as long as it satisfies $\lambda \in (0, \lambda_0)$ with $\lambda_0 = \Theta\left(\frac{1}{d^{(q-1)/4} n \cdot \text{polylog}(n)}\right)$. We claim that the $\lambda_0$ is the largest feasible regularization parameter that the training process will not stuck at the origin point (recall that $L(\mathbf{W})$ admits zero gradient at $\mathbf{W} = \mathbf{0}$.)

**Training algorithms.** In this paper, we consider gradient descent and Adam with full gradient. In particular, starting from initialization $\mathbf{W}^{(0)} = \{\mathbf{w}_{j,r}^{(0)}, j = \{\pm 1\}, r \in [m]\}$, the gradient descent update rule is

$$\mathbf{w}_{j,r}^{(t+1)} = \mathbf{w}_{j,r}^{(t)} - \eta \cdot \nabla_{\mathbf{w}_{j,r}} L(\mathbf{W}^{(t)}),$$

where $\eta$ is the learning rate. Meanwhile, Adam store historical gradient information in the momentum $\mathbf{m}^{(t)}$ and a vector $\mathbf{v}^{(t)}$ as follows

$$\mathbf{m}_{j,r}^{(t+1)} = \beta_1 \mathbf{m}_{j,r}^{(t)} + (1 - \beta_1) \cdot \nabla_{\mathbf{w}_{j,r}} L(\mathbf{W}^{(t)}), \quad (3.3)$$

$$\mathbf{v}_{j,r}^{(t+1)} = \beta_2 \mathbf{v}_{j,r}^{(t)} + (1 - \beta_2) \cdot [\nabla_{\mathbf{w}_{j,r}} L(\mathbf{W}^{(t)})]^2, \tag{3.4}$$

and entry-wisely adjusts the learning rate:

$$\mathbf{w}_{j,r}^{(t+1)} = \mathbf{w}_{j,r}^{(t)} - \eta \cdot \mathbf{m}_{j,r}^{(t)} / \sqrt{\mathbf{v}_{j,r}^{(t)}}, \tag{3.5}$$

where $\beta_1, \beta_2$ are the hyperparameters of Adam (a popular choice in practice is $\beta_1 = 0.9$, and $\beta_2 = 0.99$), and in (3.4) and (3.5), the square $(\cdot)^2$, square root $\sqrt{\cdot}$, and division $\cdot/\cdot$ all denote entry-wise calculations. We do not consider the initialization bias correction in the original Adam paper (Kingma & Ba, 2015) for the ease of analysis.

## 4 MAIN RESULTS

In this section we will state the main theorems in this paper. We first provide the learning guarantees of Adam and Gradient descent for training a two-layer CNN model in the following theorem. Recall that in this setting the training objective is nonconvex.

**Theorem 4.1** (Nonconvex setting). Consider a two-layer CNN defined in (3.1) with $d = \Omega(n^4)$ and regularized training objective (3.2) with a regularization parameter $\lambda > 0$, suppose the network width is $m = \text{polylog}(n)$ and the data distribution follows Definition 3.1, then we have the following guarantees on the training and test errors for the models trained by Adam and Gradient descent:

1. Suppose we run **Adam** for $T = \frac{\text{poly}(n)}{\eta}$ iterations with $\eta = \frac{1}{\text{poly}(n)}$, then with probability at least $1 - O(n^{-1})$, we can find a NN model $\mathbf{W}_{\text{Adam}}^*$ such that $\|\nabla L(\mathbf{W}_{\text{Adam}}^*)\|_1 \leq \frac{1}{T\eta}$. Moreover, the model $\mathbf{W}_{\text{Adam}}^*$ also satisfies:

   - Training error is zero: $\frac{1}{n} \sum_{i=1}^n \mathbb{1}\left[F_{y_i}(\mathbf{W}_{\text{Adam}}^*, \mathbf{x}_i) \leq F_{-y_i}(\mathbf{W}_{\text{Adam}}^*, \mathbf{x}_i)\right] = 0$.
   - Test error is high: $\mathbb{P}_{(\mathbf{x},y)\sim\mathcal{D}}\left[F_y(\mathbf{W}_{\text{Adam}}^*, \mathbf{x}) \leq F_{-y}(\mathbf{W}_{\text{Adam}}^*, \mathbf{x})\right] \geq \frac{1}{2}$.

2. Suppose we run **gradient descent** for $T = \frac{\text{poly}(n)}{\eta}$ iterations with learning rate $\eta = \frac{1}{\text{poly}(n)}$, then with probability at least $1 - O(n^{-1})$, we can find a NN model $\mathbf{W}_{\text{GD}}^*$ such that $\|\nabla L(\mathbf{W}_{\text{GD}}^*)\|_F^2 \leq \frac{1}{T\eta}$. Moreover, the model $\mathbf{W}_{\text{GD}}^*$ also satisfies:

   - Training error is zero: $\frac{1}{n} \sum_{i=1}^n \mathbb{1}\left[F_{y_i}(\mathbf{W}_{\text{GD}}^*, \mathbf{x}_i) \leq F_{-y_i}(\mathbf{W}_{\text{GD}}^*, \mathbf{x}_i)\right] = 0$.
   - Test error is nearly zero: $\mathbb{P}_{(\mathbf{x},y)\sim\mathcal{D}}\left[F_y(\mathbf{W}_{\text{GD}}^*, \mathbf{x}) \leq F_{-y}(\mathbf{W}_{\text{GD}}^*, \mathbf{x})\right] = \frac{1}{\text{poly}(n)}$.

From the optimization perspective, Theorem 4.1 shows that both Adam and GD can be guaranteed to find a point with a very small gradient, which can also achieve zero classification error on the training data. Moreover, it can be seen that given the same iteration number $T$ and learning rate $\eta$, Adam can be guaranteed to find a point with up to $1/(T\eta)$ gradient norm in $\ell_1$ metric, while gradient descent can only be guaranteed to find a point with up to $1/\sqrt{T\eta}$ gradient norm in $\ell_2$ metric. This suggests that Adam could enjoy a faster convergence rate compared to SGD in the training process, which is consistent with the practice findings. We would also like to point out that there is no contradiction between our result and the recent work (Reddi et al., 2019) showing that Adam can fail to converge, as the counterexample in Reddi et al. (2019) is for the online version of Adam, while we study the full batch Adam.

In terms of the test performance, their generalization abilities are largely different, even with weight decay regularization. In particular, the output of gradient descent can generalize well and achieve nearly zero test error, while the output of Adam gives nearly $1/2$ test error. In fact, this gap is due to two major aspects of the training process: (1) At the early stage of training where weight decay exhibits negligible effect, Adam and GD behave very differently. In particular, Adam prefers the data patch of lower sparsity and thus tends to fit the noise vectors $\boldsymbol{\xi}$, gradient descent prefers the data patch of larger $\ell_2$ norm and thus will learn the feature patch; (2) At the late stage of training where the weight decay regularization cannot be ignored, both Adam and gradient descent will be enforced to converge to a *local minimum* of the regularized objective, which maintains the pattern learned in the early stage. Consequently, the model learned by Adam will be biased towards the noise patch to fit the feature noise vector $-\alpha y\mathbf{v}$, which is opposite in direction to the true feature vector and therefore leads to a test error no better than a random guess. More details about the training behaviors of Adam and gradient descent are given in Section 5.

Theorem 4.1 shows that when optimizing a nonconvex training objective, Adam and gradient descent will converge to different global solutions with different generalization errors, even with weight decay regularization. In comparison, the following theorem gives the learning guarantees of Adam and gradient descent when optimizing convex and smooth training objectives (e.g., linear model $F(\mathbf{w}, \mathbf{x}) = \mathbf{w}^\top \mathbf{x}$ with logistic loss).

**Theorem 4.2** (Convex setting). For any convex and smooth training objective with positive regularization parameter $\lambda$, suppose we run **Adam** and **gradient descent** for $T = \frac{\text{poly}(n)}{\eta}$ iterations, then with probability at least $1 - n^{-1}$, the obtained parameters $\mathbf{W}^*_{\text{Adam}}$ and $\mathbf{W}^*_{\text{GD}}$ satisfy that $\|\nabla L(\mathbf{W}^*_{\text{Adam}})\|_1 \leq \frac{1}{T\eta}$ and $\|\nabla L(\mathbf{W}^*_{\text{Adam}})\|_2^2 \leq \frac{1}{T\eta}$ respectively. Moreover, let $F(\mathbf{W}, \mathbf{x}) \in \mathbb{R}$ be the output of the convex model with parameter $\mathbf{W}$ and input $\mathbf{x}$, it holds that:

- Training errors are both zero:

$$\frac{1}{n} \sum_{i=1}^n \mathbb{1}\left[\text{sgn}\left(F(\mathbf{W}^*_{\text{Adam}}, \mathbf{x}_i)\right) \neq y_i\right] = \frac{1}{n} \sum_{i=1}^n \mathbb{1}\left[\text{sgn}\left(F(\mathbf{W}^*_{\text{GD}}, \mathbf{x}_i)\right) \neq y_i\right] = 0.$$

- Test errors are nearly the same:

$$\mathbb{P}_{(\mathbf{x}, y) \sim \mathcal{D}}\left[\text{sgn}\left(F(\mathbf{W}^*_{\text{Adam}}, \mathbf{x}_i)\right) \neq y\right] = \mathbb{P}_{(\mathbf{x}, y) \sim \mathcal{D}}\left[\text{sgn}\left(F(\mathbf{W}^*_{\text{GD}}, \mathbf{x})\right) \neq y\right] \pm 1/\text{poly}(n).$$

Theorem 4.2 shows that when optimizing a convex and smooth training objective (e.g., a linear model with logistic loss) with weight decay regularization, both Adam and gradient can converge to almost the same solution and enjoy very similar generalization performance. Combining this result and Theorem 4.1, it is clear that the inferior generalization performance is closely tied to the nonconvex landscape of deep learning, and cannot be understood by standard weight decay regularization.

## 5 Proof Outline of the Main Results

In this section we provide the proof sketch of Theorem 4.1 and explain the different generalization abilities of the models found by gradient descent and Adam.

Before moving to the proof of main results, we first give the following lemma which shows that for data generated from the data distribution $\mathcal{D}$ in Definition 3.1, with high probability all noise vectors $\{\boldsymbol{\xi}_i\}_{i=1,\ldots,n}$ have nearly disjoint supports.

**Lemma 5.1.** Let $\{(\mathbf{x}_i, y_i)\}_{i=1,\ldots,n}$ be the training dataset generated by Definition 3.1. Moreover, recall that $\mathbf{x}_i = [y_i \mathbf{v}^\top, \boldsymbol{\xi}_i^\top]^\top$ (or $\mathbf{x}_i = [\boldsymbol{\xi}_i^\top, y_i \mathbf{v}^\top]^\top$), let $\mathcal{B}_i = \text{supp}(\boldsymbol{\xi}_i) \backslash \{1\}$ be the support of $\boldsymbol{\xi}_i$ except the first coordinate. Then with probability at least $1 - n^{-2}$, $\mathcal{B}_i \cap \mathcal{B}_j = \emptyset$ for all $i \neq j$.

This lemma implies that the optimization of each coordinate of the model parameter $\mathbf{W}$, except for the first one, is mostly determined by only one training data. Technically, this lemma can greatly simplify the analysis for Adam so that we can better illustrate its optimization behavior and explain the generalization performance gap between Adam and gradient descent.

**Proof outline.** For both Adam and gradient descent, we will show that the training process can be decomposed into two stages. In the first stage, which we call *pattern learning stage*, the weight decay regularization will be less important and can be ignored, while the algorithms tend to learn the pattern from the training data. In particular, we will show that the patterns learned by these two algorithms are different: Adam tends to fit the noise patch while gradient descent will mainly learn the feature patch. In the second stage, which we call it *regularization stage*, the effect of regularization cannot be neglected, which will regularize the algorithm to converge at some local stationary points. However, due to the nonconvex landscape of the training objective, the pattern learned in the first stage will remain unchanged, even when running an infinitely number of iterations.

### 5.1 Proof sketch for Adam

Recall that in each iteration of Adam, the model weight is updated by using a moving-averaged gradient, normalized by a moving average of the historical gradient squares. As pointed out in Balles & Hennig (2018); Bernstein et al. (2018), Adam behaves similarly to sign gradient descent (signGD) when using sufficiently small step size or the moving average parameters $\beta_1, \beta_2$ are nearly zero. This motivates us to study the optimization behavior of signGD and then extends it to Adam using their similarities. In particular, sign gradient descent updates the model parameter according to the

following rule:

$$\mathbf{w}_{j,r}^{(t+1)} = \mathbf{w}_{j,r}^{(t+1)} - \eta \cdot \mathrm{sgn}(\nabla_{\mathbf{w}_{j,r}} L(\mathbf{W}^{(t)})).$$

Recall that each data has two patches: feature patch and noise patch. By Lemma 5.1 and the data distribution (see Definition 3.1), we know that all noise vectors $\{\boldsymbol{\xi}_i\}_{i=1,\dots,n}$ are supported on disjoint coordinates, except the first one. For data point $\mathbf{x}_i$, let $\mathcal{B}_i$ denote its support, except the first coordinate. In the subsequent analysis, we will always assume that those $\mathcal{B}_i$'s are disjoint, i.e., $\mathcal{B}_i \cap \mathcal{B}_j = \emptyset$ if $i \neq j$.

Next we will characterize two aspects of the training process: *feature learning* and *noise memorization*. Mathematically, we will focus on two quantities: $\langle \mathbf{w}_{j,r}^{(t)}, j \cdot \mathbf{v} \rangle$ and $\langle \mathbf{w}_{y_i,r}^{(t)}, \boldsymbol{\xi}_i \rangle$. In particular, given the training data $(\mathbf{x}_i, y_i)$ with $\mathbf{x}_i = [y_i \mathbf{v}^\top, \boldsymbol{\xi}_i^\top]^\top$, larger $\langle \mathbf{w}_{y_i,r}^{(t)}, y_i \cdot \mathbf{v} \rangle$ implies better feature learning and larger $\langle \mathbf{w}_{y_i,r}^{(t)}, \boldsymbol{\xi}_i \rangle$ represents better noise memorization.

**Stage I: Learning the pattern.** Mathematically, the first stage is defined as the iterations that the neural network output is smaller than some constant. In this stage, all training data remains under-fitted and can provide large gradient for model training, and the effect of weight decay regularization can be ignored due to our choice of $\lambda$. We will show that in this stage the inner product $\langle \mathbf{w}_{y_i,r}^{(t)}, \boldsymbol{\xi}_i \rangle$ grows much faster than $\langle \mathbf{w}_{j,r}^{(t)}, j\mathbf{v} \rangle$ since feature learning only makes use of the first coordinate of the gradient, while noise memorization could take advantage of all the coordinates in $\mathcal{B}_i$ (note that $|\mathcal{B}_i| = s \gg 1$).

**Lemma 5.2** (General results in Stage I). Suppose the training data is generated according to Definition 3.1, assume $\lambda = o(\sigma_0^{q-2} \sigma_p / n)$ and $\eta = 1/\mathrm{poly}(d)$, then for any $t \leq T_0$ with $T_0 = \widetilde{O}\big(\frac{1}{\eta s \sigma_p}\big)$ and any $i \in [n]$,

$$\langle \mathbf{w}_{j,r}^{(t+1)}, j \cdot \mathbf{v} \rangle \leq \langle \mathbf{w}_{j,r}^{(t)}, j \cdot \mathbf{v} \rangle + \Theta(\eta),$$
$$\langle \mathbf{w}_{y_i,r}^{(t+1)}, \boldsymbol{\xi}_i \rangle = \langle \mathbf{w}_{y_i,r}^{(t)}, \boldsymbol{\xi}_i \rangle + \widetilde{\Theta}(\eta s \sigma_p).$$

Since $\langle \mathbf{w}_{j,r}^{(t)}, \boldsymbol{\xi}_i \rangle$ enjoys much faster increasing rate than that of $\langle \mathbf{w}_{j,r}^{(t)}, j \cdot \mathbf{v} \rangle$, after a certain number of iterations, the learning of noise patch will dominate the learning of feature patch. Thus, the model will tend to fit the feature noise in the noise patch, leading to a flipped feature learning phenomenon.

**Lemma 5.3** (Flipping the feature learning). Suppose the training data is generated according to Definition 3.1, $\alpha \geq \widetilde{\Theta}\big((s\sigma_p)^{1-q} \vee \sigma_0^{q-1}\big)$ and $\sigma_0 < \widetilde{O}\big((s\sigma_p)^{-1}\big)$, then for any $t \in [T_r, T_0]$ with $T_r = \widetilde{O}\big(\frac{\sigma_0}{\eta s \sigma_p \alpha^{1/(q-1)}}\big) \leq T_0$,

$$\langle \mathbf{w}_{j,r}^{(t+1)}, j \cdot \mathbf{v} \rangle = \langle \mathbf{w}_{j,r}^{(t)}, j \cdot \mathbf{v} \rangle - \Theta(\eta).$$

Moreover, it holds that

$$\mathbf{w}_{j,r}^{(T_0)}[k] = \begin{cases} -\mathrm{sgn}(j) \cdot \widetilde{\Omega}\big(\frac{1}{s\sigma_p}\big), & k = 1, \\ \mathrm{sgn}(\boldsymbol{\xi}_i[k]) \cdot \widetilde{\Omega}\big(\frac{1}{s\sigma_p}\big) \text{ or } \pm\widetilde{O}(\eta), & k \in \mathcal{B}_i, \text{ with } y_i = j, \\ \pm\widetilde{O}(\eta), & \text{otherwise.} \end{cases}$$

From Lemma 5.3 it can be observed that at the iteration $T_0$, the sign of the first coordinate of $\mathbf{w}_{j,r}^{(T_0)}$ is different from that of the true feature, i.e., $j \cdot \mathbf{v}$. This implies that at the end of the first training stage, the model is biased towards the noise patch to fit the feature noise.

**Stage II: Regularizing the model.** In this stage, as the neural network output becomes larger, part of training data starts to be well fitted and gives smaller gradient. As a consequence, the feature learning and noise memorization processes will be slowed down and the weight decay regularization term cannot be ignored. However, although weight decay regularization can prevent the model weight from being too large, it will maintain the pattern learned in Stage I and cannot push the model back to "forget" the noise and learn the feature and stops at some local stationary points. We summarize these results in the following lemma.

**Lemma 5.4** (Maintain the pattern). If $\alpha = O(s\sigma_p^2/n)$ and $\eta = o(\lambda)$, then let $r^* = \arg\max_{r \in [m]} \langle \mathbf{w}_{y_i,r}^{(t)}, \boldsymbol{\xi}_i \rangle$, for any $t \geq T_0$, $i \in [n]$, $j \in [2]$ and $r \in [m]$, it holds that

$$\langle \mathbf{w}_{y_i,r^*}^{(t)}, \boldsymbol{\xi}_i \rangle = \widetilde{\Theta}(1), \quad \sum_{k \in \mathcal{B}_i} |\mathbf{w}_{y_i,r^*}^{(t)}[k]| \cdot |\boldsymbol{\xi}_i[k]| = \widetilde{\Theta}(1),$$

$$\forall r \in [m], \quad \langle \mathbf{w}_{j,r}^{(t)}, \mathrm{sgn}(j) \cdot \mathbf{v} \rangle \in [-o(1), O(\lambda^{-1}\eta)].$$

Lemma 5.4 shows that in the second stage, $\langle \mathbf{w}_{y_i,r}^{(t)}, \boldsymbol{\xi}_i \rangle$ will always be large while $\langle \mathbf{w}_{y_i,r}^{(t)}, y_i \cdot \mathbf{v} \rangle$ is still negative, or positive but extremely small. Next we will show that within polynomial steps, the algorithm can be guaranteed to find a point with small gradient.

**Lemma 5.5** (Convergence guarantee). If $\eta = O(d^{-1/2})$, then for any $t$ it holds that

$$L(\mathbf{W}^{(t+1)}) - L(\mathbf{W}^{(t)}) \leq -\eta \|\nabla L(\mathbf{W}^{(t)})\|_1 + \widetilde{\Theta}(\eta^2 d).$$

Lemma 5.5 shows that we can pick a sufficiently small $\eta$ and $T = \mathrm{poly}(n)/\eta$ to ensure that the algorithm can find a point with up to $O(1/(T\eta))$ in $\ell_1$ norm. Then we can show that given the results in Lemma 5.4, the formula of the algorithm output $\mathbf{W}^*$ can be precisely characterized, which we can show that $\langle \mathbf{w}_{y_i,r}^*, y_i \cdot \mathbf{v} \rangle < 0$. This implies that the output model will be biased to fit the feature noise $-\alpha y\mathbf{v}$ but not the true one $\mathbf{v}$. Then when it comes to a fresh test example the model will fail to recognize its true feature. Also note that the noise in the test data is nearly independent of the noise in training data. Consequently, the model will not be able to identify the label of the test data and therefore cannot be better than a random guess.

## 5.2 Proof Sketch for Gradient Descent

Similar to the proof for Adam, we also decompose the entire training process into two stages.

**Stage I: Learning the pattern.** In this stage the gradient from training loss function is large and and the effect of regularization can be ignored. Unlike Adam that is sensitive to the sparsity of the feature vector or noise vector, gradient descent is more focusing on the $\ell_2$ norm of them, where the vector (which can be either feature vector or noise vector) with larger $\ell_2$ norm is more likely to be discovered and learnt by GD. Note that the feature vector has a larger $\ell_2$ norm than the noise, we can show that, in the following lemma, gradient descent will learn the feature vector very quickly, while barely tend to memorize the noise.

**Lemma 5.6.** Let $\Lambda_j^{(t)} = \max_{r \in [m]} \langle \mathbf{w}_{j,r}^{(t+1)}, j \cdot \mathbf{v} \rangle$, $\Gamma_{j,i}^{(t)} = \max_{r \in [m]} \langle \mathbf{w}_{j,r}^{(t)}, \boldsymbol{\xi}_i \rangle$, and $\Gamma_j^{(t)} = \max_{i:y_i=j} \Gamma_{j,i}^{(t)}$. Let $T_j$ be the iteration number that $\Lambda_j^{(t)}$ reaches $\Theta(1/m) = \widetilde{\Theta}(1)$, then we have

$$T_j = \widetilde{\Theta}(\sigma_0^{2-q}) \quad \text{for all } j \in \{-1, 1\}.$$

Moreover, let $T_0 = \max_j \{T_j\}$, then for all $t \leq T_0$ it holds that $\Gamma_j^{(t)} = \widetilde{O}(\sigma_0)$ for all $j \in \{-1, 1\}$.

**Stage II: Regularizing the model.** Similar to Lemma 5.4, we show that in the second stage at which the impact of weight decay regularization cannot be ignored, the pattern of the training data learned in the first stage will remain unchanged.

**Lemma 5.7.** If $\eta \leq O(\sigma_0)$, it holds that $\Lambda_j^{(t)} = \widetilde{\Theta}(1)$ and $\Gamma_j^{(t)} = \widetilde{O}(\sigma_0)$ for all $t \geq \min_j T_j$.

The following lemma further shows that within polynomial steps, gradient descent is guaranteed to find a point with small gradient.

**Lemma 5.8.** If the learning rate satisfies $\eta = o(1)$, then for any $t \geq 0$ it holds that

$$L(\mathbf{W}^{(t+1)}) - L(\mathbf{W}^{(t)}) \leq -\frac{\eta}{2} \|\nabla L(\mathbf{W}^{(t)})\|_F^2.$$

Lemma 5.8 shows that we can pick a sufficiently small $\eta$ and $T = \mathrm{poly}(n)/\eta$ to ensure that gradient descent can find a point with up to $O(1/(T\eta)^{1/2})$ in $\ell_2$ norm. By Lemma 5.7, it is clear that the output model of GD can well learn the feature vector while memorizing nearly nothing from the noise vectors, which can therefore achieve nearly zero test error.

## 6 Experiments

In this section we perform numerical experiments on the synthetic data generated according to Definition 3.1 to verify our main results. In particular, we set the problem dimension $d = 1000$, the

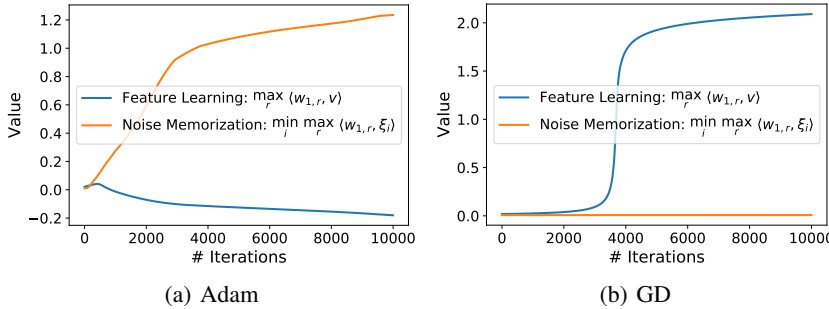

(a) Adam                                         (b) GD

Figure 2: Visualization of the feature learning ($\max_r\langle\mathbf{w}_{1,r},\mathbf{v}\rangle$) and noise memorization ($\min_i\max_r\langle\mathbf{w}_{1,r},\boldsymbol{\xi}_i\rangle$) in the training process.

training sample size $n = 200$ (100 positive examples and 100 negative examples), feature vector $\mathbf{v} = [1, 0, \ldots, 0]^\top$, noise sparsity $s = 0.1d = 100$, standard deviation of noise $\sigma_p = 1/s^{1/2} = 0.1$, feature noise strength $\alpha = 0.2$, initialization scaling $\sigma_0 = 0.01$, regularization parameter $\lambda = 1 \times 10^{-5}$, network width $m = 20$, activation function $\sigma(z) = \max\{0, z\}^3$, total iteration number $T = 1 \times 10^4$, and the learning rate $\eta = 5 \times 10^{-5}$ for Adam (default choices of $\beta_1$ and $\beta_2$ in pytorch), $\eta = 0.02$ for GD.

We first report the training error and test error achieved by the solutions found by SGD and Adam in Table 2, where the test error is calculated on a test dataset of size $10^4$. It is clear that both Adam and SGD can achieve zero training error, while they have entirely different results on the test data: SGD generalizes well and achieve zero test error; Adam generalizes worse than SGD and gives $> 0.5$ test error, which verifies our main result (Theorem 4.1).

Moreover, we also calculate the inner products: $\max_r\langle\mathbf{w}_{1,r},\mathbf{v}\rangle$ and $\min_i\max_r\langle\mathbf{w}_{1,r},\boldsymbol{\xi}_i\rangle$, representing feature learning and noise memorization respectively, to verify our key lemmas. Here we only consider positive examples as the results for negative examples are similar. The results are reported in Figure 2. For Adam, from Figure 2(a), it can be seen that the algorithm will perform feature learning in the first few iterations and then entirely forget

| Algorithm | Adam | SGD |
|---|---|---|
| Training error | 0 | 0 |
| Test error | 0.884 | 0 |

Table 2: Training and test errors achieved by GD and Adam.

the feature (but fit feature noise), i.e., the feature learning is flipped, which verifies Lemma 5.3. In the meanwhile, the noise memorization happens in the entire training process and enjoys much faster rate than feature learning, which verifies Lemma 5.2. In addition, we can also observe that there are two stages for the increasing of $\min_i\max_r\langle\mathbf{w}_{1,r},\boldsymbol{\xi}_i\rangle$: in the first stage $\min_i\max_r\langle\mathbf{w}_{1,r},\boldsymbol{\xi}_i\rangle$ increases linearly, and in the second stage its increasing speed gradually slows down and $\min_i\max_r\langle\mathbf{w}_{1,r},\boldsymbol{\xi}_i\rangle$ will remain in a constant order. This verifies Lemma 5.2 and Lemma 5.4. For GD, from Figure 2(b), it can be seen that the feature learning will dominate the noise memorization: feature learning will increases to a constant in the first stage and then remains in a constant order in the second stage; noise memorization will keep in a low level which is nearly the same as that at the initialization. This verifies Lemmas 5.6 and 5.7.

## 7 CONCLUSION AND FUTURE WORK

In this paper, we study the generalization of Adam and compare it with gradient descent. We show that when training neural networks, Adam and GD starting from the same initialization can converge to different global solutions of the training objective with significantly different generalization errors, even with proper regularization. Our analysis reveals the fundamental difference between Adam and GD in learning features, and demonstrates that this difference is tied to the nonconvex optimization landscape of neural networks. Built up on the results in this paper, there are several important research directions. First, our current result is for two-layer networks. Extending the results to deep networks could be an important next step, where we will not only look at the input data but also consider the output of each intermediate layer as "input". Second, our current data model is motivated by the image data, where Adam has been observed to perform worse than SGD in terms of generalization. Studying other types of data such as natural language data, where Adam is often observed to perform better than SGD, is another future work direction.

## ETHICS STATEMENT

This paper mainly concerns the theoretical understanding of the behaviors of different optimization algorithms in deep learning. We don't see any potential ethical issues in our work.

## REPRODUCIBILITY STATEMENT

Our theoretical results can be reproduced according to the assumptions and problem set up stated in Section 3 and the detailed proof provided in the appendix. The experimental results shown in Table 1 and Figure 1 are obtained via standard training (using SGD or Adam) on CIFAR-10 dataset (Krizhevsky et al., 2009). Our experimental results in Figure 2 and Table 2 can also be reproduced according to hyperparameters described in Section 6.

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

## A    DISCUSSIONS

In this section, we would like to reflect on some possible extensions of our work to more general settings.

**Mini-batch stochastic gradients.** One natural extension of our paper is proving the separation between mini-batch SGD and mini-batch Adam, which we believe is not difficult. In particular, it can be shown that the learning behavior of SGD will be largely similar to that of GD since using mini-batch stochastic gradients does not change the rates of feature learning and noise memorization. In contrast, compared to full-batch Adam, the rate of mini-batch Adam in memorizing noise will be scaled down by at most $n$ times (due to the preconditioning in Adam), while the rate of feature learning remains the same. However, the difference between SGD and mini-batch Adam still exists (though the difference might be smaller). This is because, in our paper, the separation between Adam and GD is characterized by a $\mathrm{poly}(d)$ factor: the speed of feature learning in Adam and GD, and the rate of noise memorization in GD are both in the order of $O(\eta)$ (in each step), while the rate of noise memorization in Adam is proportional to the number of nonzero entries, which is in the order of $\eta \cdot \mathrm{poly}(d)$. In contrast, using mini-batch stochastic gradients can only reduce the difference by at most an $O(n)$ factor, which is still dominated by the $\mathrm{poly}(d)$ separation as long as the dimension $d$ is sufficiently large (overparameterized).

**Learnable second layer.** Our results and analyses can also be extended to the case where the second layer is trained. The reason behind this is that the difference between Adam and GD comes from the learning of the hidden layer, while the second layer weights will not affect which patch and neuron is more likely to be learned by different optimization algorithms. Consequently, we can still get the result that GD tends to learn the feature patch while Adam tends to learn the noise patch.

## B    PROOF OF THEOREM 4.1: NONCONVEX CASE

In the beginning of the proof we first present the following useful lemma.

### B.1    PRELIMINARIES

We first recall the magnitude of all parameters:

$$d = \mathrm{poly}(n),\ \eta = \frac{1}{\mathrm{poly}(n)},\ s = \Theta\left(\frac{d^{1/2}}{n^2}\right),\ \sigma_p^2 = \Theta\left(\frac{1}{s \cdot \mathrm{polylog}(n)}\right),\ \sigma_0^2 = \Theta\left(\frac{1}{d^{1/2}}\right),$$

$$m = \mathrm{polylog}(n),\ \alpha = \Theta(\sigma_p \cdot \mathrm{polylog}(n)),\ \lambda = O\left(\frac{1}{d^{(q-1)/4}n \cdot \mathrm{polylog}(n)}\right).$$

Here $\mathrm{poly}(n)$ denotes a polynomial function of $n$ with degree of a sufficiently large constant, $\mathrm{poly}(n)$ denotes a polynomial function of $\log(n)$ with degree of a sufficiently large constant. Based on the parameter configuration, we claim that the following equations hold, which will be frequently used in the subsequent proof.

$$\lambda = o\left(\frac{\sigma_0^{q-2}\sigma_p}{n}\right),\ \alpha = \omega\big((s\sigma_p)^{1-q}\sigma_0^{q-1}\big),\ \sigma_0 = o\left(\frac{1}{s\sigma_p}\right),\ \alpha = o\left(\frac{s\sigma_p^2}{n}\right),\ \eta = o\big(\lambda\sigma_0^q\sigma_p^q\big).$$

**Lemma B.1** (Non-overlap support). *Let $\{(\mathbf{x}_i, y_i)\}_{i=1,\dots,n}$ be the training dataset sampled according to Definition 3.1. Moreover, let $\mathcal{B}_i = \mathrm{supp}(\boldsymbol{\xi}_i)\backslash\{1\}$ be the support of $\mathbf{x}_i$ except the first coordinate[1]. Then with probability at least $1 - n^{-2}$, $\mathcal{B}_i \cap \mathcal{B}_j = \emptyset$ for all $i, j \in [n]$.*

*Proof of Lemma B.1.* For any fixed $k \in [n]$ and $j \in \mathrm{supp}(\boldsymbol{\xi}_k)\backslash\{1\}$, by the model assumption we have

$$\mathbb{P}\{(\boldsymbol{\xi}_i)_j \neq 0\} = s/(d-1),$$

for all $i \in [n]\backslash\{k\}$. Therefore by the fact that the data samples are independent, we have

$$\mathbb{P}(\exists i \in [n]\backslash\{k\} : (\xi_i)_j \neq 0) = 1 - [1 - s/(d-1)]^n.$$

Applying a union bound over all $k \in [n]$ and $j \in \mathrm{supp}(\boldsymbol{\xi}_k)\backslash\{1\}$, we obtain

$$\mathbb{P}(\exists k \in [n], j \in \mathrm{supp}(\boldsymbol{\xi}_k)\backslash\{1\}, i \in [n]\backslash\{k\} : (\boldsymbol{\xi}_i)_j \neq 0) \leq n \cdot s \cdot \{1 - [1 - s/(d-1)]^n\}. \quad \text{(B.1)}$$

---

[1]Recall that all data inputs have nonzero first coordinate by Definition 3.1

By the data distribution assumption we have $s \leq \sqrt{d}/(2n^2)$, which clearly implies $s/(d-1) \leq 1/2$. Therefore we have

$$
\begin{aligned}
n \cdot s \cdot [1 - (1 - s/d)^n] &= n \cdot s \cdot \{1 - \exp[n \log(1 - s/(d-1))]\} \\
&\leq n \cdot s \cdot [1 - \exp(n \cdot 2s/(d-1))] \\
&\leq n \cdot s \cdot [1 - \exp(n \cdot 4s/d)] \\
&\leq n \cdot s \cdot (4ns/d) \\
&= 4n^2 s^2/d \\
&\leq n^{-2},
\end{aligned}
$$

where the first inequality follows by the inequalities $\log(1 - z) \geq -2z$ for $z \in [0, 1/2]$, the second inequality follows by $s/(d-1) \geq 2s/d$, the third inequality follows by the inequality $1 - \exp(-z) \leq z$ for $z \in \mathbb{R}$, and the last inequality follows by the assumption that $s \leq \sqrt{d}/(2n^2)$. Plugging the bound above into (B.1) finishes the proof.

$\square$

### B.2 PROOF FOR ADAM

Before moving to the detailed proof, we first state the update rules of feature learning and noise memorization when the sign gradient is applied.

$$
\begin{aligned}
\langle \mathbf{w}_{j,r}^{(t+1)}, j\mathbf{v} \rangle &= \langle \mathbf{w}_{j,r}^{(t)}, j\mathbf{v} \rangle - \eta \cdot \langle \mathrm{sgn}(\nabla_{\mathbf{w}_{j,r}} L(\mathbf{W}^{(t)})), j\mathbf{v} \rangle \\
&= \langle \mathbf{w}_{j,r}^{(t)}, j\mathbf{v} \rangle + j\eta \cdot \mathrm{sgn}\left( \sum_{i=1}^{n} y_i \ell_{j,i}^{(t)} \left[ \sigma'(\langle \mathbf{w}_{j,r}^{(t)}, y_i \mathbf{v} \rangle) - \alpha \sigma'(\langle \mathbf{w}_{j,r}^{(t)}, \boldsymbol{\xi}_i \rangle) \right] - n\lambda \mathbf{w}_{j,r}^{(t)}[1] \right),
\end{aligned}
$$

(B.2)

where $\ell_{j,i}^{(t)} := \mathbb{1}_{y_i=j} - \mathrm{logit}_j(F, \mathbf{x}_i)$ and $\mathrm{logit}_j(F, \mathbf{x}_i) = \frac{e^{F_j(\mathbf{W}, \mathbf{x}_i)}}{\sum_{k \in \{-1,1\}} e^{F_k(\mathbf{W}, \mathbf{x}_i)}}$. From (B.2) we can observe three terms in the signed gradient. Specifically, the first term represents the gradient over the feature patch, the second term stems from the feature noise term in the noise patch (see Definition 3.1), and the last term is the gradient of the weight decay regularization. On the other hand, the memorization of the noise vector $\boldsymbol{\xi}_i$ can be described by the following update rule,

$$
\begin{aligned}
\langle \mathbf{w}_{y_i,r}^{(t+1)}, \boldsymbol{\xi}_i \rangle &= \langle \mathbf{w}_{y_i,r}^{(t)}, \boldsymbol{\xi}_i \rangle - \eta \cdot \langle \mathrm{sgn}(\nabla_{\mathbf{w}_{y_i,r}} L(\mathbf{W}^{(t)})), \boldsymbol{\xi}_i \rangle \\
&= \langle \mathbf{w}_{y_i,r}^{(t)}, \boldsymbol{\xi}_i \rangle + \eta \cdot \sum_{k \in \mathcal{B}_i} \left\langle \mathrm{sgn}\left( \ell_{y_i,i}^{(t)} \sigma'(\langle \mathbf{w}_{y_i,r}^{(t)}, \boldsymbol{\xi}_i \rangle) \boldsymbol{\xi}_i[k] - n\lambda \mathbf{w}_{y_i,r}^{(t)}[k] \right), \boldsymbol{\xi}_i[k] \right\rangle \\
&\quad - \alpha y_i \eta \cdot \mathrm{sgn}\left( \sum_{i=1}^{n} y_i \ell_{y_i,i}^{(t)} \left[ \sigma'(\langle \mathbf{w}_{y_i,r}^{(t)}, y_i \mathbf{v} \rangle) - \alpha \sigma'(\langle \mathbf{w}_{y_i,r}^{(t)}, \boldsymbol{\xi}_i \rangle) \right] - n\lambda \mathbf{w}_{y_i,r}^{(t)}[1] \right).
\end{aligned}
$$

(B.3)

In this subsection we first provide the following lemma that shows for most of the coordinate (with slightly large gradient), the Adam update is similar to signGD update (up to some constant factors). In the remaining proof for Adam, we will largely apply this lemma to get a signGD-like result for Adam (similar to the technical lemmas in Section 5). Besides, the proofs for all lemmas in Section 5 can be viewed as a simplified version of the proofs for technical lemmas for Adam, thus are omitted in the paper.

**Lemma B.2** (Closeness to SignGD). Recall the update rule of Adam, let $\mathbf{W}^{(t)}$ be the $t$-th iterate of the Adam algorithm. Suppose that $\langle \mathbf{w}_{j,r}^{(t)}, \mathbf{v} \rangle, \langle \mathbf{w}_{j,r}^{(t)}, \boldsymbol{\xi}_i \rangle = \widetilde{\Theta}(1)$ for all $j \in \{\pm 1\}$ and $r \in [m]$. Then if $\beta_2 \geq \beta_1^2$, we have

- For all $k \in [d]$,

$$
\left| \frac{\mathbf{m}_{j,r}^{(t)}[k]}{\sqrt{\mathbf{v}_{j,r}^{(t)}[k]}} \right| \leq \Theta(1).
$$

- For every $k \notin \cup_{i=1}^{n} \mathcal{B}_i$ (including $k = 1$) we have either $|\nabla_{\mathbf{w}_{j,r}} L(\mathbf{W}^{(t)})[k]| \leq \widetilde{\Theta}(\eta)$ or

$$\frac{\mathbf{m}_{j,r}^{(t)}[k]}{\sqrt{\mathbf{v}_{j,r}^{(t)}[k]}} = \mathrm{sgn}\big(\nabla_{\mathbf{w}_{j,r}} L(\mathbf{W}^{(t)})[k]\big) \cdot \Theta(1).$$

- For every $k \in \mathcal{B}_i$, we have $|\nabla_{\mathbf{w}_{j,r}} L(\mathbf{W}^{(t)})[k]| \leq \widetilde{\Theta}\big(\eta n^{-1} s \sigma_p |\ell_{j,i}^{(t)}|\big) \leq \widetilde{\Theta}(\eta s \sigma_p)$ or

$$\frac{\mathbf{m}_{j,r}^{(t)}[k]}{\sqrt{\mathbf{v}_{j,r}^{(t)}[k]}} = \mathrm{sgn}\big(\nabla_{\mathbf{w}_{j,r}} L(\mathbf{W}^{(t)})[k]\big) \cdot \Theta(1).$$

*Proof.* First recall that the gradient $\nabla_{\mathbf{w}_{j,r}} L(\mathbf{W}^{(t)})$ can be calculated as

$$\nabla_{\mathbf{w}_{j,r}} L(\mathbf{W}^{(t)}) = -\frac{1}{n}\Bigg[ \sum_{i=1}^{n} y_i \ell_{j,i}^{(t)} \sigma'(\langle \mathbf{w}_{j,r}^{(t)}, y_i \mathbf{v}\rangle) \cdot \mathbf{v} + \sum_{i=1}^{n} \ell_{j,i}^{(t)} \cdot \sigma'(\langle \mathbf{w}_{j,r}^{(t)}, y_i \boldsymbol{\xi}_i\rangle) \cdot \boldsymbol{\xi}_i \Bigg] + \lambda \mathbf{w}_{j,r}^{(t)}.$$

More specifically, for the first coordinate of $\nabla_{\mathbf{w}_{j,r}} L(\mathbf{W}^{(t)})$, we have

$$\nabla_{\mathbf{w}_{j,r}} L(\mathbf{W}^{(t)})[1] = -\frac{1}{n}\Bigg[ \sum_{i=1}^{n} y_i \ell_{j,i}^{(t)} \sigma'(\langle \mathbf{w}_{j,r}^{(t)}, y_i \mathbf{v}\rangle) - \alpha \sum_{i=1}^{n} y_i \ell_{j,i}^{(t)} \cdot \sigma'(\langle \mathbf{w}_{j,r}^{(t)}, \boldsymbol{\xi}_i\rangle) \Bigg] + \lambda \mathbf{w}_{j,r}^{(t)}[1].$$

(B.4)

For any $k \in \mathcal{B}_i$, by Lemma B.1 we know that the gradient over this coordinate only depends on the training data $\boldsymbol{\xi}_i$, therefore, we have

$$\nabla_{\mathbf{w}_{j,r}} L(\mathbf{W}^{(t)})[k] = -\frac{1}{n} \ell_{j,i}^{(t)} \sigma'(\langle \mathbf{w}_{j,r}^{(t)}, \boldsymbol{\xi}_i\rangle) \boldsymbol{\xi}_i[k] + \lambda \mathbf{w}_{j,r}^{(t)}[k]. \tag{B.5}$$

For the remaining coordinates, we have

$$\nabla_{\mathbf{w}_{j,r}} L(\mathbf{W}^{(t)})[k] = \lambda \mathbf{w}_{j,r}^{(t)}[k]. \tag{B.6}$$

Now let us focus on the moving averaged gradient $\mathbf{m}_{j,r}^{(t)}$ and squared gradient $\mathbf{v}_{j,r}^{(t)}$. We first show that for all $k \in [d]$, it holds that

$$\frac{\big|\mathbf{m}_{j,r}^{(t)}[k]\big|}{\sqrt{\mathbf{v}_{j,r}^{(t)}[k]}} \leq \Theta(1). \tag{B.7}$$

By the update rule of $\mathbf{m}_{j,r}^{(t)}$, we have

$$\mathbf{m}_{j,r}^{(t)}[k] = \beta_1 \mathbf{m}_{j,r}^{(t-1)}[k] + (1 - \beta_1) \cdot \nabla_{\mathbf{w}_{j,r}} L(\mathbf{W}^{(t)})[k]$$

$$= \sum_{\tau=0}^{t} \beta_1^{\tau}(1 - \beta_1) \cdot \nabla_{\mathbf{w}_{j,r}} L(\mathbf{W}^{(t-\tau)})[k].$$

Similarly, we also have

$$\mathbf{v}_{j,r}^{(t)}[k] = \sum_{\tau=0}^{t} \beta_2^{\tau}(1 - \beta_2) \cdot \nabla_{\mathbf{w}_{j,r}} L(\mathbf{W}^{(t-\tau)})[k]^2.$$

Then by Cauchy-Schwartz inequality we have

$$\big(\mathbf{m}_{j,r}^{(t)}[k]\big)^2 \leq \Bigg( \sum_{\tau=0}^{t} \frac{[\beta_1^{\tau}(1 - \beta_1)]^2}{\alpha_{\tau}^2} \cdot \nabla_{\mathbf{w}_{j,r}} L(\mathbf{W}^{(t-\tau)})[k]^2 \Bigg) \cdot \Bigg( \sum_{\tau=0}^{t} \alpha_{\tau}^2 \Bigg).$$

Let $\alpha_{\tau}^2 = \frac{[\beta_1^{\tau}(1-\beta_1)]^2}{\beta_2^{\tau}(1-\beta_2)}$, which forms an exponentially decaying sequence if $\beta_2 \geq \beta_1^2$. Therefore, we have $\sum_{\tau=0}^{t} \alpha_{\tau}^2 = \Theta(1)$ and the above inequality implies that

$$\big(\mathbf{m}_{j,r}^{(t)}[k]\big)^2 \leq \mathbf{v}_{j,r}^{(t)}[k] \cdot \Theta(1),$$

which proves (B.7).

Now we are going to prove the main argument of this lemma. Note that $\mathbf{m}_{j,r}^{(t)}$, which is a weighted average of all historical gradients, where the weights decay exponentially fast, then we can take on a threshold $\bar{\tau} = \text{polylog}(\eta^{-1})$ such that $\sum_{\tau=\bar{\tau}}^{t} \beta_1^{\tau}(1-\beta_1) = \frac{1}{\text{poly}(\eta^{-1})}$. Then for each $k \in [d]$ we have

$$\mathbf{m}_{j,r}^{(t)}[k] = \sum_{\tau=0}^{\bar{\tau}} \beta_1^{\tau}(1-\beta_1) \cdot \nabla_{\mathbf{w}_{j,r}} L(\mathbf{W}^{(t-\tau)})[k] + \sum_{\tau=\bar{\tau}}^{t} \beta_1^{\tau}(1-\beta_1) \cdot \nabla_{\mathbf{w}_{j,r}} L(\mathbf{W}^{(t-\tau)})[k]$$

$$= \sum_{\tau=0}^{\bar{\tau}} \beta_1^{\tau}(1-\beta_1) \cdot \nabla_{\mathbf{w}_{j,r}} L(\mathbf{W}^{(t-\tau)})[k] \pm \frac{1}{\text{poly}(\eta^{-1})},$$

where in the last inequality we use the fact that $|\nabla_{\mathbf{w}_{j,r}} L(\mathbf{W}^{(t-\tau)})[k]| = \widetilde{O}(1)$ for all $k \in [d]$. Similarly, we can also have the following on $\mathbf{v}_{j,r}^{(t)}$,

$$\mathbf{v}_{j,r}^{(t)}[k] = \sum_{\tau=0}^{\bar{\tau}} \beta_2^{\tau}(1-\beta_2) \cdot \nabla_{\mathbf{w}_{j,r}} L(\mathbf{W}^{(t-\tau)})[k]^2 \pm \frac{1}{\text{poly}(\eta^{-1})}.$$

Here we slightly abuse the notation by using the same $\bar{\tau}$. Then we have

$$\frac{\mathbf{m}_{j,r}^{(t)}[k]}{\sqrt{\mathbf{v}_{j,r}^{(t)}[k]}} = \frac{\sum_{\tau=0}^{\bar{\tau}} \beta_1^{\tau}(1-\beta_1) \cdot \nabla_{\mathbf{w}_{j,r}} L(\mathbf{W}^{(t-\tau)})[k] \pm \frac{1}{\text{poly}(\eta^{-1})}}{\sqrt{\sum_{\tau=\bar{\tau}}^{\bar{\tau}} \beta_2^{\tau}(1-\beta_2) \cdot \nabla_{\mathbf{w}_{j,r}} L(\mathbf{W}^{(t-\tau)})[k]^2 \pm \frac{1}{\text{poly}(\eta^{-1})}}}.$$

In order to prove the main argument of this lemma, the key is to show that within $\bar{\tau}$ iterations, the gradient $\nabla_{\mathbf{w}_{j,r}} L(\mathbf{W}^{(t)})[k]$ barely changes. In particular, by (B.7), we have the update of each coordinate in one step is at most $\Theta(\eta)$. This implies that

$$\left| \langle \mathbf{w}_{j,r}^{(t)}, \mathbf{v} \rangle - \langle \mathbf{w}_{j,r}^{(\tau)}, \mathbf{v} \rangle \right| \leq \Theta(\eta\bar{\tau}),$$

$$\left| \langle \mathbf{w}_{j,r}^{(t)}, \boldsymbol{\xi}_i \rangle - \langle \mathbf{w}_{j,r}^{(\tau)}, \boldsymbol{\xi}_i \rangle \right| \leq \Theta(\eta\bar{\tau} s\sigma_p),$$

$$|\mathbf{w}_{j,r}^{(t)}[k] - \mathbf{w}_{j,r}^{(\tau)}[k]| \leq \Theta(\eta\bar{\tau}).$$

Then applying the fact that $|\langle \mathbf{w}_{j,r}^{(\tau)}, \mathbf{v} \rangle| \leq \widetilde{\Theta}(1)$ and $|\langle \mathbf{w}_{j,r}^{(\tau)}, \boldsymbol{\xi}_i \rangle| \leq \widetilde{\Theta}(1)$, we further have

$$\left| F_j(\mathbf{W}^{(\tau)}, \mathbf{x}_i) - F_j(\mathbf{W}^{(t)}, \mathbf{x}_i) \right| \leq \Theta(m\eta\bar{\tau} s\sigma_p) = \widetilde{\Theta}(\eta\bar{\tau} s\sigma_p),$$

where we use the fact that $m = \widetilde{\Theta}(1)$ and $s\sigma_p = \omega(1)$. Then it holds that

$$\ell_{j,i}^{(\tau)} = \frac{e^{F_j(\mathbf{W}^{(\tau)}, \mathbf{x}_i)}}{\sum_{k \in \{-1,1\}} e^{F_k(\mathbf{W}^{(\tau)}, \mathbf{x}_i)}}$$

$$\leq \frac{e^{F_j(\mathbf{W}^{(t)}, \mathbf{x}_i) + \widetilde{\Theta}(\eta\bar{\tau} s\sigma_p)}}{e^{F_j(\mathbf{W}^{(\tau)}, \mathbf{x}_i) + \widetilde{\Theta}(\eta\bar{\tau} s\sigma_p)} + e^{F_{-j}(\mathbf{W}^{(t)}, \mathbf{x}_i) - \widetilde{\Theta}(\eta\bar{\tau} s\sigma_p)}}$$

$$= \text{sgn}(\ell_{j,i}^{(t)}) \cdot \Theta(|\ell_{j,i}^{(t)}|),$$

where we use the fact that $\widetilde{\Theta}(\eta\bar{\tau} s\sigma_p) = o(1)$. Similarly, we can also show that $\ell_{j,i}^{(\tau)} \geq \text{sgn}(\ell_{j,i}^{(t)}) \cdot \Theta(|\ell_{j,i}^{(t)}|)$, which further implies

$$\ell_{j,i}^{(\tau)} = \text{sgn}(\ell_{j,i}^{(t)}) \cdot \Theta(|\ell_{j,i}^{(t)}|)$$

for all $\tau \in [t - \bar{\tau}, t]$. Note that $|\ell_{j,i}^{(\tau)}| \leq 1$, then it holds that

$$\ell_{j,i}^{(\tau)} \sigma'(\langle \mathbf{w}_{j,r}^{(\tau)}, \mathbf{v} \rangle) = \text{sgn}(\ell_{j,i}^{(t)}) \cdot \Theta(|\ell_{j,i}^{(t)}|) \cdot \sigma'(\langle \mathbf{w}_{j,r}^{(\tau)}, \mathbf{v} \rangle)$$

$$\leq \text{sgn}(\ell_{j,i}^{(t)}) \cdot \Theta(|\ell_{j,i}^{(t)}|) \cdot \sigma'(\langle \mathbf{w}_{j,r}^{(t)}, \mathbf{v} \rangle) + \Theta(|\ell_{j,i}^{(t)}|) \cdot \widetilde{\Theta}(\eta\bar{\tau}).$$

We can also similarly derive the following

$$\ell_{j,i}^{(\tau)} \sigma'(\langle \mathbf{w}_{j,r}^{(\tau)}, \mathbf{v} \rangle) \geq \mathrm{sgn}(\ell_{j,i}^{(t)}) \cdot \Theta(|\ell_{j,i}^{(t)}|) \cdot \sigma'(\langle \mathbf{w}_{j,r}^{(t)}, \mathbf{v} \rangle) - \Theta(|\ell_{j,i}^{(t)}|) \cdot \widetilde{\Theta}(\eta \bar{\tau}),$$

$$\ell_{j,i}^{(\tau)} \sigma'(\langle \mathbf{w}_{j,r}^{(\tau)}, \boldsymbol{\xi}_i \rangle) \leq \mathrm{sgn}(\ell_{j,i}^{(t)}) \cdot \Theta(|\ell_{j,i}^{(t)}|) \cdot \sigma'(\langle \mathbf{w}_{j,r}^{(t)}, \boldsymbol{\xi}_i \rangle) + \Theta(|\ell_{j,i}^{(t)}|) \cdot \widetilde{\Theta}(\eta \bar{\tau} s \sigma_p),$$

$$\ell_{j,i}^{(\tau)} \sigma'(\langle \mathbf{w}_{j,r}^{(\tau)}, \boldsymbol{\xi}_i \rangle) \geq \mathrm{sgn}(\ell_{j,i}^{(t)}) \cdot \Theta(|\ell_{j,i}^{(t)}|) \cdot \sigma'(\langle \mathbf{w}_{j,r}^{(t)}, \boldsymbol{\xi}_i \rangle) - \Theta(|\ell_{j,i}^{(t)}|) \cdot \widetilde{\Theta}(\eta \bar{\tau} s \sigma_p).$$

Combining the above results, applying (B.4), (B.5), and (B.6), we can show that for the first coordinate, we have

$$\nabla_{\mathbf{w}_{j,r}} L(\mathbf{W}^{(\tau)})[1] = \Theta\left(\nabla_{\mathbf{w}_{j,r}} L(\mathbf{W}^{(t)})[1]\right) \pm \Theta\left(\frac{1}{n} \sum_{i=1}^{n} |\ell_{j,i}^{(t)}|\right) \cdot \widetilde{O}(\eta \bar{\tau}) \pm \Theta(\lambda \eta \bar{\tau});$$

for any $k \in \mathcal{B}_j$, we have

$$\nabla_{\mathbf{w}_{j,r}} L(\mathbf{W}^{(\tau)})[k] = \Theta\left(\nabla_{\mathbf{w}_{j,r}} L(\mathbf{W}^{(t)})[k]\right) \pm \Theta\left(\frac{|\ell_{j,i}^{(t)}|}{n}\right) \cdot \widetilde{O}(\eta \bar{\tau} s \sigma_p) \pm \Theta(\lambda \eta \bar{\tau});$$

and for remaining coordinates, we have

$$\nabla_{\mathbf{w}_{j,r}} L(\mathbf{W}^{(\tau)})[k] = \Theta\left(\nabla_{\mathbf{w}_{j,r}} L(\mathbf{W}^{(t)})[k]\right) \pm \Theta(\lambda \eta \widetilde{\tau}).$$

Now we can plug the above results into the formula of $\mathbf{m}_{j,r}^{(t)}$ and $\mathbf{v}_{j,r}^{(t)}$. Using the fact that $\bar{\tau} = \widetilde{\Theta}(1)$, $\lambda = o(1)$, and $|\ell_{j,i}^{(t)}| \leq 1$, we have for all $k = 1$ or $k \notin \mathcal{B}_i$ for any $i$,

$$\frac{\mathbf{m}_{j,r}^{(t)}[k]}{\sqrt{\mathbf{v}_{j,r}^{(t)}[k]}} = \frac{\nabla_{\mathbf{w}_{j,r}} L(\mathbf{W}^{(t)})[k] \pm \widetilde{\Theta}(\eta)}{\Theta\left(|\nabla_{\mathbf{w}_{j,r}} L(\mathbf{W}^{(t)})[k]|\right) \pm \widetilde{\Theta}(\eta)}.$$

For $k \in \mathcal{B}_i$ we have

$$\frac{\mathbf{m}_{j,r}^{(t)}[k]}{\sqrt{\mathbf{v}_{j,r}^{(t)}[k]}} = \frac{\nabla_{\mathbf{w}_{j,r}} L(\mathbf{W}^{(t)})[k] \pm \widetilde{\Theta}\left(\frac{\eta s \sigma_p |\ell_{j,i}^{(t)}|}{n}\right) \pm \widetilde{\Theta}(\lambda \eta)}{\Theta\left(|\nabla_{\mathbf{w}_{j,r}} L(\mathbf{W}^{(t)})[k]|\right) \pm \widetilde{\Theta}\left(\frac{\eta s \sigma_p |\ell_{j,i}^{(t)}|}{n}\right) \pm \widetilde{\Theta}(\lambda \eta)}.$$

Then, we can conclude that for all $k = 1$ or $k \notin \mathcal{B}_i$ for any $i$, we have either $|\nabla_{\mathbf{w}_{j,r}} L(\mathbf{W}^{(t)})[k]| \leq \widetilde{\Theta}(\eta)$ or

$$\frac{\mathbf{m}_{j,r}^{(t)}[k]}{\sqrt{\mathbf{v}_{j,r}^{(t)}[k]}} = \mathrm{sgn}\left(\nabla_{\mathbf{w}_{j,r}} L(\mathbf{W}^{(t)})[k]\right) \cdot \Theta(1).$$

For any $k \in \mathcal{B}_i$, we have either $|\nabla_{\mathbf{w}_{j,r}} L(\mathbf{W}^{(t)})[k]| \leq \widetilde{\Theta}\left(\eta n^{-1} s \sigma_p |\ell_{j,i}^{(t)}| + \lambda \eta\right)$ or

$$\frac{\mathbf{m}_{j,r}^{(t)}[k]}{\sqrt{\mathbf{v}_{j,r}^{(t)}[k]}} = \mathrm{sgn}\left(\nabla_{\mathbf{w}_{j,r}} L(\mathbf{W}^{(t)})[k]\right) \cdot \Theta(1).$$

This completes the proof.

$\square$

**Lemma B.3** (Lemma 5.2, restated). Suppose the training data is generated according to Definition 3.1, assume $\lambda = o(\sigma_0^{q-2} \sigma_p / n)$ and $\eta = 1/\mathrm{poly}(d)$, then for any $t \leq T_0$ with $T_0 = \widetilde{O}\left(\frac{1}{\eta s \sigma_p}\right)$ and any $i \in [n]$,

$$\langle \mathbf{w}_{j,r}^{(t+1)}, j \cdot \mathbf{v} \rangle \leq \langle \mathbf{w}_{j,r}^{(t)}, j \cdot \mathbf{v} \rangle + \Theta(\eta),$$

$$\langle \mathbf{w}_{y_i,r}^{(t+1)}, \boldsymbol{\xi}_i \rangle = \langle \mathbf{w}_{y_i,r}^{(t)}, \boldsymbol{\xi}_i \rangle + \widetilde{\Theta}(\eta s \sigma_p).$$

*Proof.* At the initialization, we have

$$|\langle \mathbf{w}_{j,r}^{(0)}, \mathbf{v}\rangle| = \widetilde{\Theta}(\sigma_0), \quad |\langle \mathbf{w}_{j,r}^{(0)}, \boldsymbol{\xi}_i\rangle| = \widetilde{\Theta}(s^{1/2}\sigma_p\sigma_0 + \alpha) = \widetilde{\Theta}(s^{1/2}\sigma_p\sigma_0), \quad \mathbf{w}_{j,r}^{(0)}[k] = \widetilde{\Theta}(\sigma_0),$$

which also imply that $|\ell_{j,i}^{(0)}| = \Theta(1)$. Besides, note that $\ell_{j,i}^{(t)} = \mathbb{1}_{j=y_i} - \text{logit}_j(F^{(t)}, \mathbf{x}_i)$, we have

$$\text{sgn}(y_i \ell_{j,i}^{(t)}) = \text{sgn}(j),$$

where we recall that $j \in \{-1, 1\}$. Therefore, given that $\lambda = o(\sigma_0^{q-1})$, $\alpha = o(1)$, $s^{1/2}\sigma_p = \widetilde{O}(1)$, and assume $\ell_{j,i}^{(t)} = \Theta(1)$ (which will be verified later),

$$\text{sgn}\left( \sum_{i=1}^n y_i \ell_{j,i}^{(t)} \sigma'(\langle \mathbf{w}_{j,r}^{(t)}, y_i \mathbf{v}\rangle) - \alpha \sum_{i=1}^n y_i \ell_{j,i}^{(t)} \sigma'(\langle \mathbf{w}_{j,r}^{(t)}, \boldsymbol{\xi}_i\rangle) - n\lambda \mathbf{w}_{j,r}^{(t)}[1] \right)$$
$$= \text{sgn}\left[ j \cdot \widetilde{\Theta}(n\sigma_0^{q-1}) - j \cdot \widetilde{\Theta}(\alpha n(s^{1/2}\sigma_p\sigma_0)^{q-1}) \pm o(\sigma_0^{q-1}\sigma_p) \right]$$
$$= \text{sgn}(j).$$

Since $\mathbf{v}$ is 1-sparse, then by Lemma B.2, the following inequality naturally holds,

$$\langle \mathbf{w}_{j,r}^{(t+1)}, j \cdot \mathbf{v}\rangle \leq \langle \mathbf{w}_{j,r}^{(t)}, j \cdot \mathbf{v}\rangle - \eta\left\langle \mathbf{m}_{j,r}^{(t)}/\sqrt{\mathbf{v}_{j,r}^{(t)}}, j \cdot \mathbf{v}\right\rangle \leq \langle \mathbf{w}_{j,r}^{(t)}, j \cdot \mathbf{v}\rangle + \Theta(\eta).$$

Additionally, in terms of the memorization of noise, we first consider the iterate in the initialization. By the condition that $\eta = o(1/d) = o(1/(s\sigma_p))$ and note that for a sufficiently large fraction of $k \in \mathcal{B}_i$ (e.g., 0.99), we have $|\boldsymbol{\xi}_i[k]| \geq \widetilde{\Theta}(\sigma_p) \geq \widetilde{\Theta}(\eta n^{-1} s\sigma_p |\ell_{j,i}^{(0)}|)$ and thus

$$\text{sgn}(\nabla_{\mathbf{w}_{y_i,r}} L(\mathbf{W}^{(0)})[k]) = \text{sgn}\left( \ell_{y_i,i}^{(t)} \sigma'(\langle \mathbf{w}_{y_i,r}^{(t)}, \boldsymbol{\xi}_i\rangle) \boldsymbol{\xi}_i[k] - n\lambda \mathbf{w}_{y_i,r}^{(0)}[k] \right)$$
$$= -\text{sgn}\left[ \widetilde{\Theta}\left( (d^{1/2}\sigma_p\sigma_0)^{q-1}\sigma_p \cdot \text{sgn}(\boldsymbol{\xi}_i[k]) \right) \pm o(\sigma_0^{q-1}\sigma_p) \right] = -\text{sgn}(\boldsymbol{\xi}_i[k]). \tag{B.8}$$

Therefore, by Lemma B.2 we have the following according to (B.3),

$$\langle \mathbf{w}_{y_i,r}^{(1)}, \boldsymbol{\xi}_i\rangle = \langle \mathbf{w}_{y_i,r}^{(0)}, \boldsymbol{\xi}_i\rangle - \eta\left\langle \mathbf{m}_{j,r}^{(t)}/\sqrt{\mathbf{v}_{y_i,r}^{(t)}}, \boldsymbol{\xi}_i\right\rangle$$
$$\geq \langle \mathbf{w}_{y_i,r}^{(0)}, \boldsymbol{\xi}_i\rangle + \Theta(\eta) \cdot \sum_{k \in \mathcal{B}_i} \langle \text{sgn}(\boldsymbol{\xi}_i[k]), \boldsymbol{\xi}_i[k]\rangle - O(\eta s\sigma_p) - O(\eta\alpha)$$
$$= \langle \mathbf{w}_{y_i,r}^{(0)}, \boldsymbol{\xi}_i\rangle + \widetilde{\Theta}(\eta s\sigma_p),$$

where in the first inequality the term $O(\eta s\sigma_p)$ represents the coordinates that $|\boldsymbol{\xi}_i[k]| \leq O(\sigma_p)$ (so that we cannot use the sign information of $\nabla_{y_i,r} L(\mathbf{W}^{(0)})$ but directly bound it by $\Theta(1)$) and the last inequality is due to the fact that $|\mathcal{B}_i| \geq s - 1$ and $\alpha = o(1)$. For general $t$, we will consider the following induction hypothesis:

$$\langle \mathbf{w}_{y_i,r}^{(t+1)}, \boldsymbol{\xi}_i\rangle = \langle \mathbf{w}_{y_i,r}^{(t)}, \boldsymbol{\xi}_i\rangle + \widetilde{\Theta}(\eta s\sigma_p), \tag{B.9}$$

which has already been verified for $t = 0$. By Hypothesis (B.9), the following holds at time $t$,

$$\langle \mathbf{w}_{y_i,r}^{(t)}, \boldsymbol{\xi}_i\rangle = \langle \mathbf{w}_{y_i,r}^{(0)}, \boldsymbol{\xi}_i\rangle + \widetilde{\Theta}(t\eta s\sigma_p) = \widetilde{\Theta}(s^{1/2}\sigma_p\sigma_0 + t\eta s\sigma_p).$$

In the meanwhile, we have the following upper bound for $|\mathbf{w}_{j,r}^{(t)}[k]|$,

$$|\mathbf{w}_{j,r}^{(t)}[k]| \leq |\mathbf{w}_{j,r}^{(t)}[k]| + \eta|\text{sign}(\nabla_{\mathbf{w}_{j,r}} L(\mathbf{W}^{(t)}))| \leq |\mathbf{w}_{j,r}^{(0)}[k]| + t\eta = \widetilde{\Theta}(\sigma_0 + t\eta). \tag{B.10}$$

Besides, it is also easy to verify that for any $t \leq T_0 = \widetilde{\Theta}\left(\frac{1}{s\sigma_p\eta m}\right) = \widetilde{\Theta}\left(\frac{1}{s\sigma_p\eta}\right)$, we have $\langle \mathbf{w}_{y_i,r}^{(t)}, \boldsymbol{\xi}_i\rangle, \langle \mathbf{w}_{y_i,r}^{(t)}, j \cdot \mathbf{v}\rangle < \Theta(1/m)$ and thus $|\ell_{j,i}^{(t)}| = \Theta(1)$. Then similar to (B.8), we have

$$\text{sgn}(\nabla_{\mathbf{w}_{y_i,r}} L(\mathbf{W}^{(t)})[k])$$
$$= \text{sgn}\left( \ell_{y_i,i}^{(t)} \sigma'(\langle \mathbf{w}_{y_i,r}^{(t)}, \boldsymbol{\xi}_i\rangle) \boldsymbol{\xi}_i[k] - n\lambda \mathbf{w}_{y_i,r}^{(0)}[k] \right)$$

$$
\begin{aligned}
&= -\operatorname{sgn}\big(\widetilde{\Theta}\big[(s^{1/2}\sigma_p\sigma_0 + t\eta s\sigma_p)^{q-1}\sigma_p \cdot \operatorname{sgn}(\boldsymbol{\xi}_i[k])\big) \pm o\big(\sigma_0^{q-2}\sigma_p \cdot (\sigma_0 + t\eta)\big)\big] \\
&= -\operatorname{sgn}(\boldsymbol{\xi}_i[k]). \tag{B.11}
\end{aligned}
$$

This further implies that

$$
\begin{aligned}
\langle \mathbf{w}_{y_i,r}^{(t+1)}, \boldsymbol{\xi}_i \rangle &\geq \langle \mathbf{w}_{y_i,r}^{(t)}, \boldsymbol{\xi}_i \rangle - \Theta(\eta) \cdot \sum_{k \in \mathcal{B}_i} \langle \operatorname{sgn}\big(\nabla_{\mathbf{w}_{y_i,r}} L(\mathbf{W}^{(t)})[k]\big), \boldsymbol{\xi}_i[k] \rangle - O(\eta^2 s^2 \sigma_p^2) - O(\eta\alpha) \\
&= \langle \mathbf{w}_{y_i,r}^{(t)}, \boldsymbol{\xi}_i \rangle + \widetilde{\Theta}(\eta s \sigma_p),
\end{aligned}
$$

where the term $-O(\eta^2 s^2 \sigma_p^2)$ is contributed by the gradient coordinates that are smaller than $\Theta(\eta s \sigma_p)$. This verifies Hypothesis (B.9) at time $t$ and thus completes the proof. $\qquad\square$

From Lemma B.3, note that $s\sigma_p = \omega(1)$, then it can be seen that $\langle \mathbf{w}_{j,r}^{(t)}, j \cdot \mathbf{v} \rangle$ increases much faster than $\langle \mathbf{w}_{j,r}^{(t)}, j \cdot \mathbf{v} \rangle$. By looking at the update rule of $\langle \mathbf{w}_{j,r}^{(t)}, j \cdot \mathbf{v} \rangle$ (see (B.2)), it will keeps increasing only when, roughly speaking, $\sigma'(\langle \mathbf{w}_{j,r}^{(t)}, j \cdot \mathbf{v} \rangle) > \alpha\sigma'(\langle \mathbf{w}_{j,r}^{(t)}, \boldsymbol{\xi}_i \rangle)$. Since $\langle \mathbf{w}_{j,r}^{(t)}, \boldsymbol{\xi}_i \rangle$ increases much faster than $\langle \mathbf{w}_{j,r}^{(t)}, j \cdot \mathbf{v} \rangle$, it can be anticipated after a certain number of iterations, $\langle \mathbf{w}_{j,r}^{(t)}, j \cdot \mathbf{v} \rangle$ will start to decrease. In the following lemma, we provide an upper bound on the iteration number such that this decreasing occurs.

**Lemma B.4** (Lemma B.4, restated). *Suppose the training data is generated according to Definition 3.1, $\alpha \geq \widetilde{\Theta}\big((s\sigma_p)^{1-q} \vee \sigma_0^{q-1}\big)$ and $\sigma_0 < \widetilde{O}((s\sigma_p)^{-1})$, then for any $t \in [T_r, T_0]$ with $T_r = \widetilde{O}\big(\frac{\sigma_0}{\eta s \sigma_p \alpha^{1/(q-1)}}\big) \leq T_0$,*

$$
\langle \mathbf{w}_{j,r}^{(t+1)}, j \cdot \mathbf{v} \rangle = \langle \mathbf{w}_{j,r}^{(t)}, j \cdot \mathbf{v} \rangle - \Theta(\eta).
$$

*Moreover, it holds that*

$$
\mathbf{w}_{j,r}^{(T_0)}[k] = \begin{cases} -\operatorname{sgn}(j) \cdot \widetilde{\Omega}\big(\frac{1}{s\sigma_p}\big), & k = 1, \\ \operatorname{sgn}(\boldsymbol{\xi}_i[k]) \cdot \widetilde{\Omega}\big(\frac{1}{s\sigma_p}\big) \text{ or } \pm\widetilde{O}(\eta), & k \in \mathcal{B}_i, \text{ with } y_i = j, \\ \pm\widetilde{O}(\eta), & \text{otherwise.} \end{cases}
$$

*Proof.* Recall from Lemma B.3 that for any $t \leq T_0$ we have

$$
\begin{aligned}
\langle \mathbf{w}_{j,r}^{(t+1)}, j \cdot \mathbf{v} \rangle &\leq \langle \mathbf{w}_{j,r}^{(t)}, j \cdot \mathbf{v} \rangle + \Theta(\eta) \leq \langle \mathbf{w}_{j,r}^{(0)}, j \cdot \mathbf{v} \rangle + \Theta(t\eta), \\
\langle \mathbf{w}_{y_s,r}^{(t+1)}, \boldsymbol{\xi}_s \rangle &= \langle \mathbf{w}_{y_s,r}^{(t)}, \boldsymbol{\xi}_s \rangle + \widetilde{\Theta}(\eta s \sigma_p) \leq \langle \mathbf{w}_{y_s,r}^{(0)}, \boldsymbol{\xi}_s \rangle + \widetilde{\Theta}(t\eta s \sigma_p).
\end{aligned}
$$

Besides, by Lemma B.2 we also have $|\mathbf{w}_{j,r}^{(t)}[k]| \leq |\mathbf{w}_{j,r}^{(0)}[k]| + O(t\eta)$. Then it can be verified that for some $T_r = \widetilde{O}\big(\frac{\sigma_0}{\eta s \sigma_p \alpha^{1/(q-1)}}\big)$, we have for all $i \in [n]$ and $t \in [T_r, T_0]$

$$
\alpha\sigma'(\langle \mathbf{w}_{y_i,r}^{(t)}, \boldsymbol{\xi}_i \rangle) \geq C \cdot \big[\sigma'(\langle \mathbf{w}_{j,r}^{(t)}, j \cdot \mathbf{v} \rangle) + \lambda n |\mathbf{w}_{j,r}^{(t)}[1]|\big]
$$

for some constant $C$. This further implies that

$$
\begin{aligned}
&\operatorname{sgn}\big(\nabla_{\mathbf{w}_{j,r}} L(\mathbf{W}^{(t)})[1]\big) \\
&= -\operatorname{sgn}\bigg(\sum_{i=1}^n y_i \ell_{j,i}^{(t)} \sigma'(\langle \mathbf{w}_{j,r}^{(t)}, y_i \mathbf{v} \rangle) - \alpha \sum_{i=1}^n y_i \ell_{j,i}^{(t)} \sigma'(\langle \mathbf{w}_{j,r}^{(t)}, \boldsymbol{\xi}_i \rangle) - n\lambda \mathbf{w}_{j,r}^{(t)}[1]\bigg) \\
&= -\operatorname{sgn}\big[-\alpha \sum_{i=1}^n y_i \ell_{j,i}^{(t)} \sigma'(\langle \mathbf{w}_{j,r}^{(t)}, \boldsymbol{\xi}_i \rangle)\big] \\
&= \operatorname{sgn}(j),
\end{aligned}
$$

where we use the fact that $\operatorname{sgn}(y_i \ell_{j,i}^{(t)}) = \operatorname{sgn}(j)$ for all $i \in [n]$. Then by Lemma B.2 and (B.2), we have for all $t \in [T_r, T_0]$,

$$
\langle \mathbf{w}_{j,r}^{(t+1)}, j \cdot \mathbf{v} \rangle = \langle \mathbf{w}_{j,r}^{(t)}, j \cdot \mathbf{v} \rangle - \Theta(\eta) \cdot \operatorname{sgn}(j) \cdot \operatorname{sgn}\big(\nabla_{\mathbf{w}_{j,r}} L(\mathbf{W}^{(t)})[1]\big) = \langle \mathbf{w}_{j,r}^{(t)}, j \cdot \mathbf{v} \rangle - \Theta(\eta).
$$

Then at iteration $T_0$, for the first coordinate we have

$$\mathbf{w}_{j,r}^{(T_0)}[1] = \mathbf{w}_{j,r}^{(0)}[1] + \text{sgn}(j) \cdot \Theta(T_r\eta) - \text{sgn}(j) \cdot \Theta((T_0 - T_r)\eta) \geq -\text{sgn}(j) \cdot \widetilde{\Omega}\left(\frac{1}{s\sigma_p}\right)$$

For any $k \in \mathcal{B}_i$ with $y_i = j$, we have either the coordinate will increase at a rate of $\Theta(1)$ or fall into $0$. As a consequence we have either $\mathbf{w}_{j,r}^{(T_0)}[k] \in [-\widetilde{\Theta}(\eta), \widetilde{\Theta}(\eta)]$ or

$$\mathbf{w}_{j,r}^{(T_0)}[k] = \mathbf{w}_{j,r}^{(0)}[k] + \text{sgn}(\boldsymbol{\xi}_i[k]) \cdot \Theta(T_0\eta) \geq \text{sgn}(\boldsymbol{\xi}_i[k]) \cdot \widetilde{\Omega}\left(\frac{1}{s\sigma_p}\right).$$

For the remaining coordinate, its update will be determined by the regularization term, which will finally fall into the region around zero since we have $T_0\eta = \omega(\sigma_0)$. By Lemma B.2 it is clear that $\mathbf{w}_{j,r}^{(T_0)}[k] \in [-\widetilde{\Theta}(\eta), \widetilde{\Theta}(\eta)]$. □

**Lemma B.5** (Lemma 5.4, restated)**.** If $\alpha = O\left(\frac{s\sigma_p^2}{n}\right)$ and $\eta = o(\lambda)$, then let $r^* = \arg\max_{r\in[m]}\langle\mathbf{w}_{y_i,r}^{(t)}, \boldsymbol{\xi}_i\rangle$, for any $t \geq T_0$, $i \in [n]$, $j \in [2]$ and $r \in [m]$, it holds that

$$\langle\mathbf{w}_{y_i,r^*}^{(t)}, \boldsymbol{\xi}_i\rangle = \widetilde{\Theta}(1), \quad \sum_{k\in\mathcal{B}_i} |\mathbf{w}_{y_i,r^*}^{(t)}[k]| \cdot |\boldsymbol{\xi}_i[k]| = \widetilde{\Theta}(1),$$

$$\forall r \in [m], \quad \langle\mathbf{w}_{j,r}^{(t)}, \text{sgn}(j) \cdot \mathbf{v}\rangle \in [-\widetilde{O}\left(\frac{n\alpha}{s\sigma_p^2}\right), O(\lambda^{-1}\eta)].$$

*Proof.* The proof will be relying on the following three induction hypothesis:

$$\langle\mathbf{w}_{y_i,r^*}^{(t)}, \boldsymbol{\xi}_i\rangle = \widetilde{\Omega}(1), \tag{B.12}$$

$$\sum_{k\in\mathcal{B}_i} |\mathbf{w}_{y_i,r^*}^{(t+1)}[k]| \cdot |\boldsymbol{\xi}_i[k]| = \widetilde{\Theta}(1), \tag{B.13}$$

$$\forall r \in [m], \ \langle\mathbf{w}_{j,r}^{(t)}, \text{sgn}(j) \cdot \mathbf{v}\rangle \in \left[-\widetilde{O}\left(\frac{n\alpha}{s\sigma_p^2}\right), O(\lambda^{-1}\eta)\right], \tag{B.14}$$

which we assume they hold for all $\tau \leq t$ and $r \in [m]$, $i \in [n]$, and $j \in [2]$. It is clear that all hypothesis hold when $t = T_0$ according to Lemma B.4.

**Verifying Hypothesis** (B.12)**.** We first verify Hypothesis (B.12). Recall that the update rule for $\langle\mathbf{w}_{y_i,r}^{(t)}, \boldsymbol{\xi}_i\rangle$ is given as follows,

$$\langle\mathbf{w}_{y_i,r}^{(t+1)}, \boldsymbol{\xi}_i\rangle$$

$$= \langle\mathbf{w}_{y_i,r}^{(t)}, \boldsymbol{\xi}_i\rangle - \eta \cdot \langle\mathbf{m}_{y_i,r}^{(t)}/\sqrt{\mathbf{v}_{y_i,r}^{(t)}}, \boldsymbol{\xi}_i\rangle$$

$$\geq \langle\mathbf{w}_{y_i,r}^{(t)}, \boldsymbol{\xi}_i\rangle - \Theta(\eta) \cdot \langle\text{sgn}(\nabla_{\mathbf{w}_{y_i,r}} L(\mathbf{W}^{(t)})), \boldsymbol{\xi}_i\rangle - \widetilde{\Theta}(\eta^2 s^2 \sigma_p^2)$$

$$= \langle\mathbf{w}_{y_i,r}^{(t)}, \boldsymbol{\xi}_i\rangle + \Theta(\eta) \cdot \sum_{k\in\mathcal{B}_i} \left\langle\text{sgn}\left(\ell_{y_i,i}^{(t)}\sigma'(\langle\mathbf{w}_{y_i,r}^{(t)}, \boldsymbol{\xi}_i\rangle)\boldsymbol{\xi}_i[k] - n\lambda\mathbf{w}_{y_i,r}^{(t)}[k]\right), \boldsymbol{\xi}_i[k]\right\rangle$$

$$- \alpha y_i\Theta(\eta) \cdot \text{sgn}\left(\sum_{i=1}^n y_i\ell_{j,i}^{(t)}\sigma'(\langle\mathbf{w}_{j,r}^{(t)}, y_i\mathbf{v}\rangle) - \alpha\sum_{i=1}^n y_i\ell_{j,i}^{(t)}\sigma'(\langle\mathbf{w}_{j,r}^{(t)}, \boldsymbol{\xi}_i\rangle) - n\lambda\mathbf{w}_{j,r}^{(t)}[1]\right)$$

$$- \widetilde{\Theta}(\eta^2 s^2 \sigma_p^2). \tag{B.15}$$

Note that for any $a$ and $b$ we have $\text{sgn}(a - b) \cdot a \geq |a| - 2|b|$. Then it follows that

$$\sum_{k\in\mathcal{B}_i} \left\langle\text{sgn}\left(\ell_{y_i,i}^{(t)}\sigma'(\langle\mathbf{w}_{y_i,r}^{(t)}, \boldsymbol{\xi}_i\rangle)\boldsymbol{\xi}_i[k] - n\lambda\mathbf{w}_{y_i,r}^{(t)}[k]\right), \boldsymbol{\xi}_i[k]\right\rangle \geq \sum_{k\in\mathcal{B}_i} \left(|\boldsymbol{\xi}_i[k]| - \frac{2n\lambda|\mathbf{w}_{y_i,r}^{(t)}[k]|}{\ell_{y_i,i}^{(t)}\sigma'(\langle\mathbf{w}_{y_i}^{(t)}, \boldsymbol{\xi}_i\rangle)}\right)$$

$$\geq \widetilde{\Theta}(s\sigma_p) - \widetilde{\Theta}\left(\frac{n\lambda}{\ell_{y_i,i}^{(t)}\sigma_p}\right),$$

where the last inequality follows from Hypothesis (B.12) and (B.13). Further recall that $\lambda = o(\sigma_0^{q-2}\sigma_p/n)$, plugging the above inequality to (B.15) gives

$$
\begin{aligned}
\langle \mathbf{w}_{y_i,r}^{(t+1)}, \boldsymbol{\xi}_i \rangle &\geq \langle \mathbf{w}_{y_i,r}^{(t)}, \boldsymbol{\xi}_i \rangle + \widetilde{\Theta}(\eta s \sigma_p) - \widetilde{\Theta}\left( \frac{\eta n \lambda}{\ell_{y_i,i}^{(t)} \sigma_p} \right) - \widetilde{\Theta}(\eta^2 s^2 \sigma_p^2) \\
&\geq \langle \mathbf{w}_{y_i,r}^{(t)}, \boldsymbol{\xi}_i \rangle + \widetilde{\Theta}(\eta s \sigma_p) - \Theta(\alpha \eta) - \widetilde{\Theta}\left( \frac{\eta \sigma_0^{q-2}}{\ell_{y_i,i}^{(t)}} \right).
\end{aligned}
\tag{B.16}
$$

Then it is clear that $\langle \mathbf{w}_{y_i,r}^{(t)}, \boldsymbol{\xi}_i \rangle$ will increase by $\widetilde{\Theta}(\eta s \sigma_p)$ if $\ell_{y_i,i}^{(t)}$ is larger than some constant of order $\widetilde{\Omega}(\frac{n\lambda}{s\sigma_p^2}) = \widetilde{\Omega}(\frac{\sigma_0^{q-2}}{s\sigma_p})$. We will first show that as soon as there is a iterate $\mathbf{W}^{(\tau)}$ satisfying $\ell_{y_i,i}^{(\tau)} \leq \widetilde{O}(\frac{n\lambda}{s\sigma_p^2})$ for some $\tau \leq t$, then it must hold that $\ell_{y_i,i}^{(\tau')}$ will also be smaller than some constant in the order of $\widetilde{O}(\frac{n\lambda}{s\sigma_p^2})$ for all $\tau' \in [\tau, t+1]$. To prove this, we first note that if $\ell_{y_i,i}^{(t)}$ reaches some constant in the order of $\widetilde{O}(\frac{n\lambda}{s\sigma_p^2})$, we have for all $r \in [m]$ by (B.16)

$$
\begin{aligned}
\langle \mathbf{w}_{y_i,r}^{(t+1)}, \boldsymbol{\xi}_i \rangle &\geq \langle \mathbf{w}_{y_i,r}^{(t)}, \boldsymbol{\xi}_i \rangle + \widetilde{\Theta}(\eta s \sigma_p), \\
\langle \mathbf{w}_{-y_i,r}^{(t+1)}, \boldsymbol{\xi}_i \rangle &\leq \langle \mathbf{w}_{-y_i,r}^{(t)}, \boldsymbol{\xi}_i \rangle + O(\alpha \eta), \\
|\langle \mathbf{w}_{j,r}^{(t+1)}, \mathbf{v} \rangle| &\leq |\langle \mathbf{w}_{j,r}^{(t)}, \mathbf{v} \rangle| + O(\eta).
\end{aligned}
\tag{B.17}
$$

Therefore, we have

$$
\begin{aligned}
\ell_{y_i,i}^{(t+1)} &= \frac{e^{F_{-y_i}(\mathbf{W}^{(t+1)}, \mathbf{x}_i)}}{\sum_{j \in \{-1,1\}} e^{F_j(\mathbf{W}^{(t+1)}, \mathbf{x}_i)}} \\
&= \frac{1}{1 + \exp\left[ \sum_{r=1}^m \left[ \sigma(\langle \mathbf{w}_{y_i,r}^{(t+1)}, \mathbf{v} \rangle) + \sigma(\langle \mathbf{w}_{y_i,r}^{(t+1)}, \boldsymbol{\xi}_i \rangle) - \sigma(\langle \mathbf{w}_{-y_i,r}^{(t+1)}, \mathbf{v} \rangle) - \sigma(\langle \mathbf{w}_{-y_i,r}^{(t+1)}, \boldsymbol{\xi}_i \rangle) \right] \right]} \\
&\leq \frac{1}{1 + \exp\left[ \sum_{r=1}^m \left[ \sigma(\langle \mathbf{w}_{y_i,r}^{(t)}, \mathbf{v} \rangle) + \sigma(\langle \mathbf{w}_{y_i,r}^{(t)}, \boldsymbol{\xi}_i \rangle) - \sigma(\langle \mathbf{w}_{-y_i,r}^{(t)}, \mathbf{v} \rangle) - \sigma(\langle \mathbf{w}_{-y_i,r}^{(t)}, \boldsymbol{\xi}_i \rangle) \right] + \widetilde{\Theta}(\eta s \sigma_p^2) \right]} \\
&\leq \frac{1}{1 + \exp\left[ \sum_{r=1}^m \left[ \sigma(\langle \mathbf{w}_{y_i,r}^{(t)}, \mathbf{v} \rangle) + \sigma(\langle \mathbf{w}_{y_i,r}^{(t)}, \boldsymbol{\xi}_i \rangle) \right] - \sigma(\langle \mathbf{w}_{-y_i,r}^{(t)}, \mathbf{v} \rangle) - \sigma(\langle \mathbf{w}_{-y_i,r}^{(t)}, \boldsymbol{\xi}_i \rangle) \right]\right]} \\
&= \ell_{y_i,i}^{(t)},
\end{aligned}
$$

where inequality follows from (B.17). Therefore, this implies that as long as $\ell_{y_i,i}^{(t)}$ is larger than some constant $b = \widetilde{O}(\frac{n\lambda}{s\sigma_p^2})$, then the adam algorithm will prevent it from further increasing. Besides, since $m\eta \sigma_p^2 = o(1)$, then we must have $\ell_{y_i,i}^{(t+1)} \in [0.5\ell_{y_i,i}^{(t)}, 2\ell_{y_i,i}^{(t)}]$. As a consequence, we can deduce that $\ell_{y_i,i}^{(t)}$ cannot be larger than $2b$, since otherwise there must exists a iterate $\mathbf{W}^{(\tau)}$ with $\tau \leq t$ such that $\ell_{y_i,i}^{(\tau)} \in [b, 2b]$ and $\ell_{y_i,i}^{(\tau+1)} \geq \ell_{y_i,i}^{(\tau)}$, which contradicts the fact that $\ell_{y_i,i}^{(\tau)}$ should decreases if $\ell_{y_i,i}^{(\tau)} \geq b$. Therefore, we can claim that if $\ell_{y_i,i}^{(\tau)} \leq b = \widetilde{O}(\frac{n\lambda}{s\sigma_p^2})$ for some $\tau \leq t$, then we have

$$
\ell_{y_i,i}^{(\tau')} \leq \widetilde{O}\left( \frac{n\lambda}{s\sigma_p^2} \right)
\tag{B.18}
$$

for all $\tau' \in [\tau, t+1]$. Then further note that

$$
\begin{aligned}
2\ell_{y_i,i}^{(t+1)} \geq \ell_{y_i,i}^{(t)} &= \frac{e^{F_{-y_i}(\mathbf{W}^{(t)}, \mathbf{x}_i)}}{\sum_{j \in \{-1,1\}} e^{F_j(\mathbf{W}^{(t)}, \mathbf{x}_i)}} \\
&\geq \exp\left( -\sum_{r=1}^m \left[ \sigma(\langle \mathbf{w}_{y_i,r}^{(t)}, y_i \mathbf{v} \rangle) + \sigma(\langle \mathbf{w}_{y_i,r}^{(t)}, \boldsymbol{\xi}_i \rangle) \right] \right) \\
&\geq \exp\left( -\Theta\left( m \max_{r \in [m]} \sigma(\langle \mathbf{w}_{y_i,r}^{(t)}, \boldsymbol{\xi}_i \rangle) \right) \right),
\end{aligned}
\tag{B.19}
$$

where in the last inequality we use Hypothesis (B.14). Then by the fact that $\ell_{y_i,i}^{(t+1)} \leq \widetilde{O}\left(\frac{n\lambda}{s\sigma_p^2}\right) = o(1)$ and $m = \widetilde{\Theta}(1)$, it is clear that $\exp\left(-\Theta\left(m\max_{r\in[m]}\sigma(\langle\mathbf{w}_{y_i,r}^{(t+1)},\boldsymbol{\xi}_i\rangle)\right)\right) = o(1)$ so that $\max_{r\in[m]}\langle\mathbf{w}_{y_i,r}^{(t+1)},\boldsymbol{\xi}_i\rangle = \widetilde{\Omega}(1)$. This verifies Hypothesis (B.12).

**Verifying Hypothesis (B.13).** Now we will verify Hypothesis (B.13). First, note that we have already shown that $\langle\mathbf{w}_{y_i,r^*}^{(t+1)},\boldsymbol{\xi}_i\rangle = \widetilde{\Omega}(1)$ so it holds that

$$\sum_{k\in\mathcal{B}_i} |\mathbf{w}_{y_i,r^*}^{(t+1)}[k]| \cdot |\boldsymbol{\xi}_i[k]| + \alpha|\mathbf{w}_{y_i,r^*}^{(t+1)}[1]| \geq \langle\mathbf{w}_{y_i,r^*}^{(t+1)},\boldsymbol{\xi}_i\rangle = \widetilde{\Omega}(1).$$

By Hypothesis (B.14), we have $|\mathbf{w}_{y_i,r^*}^{(t+1)}[1]| \leq |\mathbf{w}_{y_i,r^*}^{(t)}[1]| + \eta = o(1)$. Besides, since each coordinate in $\boldsymbol{\xi}_i$ is a Gaussian random variable, then $\max_{k\in\mathcal{B}_i}|\boldsymbol{\xi}_i[k]| = \widetilde{O}(\sigma_p)$. This immediately implies that

$$\sum_{k\in\mathcal{B}_i} |\mathbf{w}_{y_i,r^*}^{(t+1)}[k]| \cdot |\boldsymbol{\xi}_i[k]| = \widetilde{\Omega}(1).$$

Then we will prove the upper bound of $\sum_{k\in\mathcal{B}_i}|\mathbf{w}_{y_i,r}^{(t+1)}[k]|\cdot|\boldsymbol{\xi}_i[k]|$. Recall that by Lemma B.2, for any $k\in\mathcal{B}_i$ such that $\nabla_{\mathbf{w}_{y_i,r}}L(\mathbf{W}^{(t)})[k] \geq \widetilde{\Theta}(n^{-1}\eta s\sigma_p\ell_{y_i,i}^{(t)})$, we have

$$\mathbf{w}_{y_i,r}^{(t+1)}[k] = \mathbf{w}_{y_i,r}^{(t)}[k] + \Theta(\eta)\cdot\mathrm{sgn}\left(\ell_{y_i,i}^{(t)}\sigma'(\langle\mathbf{w}_{y_i,r}^{(t)},\boldsymbol{\xi}_i\rangle)\boldsymbol{\xi}_i[k] - n\lambda\mathbf{w}_{y_i,r}^{(t)}[k]\right).$$

Note that by Lemma B.4, for every $k\in\mathcal{B}_i$, we have either $\mathbf{w}_{y_i,r}^{(T_0)}[k] = \mathrm{sgn}(\boldsymbol{\xi}_i[k])\cdot\widetilde{\Theta}\left(\frac{1}{s\sigma_p}\right)$ or $|\mathbf{w}_{y_i,r}^{(T_0)}[k]| \leq \eta$. Then during the training process after $T_0$, we have either $\mathrm{sgn}(\mathbf{w}_{y_i,r}^{(t)}[k]) = \mathrm{sgn}(\boldsymbol{\xi}_i[k])$ or $\mathrm{sgn}(\boldsymbol{\xi}_i[k])\cdot\mathbf{w}_{y_i,r}^{(t)} \geq -\widetilde{O}(\eta)$ since if for some iteration number $t'$ that we have $\mathrm{sgn}(\mathbf{w}_{y_i,r}^{(t')}[k]) = -\mathrm{sgn}(\boldsymbol{\xi}_i[k])$ but $\mathrm{sgn}(\mathbf{w}_{y_i,r}^{(t'-1)}[k]) = \mathrm{sgn}(\boldsymbol{\xi}_i[k])$, then after $\bar{\tau} = \widetilde{O}(1)$ steps (see the proof of Lemma B.2 for the definition of $\bar{\tau}$) in the constant number of steps the gradient will must be in the same direction of $\boldsymbol{\xi}_i[k]$, which will push $\mathbf{w}_{y_i,r}[k]$ back to zero or become positive along the direction of $\boldsymbol{\xi}_i[k]$. Therefore, based on this property we have the following regarding the inner product $\langle\mathbf{w}_{y_i,r}^{(t)},\boldsymbol{\xi}_i\rangle$,

$$\begin{aligned}
\langle\mathbf{w}_{y_i,r}^{(t)},\boldsymbol{\xi}_i\rangle &= \sum_{k\in\mathcal{B}_i\cup\{1\}} \mathbf{w}_{y_i,r}^{(t)}[k]\cdot\boldsymbol{\xi}_i[k] \\
&\geq \sum_{k\in\mathcal{B}_i\cup\{1\}} |\mathbf{w}_{y_i,r}^{(t)}[k]|\cdot|\boldsymbol{\xi}_i[k]| - \widetilde{O}(\eta)\cdot\sum_{k\in\mathcal{B}_i\cup\{1\}}|\boldsymbol{\xi}_i[k]| \\
&= \sum_{k\in\mathcal{B}_i\cup\{1\}} |\mathbf{w}_{y_i,r}^{(t)}[k]|\cdot|\boldsymbol{\xi}_i[k]| - \widetilde{O}(\eta s\sigma_p),
\end{aligned}$$

where the second inequality follows from the fact that the entry $\mathbf{w}_{y_i,r}^{(t)}[k]$ that has different sign of $\boldsymbol{\xi}_i[k]$ satisfies $|\mathbf{w}_{y_i,r}^{(t)}[k]| \leq \widetilde{O}(\eta)$. Then let $B_i^{(t)} = \sum_{j\in\mathcal{B}_i\cup\{1\}}|\mathbf{w}_{y_i,r}^{(t)}[k]|\cdot\mathbb{1}(|\mathbf{w}_{y_i,r}^{(t)}[k]|\geq\widetilde{O}(\eta))|\cdot|\boldsymbol{\xi}_i[k]|$, which satisfies $B_i^{(T_0)} = \widetilde{\Theta}(1)$ by Lemma B.4. Then assume $B_i^{(t)}$ keeps increasing and reaches some value in the order of $\Theta\left(\log(dn\eta^{-1})\right)$, it holds that according to the inequality above

$$\langle\mathbf{w}_{y_i,r}^{(t)},\boldsymbol{\xi}_i\rangle = \Theta\left(\log(dn\eta^{-1})\right) - \widetilde{\Theta}(\eta s\sigma_p) = \Theta\left(\log(dn\eta^{-1})\right),$$

where we use the condition that $\eta = O\left((s\sigma_p)^{-1}\right)$. Then by Hypothesis (B.12) and (B.14) we know that $|\langle\mathbf{w}_{j,r}^{(t)},\mathbf{v}\rangle| = o(1)$, $\langle\mathbf{w}_{y_i,r^*}^{(t)},\boldsymbol{\xi}_i\rangle = \widetilde{\Omega}(1)$, and $|\langle\mathbf{w}_{-y_i,r^*}^{(t)},\boldsymbol{\xi}_i\rangle| = \widetilde{O}(d\eta)+\alpha|\langle\mathbf{w}_{-y_i,r^*}^{(t)},\mathbf{v}\rangle| = o(1)$ then similar to (B.19), it holds that

$$\ell_{y_i,i}^{(t)} = \frac{e^{F_{-y_i}(\mathbf{W}^{(t)},\mathbf{x}_i)}}{\sum_{j\in\{-1,1\}}e^{F_j(\mathbf{W}^{(t)},\mathbf{x}_i)}} \leq \exp\left(-\Theta\left(\sigma(\langle\mathbf{w}_{y_i,r^*}^{(t)},\boldsymbol{\xi}_i\rangle)\right)\right) \leq \mathrm{poly}(d^{-1},n^{-1},\eta).$$

Therefore, at this time we have for all $k\in\mathcal{B}_i$,

$$\ell_{y_i,i}^{(t)}\sigma'\langle(\mathbf{w}_{y_i,r}^{(t)},\boldsymbol{\xi}_i\rangle)\boldsymbol{\xi}_i[k] \leq \mathrm{poly}(d^{-1},n^{-1},\eta)\cdot\Theta\left(\log^{q-1}(dn\eta^{-1})\right)\cdot\widetilde{\Theta}(\sigma_p) \leq n\lambda\eta.$$

Then for all $|\mathbf{w}_{y_i,r}^{(t)}[k]| \geq \widetilde{O}(\eta)$, the sign of the gradient satisfies

$$\text{sgn}\big(\nabla_{\mathbf{w}_{y_i,r}}L(\mathbf{W}^{(t)})[k]\big) = -\text{sgn}\bigg(\ell_{y_i,i}^{(t)}\sigma'(\langle\mathbf{w}_{y_i,r}^{(t)},\boldsymbol{\xi}_i\rangle)\boldsymbol{\xi}_i[k] - n\lambda\mathbf{w}_{y_i,r}^{(t)}[k]\bigg)$$

$$= \text{sgn}(n\lambda\eta - \mathbf{w}_{y_i,r}^{(t)}[k])$$

$$= \text{sgn}(\mathbf{w}_{y_i,r}^{(t)}[k]).$$

Then note that $|\nabla_{\mathbf{w}_{y_i,r}}L(\mathbf{W}^{(t)})[k]| = \Theta(|\lambda\mathbf{w}_{y_i,r}^{(t)}[k]|) \geq \Theta(n^{-1}\eta s\sigma_p\ell_{y_i,i}^{(t)} + \lambda\eta)$, by the update rule of $\mathbf{w}_{y_i,r}^{(t)}[k]$ and Lemma B.2, we know the sign gradient will dominate the update process. Then we have $|\mathbf{w}_{y_i,r}^{(t+1)}[k]| = |\mathbf{w}_{y_i,r}^{(t)}[k]| - \Theta(\eta)\cdot\text{sgn}(\mathbf{w}_{y_i,r}^{(t)}[k])| \leq |\mathbf{w}_{y_i,r}^{(t)}[k]|$, which implies that $\big|\mathbf{w}_{y_i,r}^{(t)}[k]\cdot\mathbb{1}(|\mathbf{w}_{y_i,r}^{(t)}[k]| \geq \widetilde{O}(\eta))\big|$ decreases so that $B_i^{(t)}$ also decreases. Therefore, we can conclude that $B_i^{(t)}$ will not exceed $\Theta\big(\log(dn\eta^{-1})\big)$. Then combining the results for all $i \in [n]$ gives

$$\sum_{k\in\mathcal{B}_i}|\mathbf{w}_{y_i,r^*}^{(t)}[k]|\cdot|\boldsymbol{\xi}_i[k]| \leq B_i^{(t)} + \widetilde{O}(s\eta\sigma_p) \leq \Theta\big(\log(dn\eta^{-1})\big) + O(1) = \widetilde{\Theta}(1),$$

where in the first inequality we again use the condition that $\eta = o(1/d) = o\big((s\sigma_p)^{-1}\big)$. This verifies Hypothesis (B.13). Notably, this also implies that $\langle\mathbf{w}_{y_i,r^*}^{(t)},\boldsymbol{\xi}_i\rangle = \max_{r\in[m]}\langle\mathbf{w}_{y_i,r}^{(t)},\boldsymbol{\xi}_i\rangle \leq \widetilde{\Theta}(1)$.

**Verifying Hypothesis (B.14).** In order to verify Hypothesis (B.14), let us first recall the update rule of $\langle\mathbf{w}_{j,r}^{(t)},\mathbf{v}\rangle$:

$$\langle\mathbf{w}_{j,r}^{(t+1)},\mathbf{v}\rangle = \langle\mathbf{w}_{j,r}^{(t)},\mathbf{v}\rangle - \eta\bigg\langle\frac{\mathbf{m}_{j,r}^{(t)}}{\sqrt{\mathbf{v}_{j,r}^{(t)}}},\mathbf{v}\bigg\rangle.$$

Then by Lemma B.2, we know that if $|\nabla_{\mathbf{w}_{j,r}}L(\mathbf{W}^{(t)})[1]| \leq \widetilde{\Theta}(\eta)$, then $|\mathbf{m}_{j,r}^{(t)}/\sqrt{\mathbf{v}_{j,r}^{(t)}}| \leq \Theta(1)$ and otherwise

$$\bigg\langle\frac{\mathbf{m}_{j,r}^{(t)}}{\sqrt{\mathbf{v}_{j,r}^{(t)}}},\mathbf{v}\bigg\rangle = -\text{sgn}\bigg(\sum_{i=1}^n y_i\ell_{j,i}^{(t)}\sigma'(\langle\mathbf{w}_{j,r}^{(t)},y_i\mathbf{v}\rangle) - \alpha\sum_{i=1}^n y_i\ell_{j,i}^{(t)}\sigma'(\langle\mathbf{w}_{j,r}^{(t)},\boldsymbol{\xi}_i\rangle) - n\lambda\mathbf{w}_{j,r}^{(t)}[1]\bigg)\cdot\Theta(1).$$

Without loss of generality we assume $j = 1$, then by Lemma B.4 we know that $\mathbf{w}_{1,r}^{(T_0)}[1] = -\widetilde{\Omega}\big(\frac{1}{s\sigma_p}\big)$. In the remaining proof, we will show that either $\mathbf{w}_{1,r}^{(t+1)}[1] \in [0,\widetilde{\Theta}(\lambda^{-1}\eta)]$ or $\mathbf{w}_{1,r}^{(t+1)}[1] \in \big[-\widetilde{O}\big(\frac{n\alpha}{s\sigma_p^2}\big),0\big)$.

First we will show that $\mathbf{w}_{1,r}^{(t+1)}[1] \in [0,\widetilde{\Theta}(\lambda^{-1}\eta)]$ for all $r$. Note that in the beginning of this stage, we have $\mathbf{w}_{1,r}^{(T_0)}[1] < 0$. In order to make the sign of $\mathbf{w}_{1,r}^{(t')}[1]$ flip, we must have, in some iteration $t' \leq t$ that satisfies $\mathbf{w}_{1,r}^{(t')}[1] \in [0,\widetilde{\Theta}(\lambda^{-1}\eta)]$, therefore

$$-n\nabla_{\mathbf{w}_{1,r}}L(\mathbf{W}^{(t')})[1] = \sum_{i=1}^n y_i\ell_{j,i}^{(t')}\sigma'(\langle\mathbf{w}_{j,r}^{(t')},y_i\mathbf{v}\rangle) - \alpha\sum_{i=1}^n y_i\ell_{j,i}^{(t')}\sigma'(\langle\mathbf{w}_{j,r}^{(t')},\boldsymbol{\xi}_i\rangle) - n\lambda\mathbf{w}_{j,r}^{(t')}[1]$$

$$\leq n\big[(\mathbf{w}_{j,r}^{(t')}[1])^{q-2} - \lambda\big]\cdot\mathbf{w}_{j,r}^{(t')}[1] \leq -\widetilde{\Theta}(n\eta) \leq 0,$$

where the second inequality holds since $\eta = o(\lambda^{(q-1)/(q-2)})$. Note that $|\nabla_{\mathbf{w}_{1,r}}L(\mathbf{W}^{(t')})[1]| \geq \widetilde{\Theta}(\eta)$, then by Lemma B.2 we know that Adam is similar to sign gradient descent and thus $\mathbf{w}_{1,r}^{(t'+1)}[1] = \mathbf{w}_{1,r}^{(t')}[1] - \Theta(\eta)$ which starts to decrease. This implies that if $\mathbf{w}_{1,r}^{(t+1)}[1]$ is positive, then it cannot exceed $\widetilde{\Theta}(\lambda^{-1}\eta) = o(1)$.

Then we can prove that if $\mathbf{w}_{1,r}^{(t+1)}[1]$ is negative, then $|\mathbf{w}_{1,r}^{(t+1)}[1]| = \widetilde{O}\big(\frac{n\alpha}{s\sigma_p^2}\big)$. In this case we have for all $t' \leq t$,

$$-n\nabla_{\mathbf{w}_{1,r}^{(t)}}L(\mathbf{W}^{(t')})[1] = \sum_{i=1}^n y_i\ell_{1,i}^{(t')}\sigma'(\langle\mathbf{w}_{1,r}^{(t')},y_i\mathbf{v}\rangle) - \alpha\sum_{i=1}^n y_i\ell_{1,i}^{(t')}\sigma'(\langle\mathbf{w}_{1,r}^{(t')},\boldsymbol{\xi}_i\rangle) - n\lambda\mathbf{w}_{1,r}^{(t')}[1]$$

$$\geq - \sum_{i:y_i=1} |\ell_{1,i}^{(t')}| \cdot \widetilde{\Theta}(\alpha) + n\lambda |\mathbf{w}_{1,r}^{(t')}[1]| + \sum_{i:y_i=-1} |\ell_{1,i}^{(t')}| \cdot |\mathbf{w}_{1,r}^{(t')}[1]|^{q-1},$$

$$\geq - \sum_{i:y_i=1} |\ell_{1,i}^{(t')}| \cdot \widetilde{\Theta}(\alpha) + n\lambda |\mathbf{w}_{1,r}^{(t')}[1]|,$$

where in the inequality we use Hypothesis (B.13) and (B.14) to get that

$$\langle \mathbf{w}_{y_i,r}^{(t')}, \boldsymbol{\xi}_i \rangle \leq \sum_{k \in \mathcal{B}_i} |\mathbf{w}_{y_i,r}^{(t')}[k]| \cdot \max_{k \in \mathcal{B}_i} |\boldsymbol{\xi}_i[k]| + \alpha |\langle \mathbf{w}_{y_i,r}^{(t')}, \mathbf{v} \rangle| = \widetilde{\Theta}(1).$$

Recall from (B.18) that we have $|\ell_{j,i}^{(t')}| = \widetilde{O}\left(\frac{n\lambda}{s\sigma_p^2}\right)$, therefore we have if $\mathbf{w}_{j,r}^{(t')}[1]$ is smaller than some value in the order of $-\widetilde{\Theta}\left(\frac{n\alpha}{s\sigma_p^2}\right) \cdot \text{polylog}(d)$, then

$$-n\nabla_{\mathbf{w}_{1,r}^{(t)}} L(\mathbf{W}^{(t')})[1] \geq -\widetilde{\Theta}\left(\frac{\alpha n^2\lambda}{s\sigma_p^2}\right) + \widetilde{\Theta}\left(\frac{n\lambda \cdot n\alpha}{s\sigma_p^2}\right) \cdot \text{polylog}(d) \geq \widetilde{\Theta}(n\eta),$$

which by Lemma B.2 implies that $\mathbf{w}_{j,r}^{(t')}[1]$ will increase. Therefore, we can conclude that $\mathbf{w}^{(t+1)} \in \left[ -\widetilde{O}\left(\frac{n\alpha}{s\sigma_p^2}\right), 0 \right)$ in this case, which verifies Hypothesis (B.14). $\qquad\square$

**Lemma B.6** (Lemma 5.5, restated). If the step size satisfies $\eta = O(d^{-1/2})$, then for any $t$ it holds that

$$L(\mathbf{W}^{(t+1)}) - L(\mathbf{W}^{(t)}) \leq -\eta \|\nabla L(\mathbf{W}^{(t)})\|_1 + \widetilde{\Theta}(\eta^2 d).$$

*Proof.* Let $\Delta F_{j,i} = F_j(\mathbf{W}^{(t+1)}, \mathbf{x}_i) - F_j(\mathbf{W}^{(t)}, \mathbf{x}_i)$. Then regarding the loss function

$$L_i(\mathbf{W}) = -\log \frac{e^{F_{y_i}(\mathbf{W}, \mathbf{x}_i)}}{\sum_j e^{F_j(\mathbf{W}, \mathbf{x}_i)}} = -F_{y_i}(\mathbf{W}, \mathbf{x}_i) + \log\left(\sum_j e^{F_j(\mathbf{W}, \mathbf{x}_i)}\right).$$

It is clear that the function $L_i(\mathbf{W})$ is 1-smooth with respect to the vector $[F_{-1}(\mathbf{W}, \mathbf{x}_i), F_1(\mathbf{W}, \mathbf{x}_i)]$. Then based on the definition of $\Delta F_{j,i}$, we have

$$L_i(\mathbf{W}^{(t+1)}) - L_i(\mathbf{W}^{(t)}) \leq \sum_j \frac{\partial L_i(\mathbf{W}^{(t)})}{\partial F_j(\mathbf{W}^{(t)}, \mathbf{x}_i)} \cdot \Delta F_{j,i} + \sum_j (\Delta F_{j,i})^2. \qquad \text{(B.20)}$$

Moreover, note that

$$F_j(\mathbf{W}^{(t)}, \mathbf{x}_i) = \sum_{r=1}^m \left[ \sigma(\langle \mathbf{w}_{j,r}^{(t)}, y_i\mathbf{v} \rangle) + \sigma(\langle \mathbf{w}_{j,r}^{(t)}, \boldsymbol{\xi}_i \rangle) \right].$$

By the results that $\langle \mathbf{w}_{j,r}^{(t)}, \mathbf{v} \rangle \leq \widetilde{\Theta}(1)$ and $\langle \mathbf{w}_{j,r}^{(t)}, \boldsymbol{\xi} \rangle \leq \widetilde{\Theta}(1)$, for any $\eta = O(d^{-1/2})$, we have

$$\langle \mathbf{w}_{j,r}^{(t+1)}, \mathbf{v} \rangle \leq \langle \mathbf{w}_{j,r}^{(t)}, \mathbf{v} \rangle + \eta \leq \widetilde{\Theta}(1), \quad \langle \mathbf{w}_{j,r}^{(t+1)}, \boldsymbol{\xi}_i \rangle \leq \langle \mathbf{w}_{j,r}^{(t)}, \boldsymbol{\xi}_i \rangle + \widetilde{\Theta}(\eta s^{1/2}) \leq \widetilde{\Theta}(1),$$

which implies that the smoothness parameter of the functions $\sigma(\langle \mathbf{w}_{j,r}^{(t)}, y_i\mathbf{v} \rangle)$ and $\sigma(\langle \mathbf{w}_{j,r}^{(t)}, \boldsymbol{\xi}_i \rangle)$ are at most $\widetilde{\Theta}(1)$ for any $\mathbf{w}$ in the path between $\mathbf{w}_{j,r}^{(t)}$ and $\mathbf{w}_{j,r}^{(t+1)}$. Then we can apply first Taylor expansion on $\sigma(\langle \mathbf{w}_{j,r}^{(t)}, y_i\mathbf{v} \rangle)$ and $\sigma(\langle \mathbf{w}_{j,r}^{(t)}, \boldsymbol{\xi}_i \rangle)$ and bound the second-order error as follows,

$$\left| \sigma(\langle \mathbf{w}_{j,r}^{(t+1)}, y_i\mathbf{v} \rangle) - \sigma(\langle \mathbf{w}_{j,r}^{(t)}, y_i\mathbf{v} \rangle) - \langle \nabla_{\mathbf{w}_{j,r}} \sigma(\langle \mathbf{w}_{j,r}^{(t)}, y_i\mathbf{v} \rangle), \mathbf{w}_{j,r}^{(t+1)} - \mathbf{w}_{j,r}^{(t)} \rangle \right|$$
$$\leq \widetilde{\Theta}\left( \|\mathbf{w}_{j,r}^{(t+1)} - \mathbf{w}_{j,r}^{(t)}\|_2^2 \right) = \widetilde{\Theta}(\eta^2 d), \qquad \text{(B.21)}$$

where the last inequality is due to Lemma B.2 that

$$[\mathbf{w}_{j,r}^{(t+1)} - \mathbf{w}_{j,r}^{(t)}]^2 = \eta^2 \left\| \frac{\mathbf{m}_{j,r}^{(t)}}{\sqrt{\mathbf{v}_{j,r}^{(t)}}} \right\|_2^2 \leq \Theta(\eta^2 d).$$

Similarly, we can also show that

$$\left|\sigma(\langle \mathbf{w}_{j,r}^{(t+1)}, \boldsymbol{\xi}_i\rangle) - \sigma(\langle \mathbf{w}_{j,r}^{(t)}, \boldsymbol{\xi}_i\rangle) - \langle \nabla_{\mathbf{w}_{j,r}}\sigma(\langle \mathbf{w}_{j,r}^{(t)}, \boldsymbol{\xi}_i\rangle), \mathbf{w}_{j,r}^{(t+1)} - \mathbf{w}_{j,r}^{(t)}\rangle\right| \le \Theta(\eta^2 d). \quad \text{(B.22)}$$

Combining the above bounds on the second-order errors, we have

$$\left|\Delta F_{j,i} - \langle \nabla_{\mathbf{W}} F_j(\mathbf{W}^{(t)}, \mathbf{x}_i), \mathbf{W}^{(t+1)} - \mathbf{W}^{(t)}\rangle\right| \le \widetilde{\Theta}(m\eta^2 d) = \widetilde{\Theta}(\eta^2 d), \quad \text{(B.23)}$$

where the last equation is due to our assumption that $m = \widetilde{\Theta}(1)$. Besides, by (B.21) and (B.22) the convexity property of the function $\sigma(x)$, we also have

$$
\begin{aligned}
\left|\sigma(\langle \mathbf{w}_{j,r}^{(t+1)}, y_i\mathbf{v}\rangle) - \sigma(\langle \mathbf{w}_{j,r}^{(t)}, y_i\mathbf{v}\rangle)\right| &\le \left|\langle \nabla_{\mathbf{w}_{j,r}}\sigma(\langle \mathbf{w}_{j,r}^{(t)}, y_i\mathbf{v}\rangle), \mathbf{w}_{j,r}^{(t+1)} - \mathbf{w}_{j,r}^{(t)}\rangle\right| + \widetilde{\Theta}(\eta^2 d) \\
&= \widetilde{\Theta}\big(\eta|\sigma'(\langle \mathbf{w}_{j,r}^{(t+1)}, y_i\mathbf{v}\rangle)| \cdot \|\mathbf{v}\|_1\big) + \widetilde{\Theta}(\eta^2 d) \\
&= \widetilde{\Theta}(\eta + \eta^2 d); \\
\left|\sigma(\langle \mathbf{w}_{j,r}^{(t+1)}, \boldsymbol{\xi}_i\rangle) - \sigma(\langle \mathbf{w}_{j,r}^{(t)}, \boldsymbol{\xi}_i\rangle)\right| &\le \left|\langle \nabla_{\mathbf{w}_{j,r}}\sigma(\langle \mathbf{w}_{j,r}^{(t)}, \boldsymbol{\xi}_i\rangle), \mathbf{w}_{j,r}^{(t+1)} - \mathbf{w}_{j,r}^{(t)}\rangle\right| + \widetilde{\Theta}(\eta^2 d) \\
&= \widetilde{\Theta}\big(\eta|\sigma'(\langle \mathbf{w}_{j,r}^{(t+1)}, \boldsymbol{\xi}_i\rangle)| \cdot \|\boldsymbol{\xi}\|_1\big) + \widetilde{\Theta}(\eta^2 d) \\
&= \widetilde{\Theta}(\eta s\sigma_p + \eta^2 d).
\end{aligned}
$$

These bounds further imply that

$$|\Delta F_{j,i}| \le \widetilde{\Theta}\big(m \cdot (\eta s\sigma_p + \eta^2 d)\big) = \widetilde{\Theta}\big(\eta s\sigma_p + \eta^2 d\big). \quad \text{(B.24)}$$

Now we can plug (B.23) and (B.24) into (B.20) and get

$$
\begin{aligned}
L_i(\mathbf{W}^{(t+1)}) - L_i(\mathbf{W}^{(t)}) &\le \sum_j \frac{\partial L_i(\mathbf{W}^{(t)})}{\partial F_j(\mathbf{W}^{(t)}, \mathbf{x}_i)} \cdot \Delta F_{j,i} + \sum_j (\Delta F_{j,i})^2 \\
&\le \sum_j \frac{\partial L_i(\mathbf{W}^{(t)})}{\partial F_j(\mathbf{W}^{(t)}, \mathbf{x}_i)} \cdot \langle \nabla_{\mathbf{W}} F_j(\mathbf{W}^{(t)}, \mathbf{x}_i), \mathbf{W}^{(t+1)} - \mathbf{W}^{(t)}\rangle \\
&\quad + \widetilde{\Theta}(\eta^2 d) + \widetilde{\Theta}\big((\eta s\sigma_p + \eta^2 d)^2\big) \\
&= \langle \nabla L_i(\mathbf{W}^{(t)}), \mathbf{W}^{(t+1)} - \mathbf{W}^{(t)}\rangle + \widetilde{\Theta}(\eta^2 d), \quad \text{(B.25)}
\end{aligned}
$$

where in the second inequality we use the fact that $L_i(\mathbf{W})$ is 1-Lipschitz with respect to $F_j(\mathbf{W}, \mathbf{x}_i)$ and the last equation is due to our assumption that $\sigma_p = O(s^{-1/2})$ so that $\widetilde{\Theta}((\eta s\sigma_p + \eta^2 d)^2) = \widetilde{O}(\eta^2 d)$.

Now we are ready to characterize the behavior on the entire training objective $L(\mathbf{W}) = n^{-1}\sum_{i=1}^n L_i(\mathbf{W}) + \lambda\|\mathbf{W}\|_F^2$. Note that $\lambda\|\mathbf{W}\|_F^2$ is $2\lambda$-smoothness, where $\lambda = o(1)$. Then applying (B.25) for all $i \in [n]$ gives

$$
\begin{aligned}
L(\mathbf{W}^{(t+1)}) - L(\mathbf{W}^{(t)}) &= \frac{1}{n}\sum_{i=1}^n \left[L_i(\mathbf{W}^{(t+1)}) - L_i(\mathbf{W}^{(t)})\right] + \lambda\big(\|\mathbf{W}^{(t+1)}\|_F^2 - \|\mathbf{W}^{(t)}\|_F^2\big) \\
&\le \langle \nabla L(\mathbf{W}^{(t)}), \mathbf{W}^{(t+1)} - \mathbf{W}^{(t)}\rangle + \widetilde{\Theta}(\eta^2 d),
\end{aligned}
$$

where the second equation uses the fact that $\|\mathbf{W}^{(t+1)} - \mathbf{W}^{(t)}\|_F^2 = \widetilde{\Theta}(\eta^2 d)$. Recall that we have

$$\mathbf{w}_{j,r}^{(t+1)} - \mathbf{w}_{j,r}^{(t)} = -\eta \cdot \frac{\mathbf{m}_{j,r}^{(t)}}{\sqrt{\mathbf{v}_{j,r}^{(t)}}}.$$

Then by Lemma B.2, we know that $\mathbf{m}_{j,r}^{(t)}[k]/\sqrt{\mathbf{v}_{j,r}^{(t)}[k]}$ is close to sign gradient if $\nabla L(\mathbf{w}^{(t)})[k]$ is large. Then we have

$$\left\langle \nabla_{\mathbf{w}_{j,r}} L(\mathbf{W}^{(t)}), \frac{\mathbf{m}_{j,r}^{(t)}}{\sqrt{\mathbf{v}_{j,r}^{(t)}}}\right\rangle \ge \Theta\big(\|\nabla_{\mathbf{w}_{j,r}} L(\mathbf{W}^{(t)})\|_1\big) - \widetilde{\Theta}\big(d \cdot \eta\big) - \widetilde{\Theta}(ns \cdot \eta s\sigma_p)$$

$$\geq \Theta\big(\big\|\nabla_{\mathbf{w}_{j,r}} L(\mathbf{W}^{(t)})\big\|_1\big) - \widetilde{\Theta}(d\eta),$$

where the second and last terms on the R.H.S. of the first inequality are contributed by the small gradient coordinates $k \notin \cup_{i=1}^n \mathcal{B}_i$ and $k \in \cup_{i=1}^n \mathcal{B}_i$ respectively, and the last inequality is by the fact that $ns^2\sigma_p = O(d)$. Therefore, based on this fact (B.25) further leads to

$$L(\mathbf{W}^{(t+1)}) - L(\mathbf{W}^{(t)}) \leq -\eta \|\nabla L(\mathbf{W}^{(t)})\|_1 + \widetilde{\Theta}(\eta^2 d),$$

which completes the proof.

$\square$

**Lemma B.7** (Generalization Performance of Adam). Let

$$\mathbf{W}^* = \underset{\mathbf{W} \in \{\mathbf{W}^{(1)},\ldots,\mathbf{W}^{(T)}\}}{\arg\min} \|\nabla L(\mathbf{W})\|_1.$$

Then for all training data, we have

$$\frac{1}{n}\sum_{i=1}^n \mathbb{1}\big[F_{y_i}(\mathbf{W}^*, \mathbf{x}_i) \leq F_{-y_i}(\mathbf{W}^*, \mathbf{x}_i)\big] = 0.$$

Moreover, in terms of the test data $(\mathbf{x}, y) \sim \mathcal{D}$, we have

$$\mathbb{P}_{(\mathbf{x},y)\sim\mathcal{D}}\big[F_y(\mathbf{W}^*, \mathbf{x}) \leq F_{-y}(\mathbf{W}^*, \mathbf{x})\big] \geq \frac{1}{2}.$$

*Proof.* By Lemma B.6, we know that the algorithm will converge to a point with very small gradient (up to $O(\eta d)$ in $\ell_1$ norm). Then in terms of a noise vector $\boldsymbol{\xi}_i$, we have

$$\sum_{k\in\mathcal{B}_i}\big|\nabla_{\mathbf{w}_{y_i,r}} L(\mathbf{W}^*)[k]\big| \leq O(\eta d). \tag{B.26}$$

Note that

$$n\nabla_{\mathbf{w}_{y_i,r}} L(\mathbf{W}^*)[k] = \ell_{y_i,i}^*\sigma'(\langle\mathbf{w}_{y_i,r}^*, \boldsymbol{\xi}_i\rangle)\boldsymbol{\xi}_i[k] - n\lambda\mathbf{w}_{y_i,r}^*[k],$$

where $\ell_{y_i,i}^* = 1 - \text{logit}_{y_i}(F^*, \mathbf{x}_i)$. Then by triangle inequality and (B.26), we have for any $r \in [m]$,

$$\bigg|\sum_{k\in\mathcal{B}_i}|\ell_{y_i,i}^*|\sigma'(\langle\mathbf{w}_{y_i,r}^*, \boldsymbol{\xi}_i\rangle)|\boldsymbol{\xi}_i[k]| - n\lambda\sum_{k\in\mathcal{B}_i}|\mathbf{w}_{y_i,r}^*[k]|\bigg| \leq n\sum_{k\in\mathcal{B}_i}\big|\nabla_{\mathbf{w}_{y_i,r}} L(\mathbf{W}^*)[k]\big| \leq O(n\eta d).$$

Then by Lemma B.5, let $r^* = \arg\max_{r\in[m]}\langle\mathbf{w}_{y_i,r}^*, \boldsymbol{\xi}_i\rangle$, we have $\langle\mathbf{w}_{y_i,r^*}, \boldsymbol{\xi}_i\rangle = \widetilde{\Theta}(1)$ and $\sum_{k\in\mathcal{B}_i}|\mathbf{w}_{y_i,r^*}^*[k]| \cdot |\boldsymbol{\xi}_i[k]| = \widetilde{\Theta}(1)$. Note that $|\boldsymbol{\xi}_i[k]| = \widetilde{O}(\sigma_p)$, we have $\sum_{k\in\mathcal{B}_i}|\mathbf{w}_{y_i,r^*}^*[k]| \geq \widetilde{\Theta}(1/\sigma_p)$. Then according to the inequality above, it holds that

$$|\ell_{y_i,i}^*| \cdot \widetilde{\Theta}(s\sigma_p) \geq \widetilde{\Theta}\bigg(n\lambda\sum_{k\in\mathcal{B}_i}|\mathbf{w}_{y_i,r}^*[k]| - n\eta d\bigg) \geq \widetilde{\Theta}\bigg(\frac{n\lambda}{\sigma_p}\bigg),$$

where the second inequality is due to our choice of $\eta$. This further implies that $|\ell_{y_i,i}^*| = |\ell_{-y_i,i}^*| = \widetilde{\Theta}\big(\frac{n\lambda}{s\sigma_p^2}\big)$ by combining the above results with (B.18). Then let us move to the gradient with respect to the first coordinate. In particular, since $\|\nabla L(\mathbf{W}^*)\|_1 \leq O(\eta d)$, we have

$$|n\nabla_{\mathbf{w}_{j,r}} L(\mathbf{W}^*)[1]| = \bigg|\sum_{i=1}^n y_i\ell_{j,i}^*\sigma'(\langle\mathbf{w}_{j,r}^*, y_i\mathbf{v}\rangle) - \alpha\sum_{i=1}^n y_i\ell_{j,i}^*\sigma'(\langle\mathbf{w}_{j,r}^*, \boldsymbol{\xi}_i\rangle) - n\lambda\mathbf{w}_{j,r}^*[1]\bigg|$$

$$\leq O(n\eta d). \tag{B.27}$$

Then note that $\text{sgn}(y_i\ell_{j,i}^*) = \text{sgn}(j)$, it is clear that $\mathbf{w}_{j,r^*}^*[1] \cdot j \leq 0$ since otherwise

$$|n\nabla_{\mathbf{w}_{j,r^*}} L(\mathbf{W}^*)[1]| \geq \bigg|\alpha\sum_{i=1}^n y_i\ell_{j,i}^*\big[\sigma'(\langle\mathbf{w}_{j,r^*}^*, \boldsymbol{\xi}_i\rangle) - \sigma'(\langle\mathbf{w}_{j,r^*}^*, y_i\mathbf{v}\rangle)\big]\bigg| \geq \widetilde{\Theta}\bigg(\frac{\alpha n^2\lambda}{s\sigma_p^2}\bigg) \geq \widetilde{\Omega}(n\eta d),$$

which contradicts (B.27). Therefore, using the fact that $\mathbf{w}_{j,r^*}^*[1] \cdot j \leq 0$, we have

$$|n\nabla_{\mathbf{w}_{j,r^*}} L(\mathbf{W}^*)[1]| = \left| \alpha \sum_{i:y_i=j}^n y_i \ell_{j,i}^* \sigma'(\langle \mathbf{w}_{j,r^*}^*, \boldsymbol{\xi}_i \rangle) - \sum_{i:y_i=-j}^n y_i \ell_{j,i}^* \sigma'(|\mathbf{w}_{j,r^*}^*[1]|) - n\lambda |\mathbf{w}_{j,r^*}^*[1]| \right|.$$

Then applying (B.27) and using the fact that $|\ell_{y_i,i}^*| = |\ell_{-y_i,i}^*| = \widetilde{\Theta}\left(\frac{n\lambda}{s\sigma_p^2}\right)$ for all $i \in [n]$, it is clear that

$$|\mathbf{w}_{j,r^*}^*[1]| \geq \widetilde{\Theta}\left(\alpha^{1/(q-1)} \wedge \frac{n\alpha}{s\sigma_p^2}\right) \geq \widetilde{\Theta}\left(\frac{n\alpha}{s\sigma_p^2}\right),$$

where the second equality is due to our choice of $\sigma_p$ and $\alpha$. Then combining with Lemma B.5 and the fact that $\mathbf{w}_{j,r^*}^*[1] \cdot j < 0$, we have

$$\mathbf{w}_{j,r^*}^*[1] \cdot j \leq -\widetilde{\Theta}\left(\frac{n\alpha}{s\sigma_p^2}\right).$$

Now we are ready to evaluate the training error and test error. In terms of training error, it is clear that by Lemma B.5, we have $\langle \mathbf{w}_{y_i,r^*}^*, \boldsymbol{\xi}_i \rangle \geq \widetilde{\Theta}(1)$, $\langle \mathbf{w}_{y_i,r}^*, \boldsymbol{\xi}_i \rangle \geq -o(1)$, and $|\langle \mathbf{w}_{y_i,r}^*, \mathbf{v} \rangle| = o(1)$, $|\langle \mathbf{w}_{-y_i,r}^*, \boldsymbol{\xi}_i \rangle| = o(1)$. Then we have for any training data $(\mathbf{x}_i, y_i)$,

$$F_{y_i}(\mathbf{W}^*, \mathbf{x}_i) = \sum_{r=1}^m \left[ \sigma(\langle \mathbf{w}_{y_i,r}^*, y_i\mathbf{v} \rangle) + \sigma(\langle \mathbf{w}_{y_i,r}^*, \boldsymbol{\xi}_i \rangle) \right] = \widetilde{\Theta}(1),$$

$$F_{-y_i}(\mathbf{W}^*, \mathbf{x}_i) = \sum_{r=1}^m \left[ \sigma(\langle \mathbf{w}_{-y_i,r}^*, -y_i\mathbf{v} \rangle) + \sigma(\langle \mathbf{w}_{-y_i,r}^*, \boldsymbol{\xi}_i \rangle) \right] = o(1),$$

which directly implies that the NN model $\mathbf{W}^*$ can correctly classify all training data and thus achieve zero training error.

In terms of the test data $(\mathbf{x}, y)$ where $\mathbf{x} = [y\mathbf{v}, \boldsymbol{\xi}]$, which is generated according to Definition 3.1. Note that for each neural, its weight $\mathbf{w}_{j,r}^*$ can be decomposed into two parts: the first coordinate and the rest $d-1$ coordinates. As previously discussed, for any $j \in [2]$ and $r = r^*$, we have $\text{sgn}(j) \cdot \mathbf{w}_{j,r}^*[1] \leq -\widetilde{\Theta}(n\alpha/(s\sigma_p^2))$ and $\text{sgn}(j) \cdot \mathbf{w}_{j,r}^*[1] \leq \widetilde{\Theta}(\lambda^{-1}\eta)$ for $r \neq r^*$. Therefore, using the fact that $\widetilde{\Theta}(n\alpha/(s\sigma_p^2)) = \omega(\lambda^{-1}\eta)$ and Lemma B.5, given the test data $(\mathbf{x}, y)$, we have

$$F_y(\mathbf{W}^*, \mathbf{x}) = \sum_{r=1}^m \left[ \sigma(\langle \mathbf{w}_{y,r}^*, y\mathbf{v} \rangle) + \sigma(\langle \mathbf{w}_{y,r}^*, \boldsymbol{\xi} \rangle) \right]$$

$$\leq \sum_{r=1}^m \widetilde{\Theta}\left( \left[ \alpha \cdot \frac{n\alpha}{s\sigma_p^2} + \zeta_{y,r} \right]_+^q \right),$$

$$F_{-y}(\mathbf{W}^*, \mathbf{x})) = \sum_{r=1}^m \left[ \sigma(\langle \mathbf{w}_{-y,r}^*, y\mathbf{v} \rangle) + \sigma(\langle \mathbf{w}_{-y,r}^*, \boldsymbol{\xi} \rangle) \right]$$

$$\geq \widetilde{\Theta}\left[ |\mathbf{w}_{-y,r^*}^*[1]|^q + [\zeta_{-y,r^*}]_+^q \right]$$

$$\geq \Theta\left( \left[ \frac{n\alpha}{s\sigma_p^2} \right]_+^q + [\zeta_{-y,r^*}]_+^q \right),$$

where the random variables $\zeta_{y,r}$ and $\zeta_{y,r}$ are symmetric and independent of $\mathbf{v}$. Besides, note that $\alpha = o(1)$, it can be clearly shown that $\alpha \cdot n\alpha/(s\sigma_p^2) \ll n\alpha/(s\sigma_p^2)$. Therefore, if the random noise $\zeta_{y,r}$ and $\zeta_{-y,r}$ are dominated by the feature noise term $\langle \mathbf{w}_{-y,r^*}^*, y\mathbf{v} \rangle$, we can directly get that $F_y(\mathbf{W}^*, \mathbf{x}) \leq F_{-y}(\mathbf{W}^*, \mathbf{x}))$ (recall that $m = \widetilde{\Theta}(1)$), which implies that the model has been biased by the feature noise and the true feature vector in the test dataset will not give any "positive" effect to the classification. Also note that $\zeta_y$ and $\zeta_{-y}$ are also independent of $\mathbf{v}$, which implies that if the random noise dominates the feature noise term, the model $\mathbf{W}^*$ will give at least $0.5$ error on test data. In sum, we can conclude that with probability at least $1/2$ it holds that $F_y(\mathbf{W}^*, \mathbf{x}) \leq F_{-y}(\mathbf{W}^*, \mathbf{x})$, which implies that the output of Adam achieves $1/2$ test error. $\qquad \square$

### B.3 PROOF FOR GRADIENT DESCENT

Recall the feature learning and noise memorization of gradient descent can be formulated by

$$
\langle \mathbf{w}_{j,r}^{(t+1)}, j \cdot \mathbf{v} \rangle = (1 - \eta\lambda) \cdot \langle \mathbf{w}_{j,r}^{(t)}, j \cdot \mathbf{v} \rangle
$$
$$
+ \frac{\eta}{n} \cdot j \cdot \left( \sum_{i=1}^{n} y_i \ell_{j,i}^{(t)} \sigma'(\langle \mathbf{w}_{j,r}^{(t)}, y_i \mathbf{v} \rangle) - \alpha \sum_{i=1}^{n} y_i \ell_{j,i}^{(t)} \sigma'(\langle \mathbf{w}_{j,r}^{(t)}, \boldsymbol{\xi}_i \rangle) \right),
$$
$$
\langle \mathbf{w}_{y_i,r}^{(t+1)}, \boldsymbol{\xi}_i \rangle = (1 - \eta\lambda) \cdot \langle \mathbf{w}_{y_i,r}^{(t)}, \boldsymbol{\xi}_i \rangle + \frac{\eta}{n} \cdot \sum_{k \in \mathcal{B}_i} \ell_{y_i,i}^{(t)} \sigma'(\langle \mathbf{w}_{y_i,r}^{(t)}, \boldsymbol{\xi}_i \rangle) \cdot \boldsymbol{\xi}_i[k]^2
$$
$$
+ \frac{\eta\alpha}{n} \cdot \left( \alpha \sum_{s=1}^{n} \ell_{y_i,s}^{(t)} \sigma'(\langle \mathbf{w}_{y_i,r}^{(t)}, \boldsymbol{\xi}_s \rangle) - \sum_{s=1}^{n} y_s \ell_{y_i,s}^{(t)} \sigma'(\langle \mathbf{w}_{y_i,r}^{(t)}, y_s \mathbf{v} \rangle) \right). \quad \text{(B.28)}
$$

Then similar to the analysis for Adam, we decompose the gradient descent process into multiple stages and characterize the algorithmic behaviors separately. The following lemma characterizes the first training stage, i.e., the stage where all outputs $F_j(\mathbf{W}^{(t)}, \mathbf{x}_i)$ remain in the constant level for all $j$ and $i$.

**Lemma B.8.** [Lemma 5.6, restated] Suppose the training data is generated according to Definition 3.1, assume $\lambda = o(\sigma_0^{q-2}\sigma_p/n)$. Let $\Lambda_j^{(t)} = \max_{r \in [m]} \langle \mathbf{w}_{j,r}^{(t+1)}, j \cdot \mathbf{v} \rangle$, $\Gamma_{j,i}^{(t)} = \max_{r \in [m]} \langle \mathbf{w}_{j,r}^{(t)}, \boldsymbol{\xi}_i \rangle$, and $\Gamma_j^{(t)} = \max_{i:y_i=j} \Gamma_{j,i}^{(t)}$. Then let $T_j$ be the iteration number that $\Lambda_j^{(t)}$ reaches $\Theta(1/m)$, we have

$$
T_j = \widetilde{\Theta}(\sigma_0^{2-q}/\eta) \quad \text{for all } j \in \{-1, 1\}.
$$

Moreover, let $T_0 = \max_j\{T_j\}$, then for all $t \le T_0$ it holds that $\Gamma_j^{(t)} = \widetilde{O}(\sigma_0)$ for all $j \in \{-1, 1\}$.

We first provide the following useful lemma.

**Lemma B.9.** Let $\{x_t, y_t\}_{t=1,\dots}$ be two positive sequences that satisfy

$$
x_{t+1} \ge x_t + \eta \cdot A x_t^{q-1},
$$
$$
y_{t+1} \le y_t + \eta \cdot B y_t^{q-1},
$$

for some $A = \Theta(1)$ and $B = o(1)$. Then for any $q \ge 3$ and suppose $y_0 = O(x_0)$ and $\eta < O(x_0)$, we have for every $C \in [x_0, O(1)]$, let $T_x$ be the first iteration such that $x_t \ge C$, then we have $T_x\eta = \Theta(x_0^{2-q})$ and

$$
y_{T_x} \le O(x_0).
$$

*Proof.* By Claim C.20 in Allen-Zhu & Li (2020c), we have $T_x\eta = \Theta(x_0^{2-q})$. Then we will show

$$
y_t \le 2x_0
$$

for all $t \le T_x$. In particular, let $T_x\eta = C'x_0^{2-q}$ for some absolute constant $C'$ and assume $C'B2^{q-1} < 1$ (this is true since $B = o(1)$), we first made the following induction hypothesis on $y_t$ for all $t \le T_a$,

$$
y_t \le y_0 + t\eta B'(2x_0)^{q-1}.
$$

Note that for any $t \le T_0$, this hypothesis clearly implies that

$$
y_t \le y_0 + T_x\eta B'2^{q-1}x_0^{q-1} \le x_0 + CB2^{q-1}x_0^{2-q} \cdot x_0^{q-1} \le 2x_0.
$$

Then we are able to verify the hypothesis at time $t + 1$ based on the recursive upper bound of $y_t$, i.e.,

$$
y_{t+1} \le y_t + \eta \cdot B y_t^{q-1}
$$
$$
\le y_0 + t\eta B(2x_0)^{q-1} + \eta \cdot B y_t^{q-1}
$$
$$
\le y_0 + (t+1)\eta B(2x_0)^{q-1}.
$$

Therefore, we can conclude that $y_t \le 2x_0$ for all $t \le T_x$. This completes the proof. $\qquad\square$

Now we are ready to complete the proof of Lemma B.8.

*Proof of Lemma B.8.* Note that at the initialization, we have $|\langle \mathbf{w}_{j,r}^{(0)}, \mathbf{v} \rangle| = \widetilde{\Theta}(\sigma_0)$ and $|\langle \mathbf{w}_{j,r}^{(0)}, \boldsymbol{\xi}_i \rangle| = \widetilde{\Theta}(s^{1/2}\sigma_p\sigma_0)$. Then it can be shown that

$$F_j(\mathbf{W}^{(0)}, \mathbf{x}_i) = \sum_{r=1}^{m} \left[ \sigma(\langle \mathbf{w}_{j,r}^{(0)}, y_i\mathbf{v} \rangle) + \sigma(\langle \mathbf{w}_{j,r}^{(0)}, \boldsymbol{\xi}_i \rangle) \right] = o(1)$$

for all $j \in \{-1, 1\}$. Then we have

$$|\ell_{j,i}^{(0)}| = \frac{e^{F_j(\mathbf{W}^{(0)}, \mathbf{x}_i)}}{\sum_j e^{F_j(\mathbf{W}^{(0)}, \mathbf{x}_i)}} = \Theta(1).$$

Then we will consider the training period where $|\ell_{j,i}^{(t)}|$ for all $j$, $i$, and $t$. Besides, note that $\text{sgn}(y_i\ell_{j,i}^{(t)}) = j$. Therefore, let $r^* = \arg\max_r \langle \mathbf{w}_{j,r}^{(t-1)}, j \cdot \mathbf{v} \rangle$, (B.28) implies that

$$\begin{aligned}
\Lambda_j^{(t)} &\geq \langle \mathbf{w}_{j,r^*}^{(t-1)}, j \cdot \mathbf{v} \rangle \\
&= (1 - \eta\lambda) \cdot \langle \mathbf{w}_{j,r^*}^{(t-1)}, j \cdot \mathbf{v} \rangle \\
&\quad + \frac{\eta}{n} \cdot \left( \sum_{i=1}^{n} |\ell_{j,i}^{(t-1)}| \sigma'(\langle \mathbf{w}_{j,r^*}^{(t-1)}, y_i\mathbf{v} \rangle) - \alpha \sum_{i=1}^{n} |\ell_{j,i}^{(t-1)}| \sigma'(\langle \mathbf{w}_{j,r^*}^{(t-1)}, \boldsymbol{\xi}_i \rangle) \right) \\
&\geq (1 - \eta\lambda) \cdot \langle \mathbf{w}_{j,r^*}^{(t-1)}, j \cdot \mathbf{v} \rangle + \Theta(\eta) \cdot \left[ \sigma'(\langle \mathbf{w}_{j,r^*}^{(t-1)}, j \cdot \mathbf{v} \rangle) - \alpha\sigma'(\Gamma_j^{(t-1)}) \right] \\
&\geq (1 - \eta\lambda)\Lambda_j^{(t-1)} + \eta \cdot \Theta\left((\Lambda_j^{(t-1)})^{q-1}\right) - \eta \cdot \Theta\left(\alpha(\Gamma_j^{(t-1)})^{q-1}\right). \quad \text{(B.29)}
\end{aligned}$$

Similarly, let $r^* = \arg\max_r \langle \mathbf{w}_{y_i,r}^{(t)}, \boldsymbol{\xi}_i \rangle$, we also have the following according to (B.28)

$$\begin{aligned}
\Gamma_{y_i,i}^{(t)} &= \langle \mathbf{w}_{y_i,r^*}^{(t)}, \boldsymbol{\xi}_i \rangle \leq (1 - \eta\lambda)\langle \mathbf{w}_{y_i,r^*}^{(t-1)}, \boldsymbol{\xi}_i \rangle + \widetilde{\Theta}\left(\frac{\eta s\sigma_p^2}{n}\right) \cdot \sigma'(\langle \mathbf{w}_{y_i,r^*}^{(t-1)}, \boldsymbol{\xi}_i \rangle) \\
&\quad + \Theta\left(\frac{\eta\alpha^2}{n}\right) \cdot \sum_{s=1}^{n} \sigma'(\langle \mathbf{w}_{y_i,r^*}^{(t-1)}, \boldsymbol{\xi}_s \rangle) \\
&\leq \Gamma_{y_i,i}^{(t-1)} + \widetilde{\Theta}\left(\frac{\eta s\sigma_p^2(\Gamma_{y_i,i}^{(t-1)})^{q-1}}{n}\right) + \Theta\left(\frac{\eta\alpha^2}{n} \cdot \sum_{s=1}^{n} (\Gamma_{y_i,s}^{(t-1)})^{q-1}\right).
\end{aligned}$$

Then by our definition of $\Gamma_j^{(t)} = \max_{i \in [n]} \Gamma_{j,i}^{(t)}$, we further get the following for all $j \in \{-1, 1\}$,

$$\Gamma_j^{(t)} \leq \Gamma_j^{(t-1)} + \widetilde{\Theta}\left(\frac{\eta s\sigma_p^2 + n\eta\alpha^2}{n} \cdot (\Gamma_j^{(t-1)})^{q-1}\right) = \Gamma_j^{(t-1)} + \Theta\left(\frac{\eta s\sigma_p^2}{n} \cdot (\Gamma_j^{(t-1)})^{q-1}\right),$$
(B.30)

where the last equation is by our assumption that $\alpha = \widetilde{O}(s\sigma_p^2/n)$.

Then we will prove the main argument for general $t$, which is based on the following two induction hypothesis

$$\Lambda_j^{(t)} \geq \Lambda_j^{(t-1)} + \eta \cdot \Theta\left((\Lambda_j^{(t-1)})^{q-1}\right), \quad \text{(B.31)}$$

$$\Gamma_j^{(t)} \leq \Gamma_j^{(t-1)} + \Theta\left(\frac{\eta s\sigma_p^2}{n} \cdot (\Gamma_j^{(t-1)})^{q-1}\right). \quad \text{(B.32)}$$

Note that when $t = 0$, we have already verified this two hypothesis in (B.29) and (B.30), where we use the fact that $\lambda = o(\sigma_0^{q-2}\sigma_p/n) \leq (\Lambda_j^{(0)})^{q-2}$ and $\alpha = o(1)$. Then at time $t$, based on Hypothesis (B.31) and (B.32) for all $\tau \leq t$, we have

$$\Gamma_j^{(\tau)} \leq O(\Lambda_j^{(\tau)}),$$

as $s\sigma^2/n = o(1)$ and $\Lambda_j^{(t)}$ increases faster than $\Gamma_j^{(t)}$. Besides, we can also show that $\lambda\Gamma_j^{(t)} \leq (\Gamma_j^{(t)})^{q-1}$, which has been verified at time $t = 0$, since $\Gamma_j^{(t)}$ keeps increasing. Therefore, (B.29) implies

$$\Lambda_j^{(t+1)} \geq (1 - \eta\lambda)\Lambda_j^{(t)} + \eta \cdot \Theta\left((\Lambda_j^{(t)})^{q-1}\right) - \eta \cdot \Theta\left(\alpha(\Gamma_j^{(t)})^{q-1}\right)$$

$$\geq \Lambda_j^{(t)} + \eta \cdot \Theta\big((\Lambda_j^{(t)})^{q-1}\big),$$

which verifies Hypothesis (B.31) at $t + 1$. Additionally, (B.30) implies

$$\Gamma_j^{(t+1)} \leq \Gamma_j^{(t)} + \Theta\bigg(\frac{\eta s \sigma_p^2}{n} \cdot \big(\Gamma_j^{(t)}\big)^{q-1}\bigg),$$

which verifies Hypothesis (B.32) at $t + 1$. Then by Lemma B.9, we have that $\Lambda_j^{(t)} = \widetilde{O}(1)$ for all $t \leq T_0 = \widetilde{\Theta}\big((\Lambda_j^{(0)})^{2-q}/\eta\big) = \widetilde{\Theta}(\sigma_0^{2-q}/\eta)$. Moreover, Lemma B.9 also shows that $\Gamma_j^{(t+1)} = O(\Lambda_j^{(0)}) = \widetilde{O}(\sigma_0)$. This completes the proof. $\qquad\square$

**Lemma B.10** (Off-diagonal correlations). For any data $(\mathbf{x}_i, y_i)$ and for any $t \leq T_{-y_i}$, it holds that $\langle \mathbf{w}_{-y_i,r}^{(t)}, \boldsymbol{\xi}_i \rangle \leq \widetilde{\Theta}(\alpha)$.

*Proof.* By the update form of GD, we have for any $k \in \mathcal{B}_i$,

$$\mathbf{w}_{-y_i,r}^{(t+1)}[k] \cdot \boldsymbol{\xi}_i[k] = (1 - \eta\lambda) \cdot \mathbf{w}_{-y_i,r}^{(t)}[k] \cdot \boldsymbol{\xi}_i[k] + \frac{\eta}{n} \cdot \sum_{k \in \mathcal{B}_i} \ell_{-y_i,i}^{(t)} \sigma'(\langle \mathbf{w}_{-y_i,r}^{(t)}, \boldsymbol{\xi}_i \rangle) \cdot \boldsymbol{\xi}_i[k]^2,$$

which keeps decreasing. Therefore, for all $r$ and $i$, we have

$$\langle \mathbf{w}_{-y_i,r}^{(t)}, \boldsymbol{\xi}_i \rangle \leq |\mathbf{w}_{-y_i,r}^{(t)}[1] \cdot \boldsymbol{\xi}_i[1]| + \bigg| \sum_{k \in \mathcal{B}_i} \mathbf{w}_{-y_i,r}^{(0)}[k]\boldsymbol{\xi}_i[k] \bigg|$$

$$\leq \widetilde{\Theta}(\alpha) + \widetilde{\Theta}(\sigma_0 \sigma_p s^{1/2})$$

$$= \widetilde{\Theta}(\alpha),$$

where the second inequality follows from the fact that $|\langle \mathbf{w}_{j,r}^{(t)}, \mathbf{v} \rangle| \leq \widetilde{\Theta}(1)$ for all $t \leq T_j$. This completes the proof. $\qquad\square$

Note that for different $j$, the iteration numbers when $\Lambda_j^{(t)}$ reaches $\widetilde{\Theta}(1/m)$ are different. Without loss of generality, we can assume $T_1 \leq T_{-1}$. Lemma B.8 has provided a clear understanding about how $\Lambda_j^{(t)}$ varies within the iteration range $[0, T_j]$. However, it remains unclear how $\Gamma_1^{(t)}$ varies within the iteration range $[T_1, T_{-1}]$ since in this period we no longer have $|\ell_{j,i}^{(t)}| = \Theta(1)$ and the effect of gradient descent on the feature learning (i.e., increase of $\langle \mathbf{w}_{j,r}, j \cdot \mathbf{v} \rangle$) becomes weaker. In the following lemma we give a characterization of $\Lambda_1^{(t)}$ for every $t \in [T_1, T_{-1}]$.

**Lemma B.11** (Stage I of GD: part II). Without loss of generality assuming $T_1 < T_{-1}$. Then it holds that $\Lambda_1^{(t)} = \widetilde{\Theta}(1)$ for all $t \in [T_1, T_{-1}]$.

*Proof.* Recall from (B.29) that we have the following general lower bound for the increase of $\Lambda_j^{(t)}$

$$\Lambda_j^{(t+1)} \geq (1 - \eta\lambda) \cdot \langle \mathbf{w}_{j,r^*}^{(t)}, j \cdot \mathbf{v} \rangle + \frac{\eta}{n} \cdot \bigg( \sum_{i=1}^n |\ell_{j,i}^{(t)}| \sigma'(\langle \mathbf{w}_{j,r^*}^{(t)}, y_i \mathbf{v} \rangle) - \alpha \sum_{i=1}^n |\ell_{j,i}^{(t)}| \sigma'(\langle \mathbf{w}_{j,r^*}^{(t)}, \boldsymbol{\xi}_i \rangle) \bigg)$$

$$\geq (1 - \eta\lambda)\Lambda_j^{(t)} + \Theta\bigg(\frac{\eta}{n}\bigg) \cdot \sum_{i:y_i=j} |\ell_{j,i}^{(t)}| \cdot \big(\Lambda_j^{(t)}\big)^{q-1} - \Theta(\alpha\eta) \cdot \big(\Gamma_j^{(t)} \vee \widetilde{\Theta}(\alpha)\big)^{q-1}, \quad \text{(B.33)}$$

where the last inequality is by Lemma B.10. Note that by Lemma B.8, we have $\Gamma_j^{(t)} = \widetilde{O}(\sigma_0)$ for all $t \leq T_{-1}$ and . Then the above inequality leads to

$$\Lambda_j^{(t+1)} \geq (1 - \eta\lambda)\Lambda_j^{(t)} + \Theta\bigg(\frac{\eta}{n}\bigg) \cdot \sum_{i:y_i=j} |\ell_{j,i}^{(t)}| \cdot \big(\Lambda_j^{(t)}\big)^{q-1} - \Theta(\alpha^q \eta), \quad \text{(B.34)}$$

where we use the fact that $\alpha = \omega(\sigma_0)$. The the remaining proof consists of two parts: (1) proving $\Lambda_j^{(t)} \geq \Theta(1/m) = \widetilde{\Theta}(1)$ and (2) $\Lambda_j^{(t)} \leq \Theta(\log(1/\lambda))$.

Without loss of generality we consider $j = 1$. Regarding the first part, we first note that Lemma B.8 implies that $\Lambda_1^{(T_1)} \geq \Theta(1/m)$. Then we consider the case when $\Lambda_1^{(t)} \leq \Theta(\log(1/\alpha)/m)$, it holds that for all $y_i = 1$,

$$
\begin{aligned}
\ell_{1,i}^{(t)} &= \frac{e^{F_{-1}(\mathbf{W}^{(t)}, \mathbf{x}_i)}}{\sum_{j \in \{-1,1\}} e^{F_j(\mathbf{W}^{(t)}, \mathbf{x}_i)}} \\
&= \exp\left(\Theta\left(\sum_{r=1}^m \left[\sigma(\langle \mathbf{w}_{-1,r}^{(t)}, y_i \mathbf{v}\rangle) + \sigma(\langle \mathbf{w}_{-1,r}^{(t)}, \boldsymbol{\xi}_i\rangle)\right] - \sum_{r=1}^m \left[\sigma(\langle \mathbf{w}_{1,r}^{(t)}, y_i \mathbf{v}\rangle) + \sigma(\langle \mathbf{w}_{1,r}^{(t)}, \boldsymbol{\xi}_i\rangle)\right]\right)\right) \\
&\geq \exp\left(-\Theta(m\Lambda_1^{(t)})\right) \\
&\geq \exp(-\Theta(\log(1/\alpha))) \\
&= \widetilde{\Theta}(\alpha).
\end{aligned}
$$

Then (B.34) implies that if $\Gamma_1^{(t)} \leq \Theta(\log(1/\sigma_0)/m)$, we have

$$
\Lambda_1^{(t+1)} \geq (1 - \eta\lambda)\Lambda_1^{(t)} + \Theta(\eta\alpha) \cdot \Lambda_1^{(t)} - \Theta(\alpha^q \eta) \geq \Lambda_1^{(t)} + \Theta(\eta\alpha) \cdot \Lambda_1^{(t)} \geq \Lambda_1^{(t)},
$$

where the second inequality is due to $\lambda = o(\alpha)$. This implies that $\Lambda_1^{(t)}$ will keep increases in this case so that it is impossible that $\Lambda_1^{(t)} \leq \Theta(1/m)$, which completes the proof of the first part.

For the second part, (B.28) implies that

$$
\Lambda_1^{(t+1)} \leq (1 - \eta\lambda)\Lambda_1^{(t)} + \Theta\left(\frac{\eta}{n}\right) \cdot \sum_{i:y_i=1} |\ell_{1,i}^{(t)}| \cdot \left(\Lambda_1^{(t)}\right)^{q-1}. \tag{B.35}
$$

Consider the case when $\Gamma_1^{(t)} \geq \Theta(\log(d))$, then for all $y_i = 1$,

$$
\begin{aligned}
\ell_{1,i}^{(t)} &= \frac{e^{F_{-1}(\mathbf{W}^{(t)}, \mathbf{x}_i)}}{\sum_{j \in \{-1,1\}} e^{F_j(\mathbf{W}^{(t)}, \mathbf{x}_i)}} \\
&= \exp\left(\Theta\left(\sum_{r=1}^m \left[\sigma(\langle \mathbf{w}_{-1,r}^{(t)}, y_i \mathbf{v}\rangle) + \sigma(\langle \mathbf{w}_{-1,r}^{(t)}, \boldsymbol{\xi}_i\rangle)\right] - \sum_{r=1}^m \left[\sigma(\langle \mathbf{w}_{1,r}^{(t)}, y_i \mathbf{v}\rangle) + \sigma(\langle \mathbf{w}_{1,r}^{(t)}, \boldsymbol{\xi}_i\rangle)\right]\right)\right) \\
&\leq \exp\left(-\Theta(\Lambda_1^{(t)})\right) \\
&\leq \exp(-\Theta(\log(1/\lambda))) \\
&= \widetilde{\Theta}(\text{poly}(\lambda)).
\end{aligned}
$$

Then (B.35) further implies that

$$
\begin{aligned}
\Lambda_1^{(t+1)} &\leq (1 - \eta\lambda)\Lambda_1^{(t)} + \Theta\left(\frac{\eta}{\text{poly}(d)}\right) \cdot \left(\Lambda_1^{(t)}\right)^{q-1} \\
&\leq \Lambda_1^{(t)} - \Theta(\eta\Lambda_1^{(t)}) \cdot \left(\lambda - \text{poly}(\lambda) \cdot \left(\Lambda_1^{(t)}\right)^{q-2}\right) \leq \Lambda_1^{(t)},
\end{aligned}
$$

which implies that $\Lambda_1^{(t)}$ will decrease. As a result, we can conclude that $\lambda_1^{(t)}$ will not exceed $\Theta(\log(1/\lambda))$, this completes the proof of the second part. $\qquad\square$

**Lemma B.12** (Lemma 5.7, restated). If $\eta \leq O(\sigma_0)$, it holds that $\Lambda_j^{(t)} = \widetilde{\Theta}(1)$ and $\Gamma_j^{(t)} = \widetilde{O}(\sigma_0)$ for all $t \in [T_{-1}, T]$.

*Proof.* We will prove the desired argument based on the following three induction hypothesis:

$$
\Lambda_j^{(t+1)} \geq (1 - \lambda\eta)\Lambda_j^{(t)} + \widetilde{\Theta}\left(\frac{\eta}{n}\right) \sum_{i:y_i=j} |\ell_{j,i}^{(t)}| - \widetilde{\Theta}(\alpha^q \eta) \cdot \frac{1}{n} \sum_{i=1}^n |\ell_{j,r}^{(t)}|, \tag{B.36}
$$

$$
\Gamma_j^{(t)} = \widetilde{O}(\sigma_0), \tag{B.37}
$$

$$\Lambda_j^{(t)} = \widetilde{\Theta}(1). \tag{B.38}$$

In terms of Hypothesis (B.36), we can apply Hypothesis (B.37) and (B.38) to (B.33) and get that

$$\Lambda_j^{(t+1)} \geq (1 - \eta\lambda)\Lambda_j^{(t)} + \Theta\left(\frac{\eta}{n}\right) \cdot \sum_{i:y_i=j} |\ell_{j,i}^{(t)}| \cdot \left(\Lambda_j^{(t)}\right)^{q-1} - \Theta(\alpha\eta) \cdot \left(\Gamma_j^{(t)} \vee \widetilde{\Theta}(\alpha)\right)^{q-1} \cdot \frac{1}{n} \sum_{i=1}^n |\ell_{j,r}^{(t)}|$$

$$\geq (1 - \lambda\eta)\Lambda_j^{(t)} + \widetilde{\Theta}\left(\frac{\eta}{n}\right) \sum_{i:y_i=j} |\ell_{j,i}^{(t)}| - \widetilde{\Theta}(\alpha^q\eta) \cdot \frac{1}{n} \sum_{i=1}^n |\ell_{j,r}^{(t)}|.$$

where the last inequality we use the fact that $\alpha \geq \sigma_0$. This verifies Hypothesis (B.36).

In order to verify Hypothesis (B.37), we have the following according to (B.36),

$$\sum_{j\in\{-1,1\}} \Lambda_j^{(t+1)} \geq (1-\lambda\eta) \sum_{j\in\{-1,1\}} \left[\Lambda_j^{(t)} + \widetilde{\Theta}\left(\frac{\eta}{n}\right) \sum_{i=1}^n |\ell_{j,i}^{(t)}| - \widetilde{\Theta}(\alpha^q\eta) \cdot \frac{1}{n} \sum_{i=1}^n |\ell_{j,r}^{(t)}|\right]$$

$$= (1-\lambda\eta) \sum_{j\in\{-1,1\}} \left[\Lambda_j^{(t)} + \widetilde{\Theta}\left(\frac{\eta}{n}\right) \sum_{i=1}^n |\ell_{j,i}^{(t)}|\right],$$

where the last equality holds since $\alpha = o(1)$. Recursively applying the above inequality from $T_{-1}$ to $t$ gives

$$\sum_{j\in\{-1,1\}} \Lambda_j^{(t)} \geq (1-\lambda\eta)^{t-T_{-1}} \sum_{j\in\{-1,1\}} \left[\Lambda_j^{(T_{-1})} + \widetilde{\Theta}\left(\frac{\eta}{n}\right) \cdot \sum_{\tau=0}^{t-T_{-1}-1} (1-\lambda\eta)^\tau \sum_{i=1}^n |\ell_{j,i}^{(t-1-\tau)}|\right].$$

Then by Hypothesis (B.38) we have

$$\widetilde{\Theta}\left(\frac{\eta}{n}\right) \cdot \sum_{\tau=0}^{t-T_{-1}-1} (1-\lambda\eta)^\tau \sum_{i=1}^n |\ell_{j,i}^{(t-1-\tau)}| \leq \widetilde{\Theta}(1).$$

Now let us look at the rate of memorizing noises. By (B.28) and use the fact that $\alpha^2 \leq O(s\sigma_p^2/n)$, we have

$$\Gamma_j^{(t)} \leq (1 - \eta\lambda)\Gamma_j^{(t-1)} + \widetilde{\Theta}\left(\frac{\eta s\sigma_p^2}{n}\right) \cdot \sum_{i=1}^n |\ell_{j,i}| \cdot \left(\Gamma_j^{(t-1)}\right)^{q-1}$$

$$\leq (1 - \eta\lambda)\Gamma_j^{(t-1)} + \widetilde{\Theta}\left(\frac{\eta s\sigma_p^2 \sigma_0^{q-1}}{n}\right) \cdot \sum_{i=1}^n |\ell_{j,i}|$$

$$\leq \Gamma_j^{(T_{-1})} + \widetilde{\Theta}\left(\frac{\eta s\sigma_p^2 \sigma_0^{q-1}}{n}\right) \cdot \sum_{\tau=0}^{t-T_{-1}-1} (1-\lambda\eta)^\tau \sum_{i=1}^n |\ell_{j,i}^{(t-1-\tau)}|$$

$$\leq \widetilde{\Theta}\left(\sigma_0 + s\sigma_p^2 \sigma_0^{q-1}\right)$$

$$\leq \widetilde{\Theta}(\sigma_0),$$

which verifies Hypothesis (B.37).

Given Hypothesis (B.36) and (B.37), the verification of (B.38) is straightforward by applying the same proof technique of Lemma B.11 and thus we omit it here. □

**Lemma B.13** (Lemma 5.8, restated). *If the step size satisfies, then for any $t \geq T_{-1}$ it holds that*

$$L(\mathbf{W}^{(t+1)}) - L(\mathbf{W}^{(t)}) \leq -\frac{\eta}{2}\|\nabla L(\mathbf{W}^{(t)})\|_F^2.$$

*Proof.* The proof of this lemma is similar to that of Lemma B.6, which is basically relying the smoothness property of the loss function $L(\mathbf{W})$ given certain constraints on the inner products $\langle \mathbf{w}_{j,r}, \mathbf{v} \rangle$ and $\langle \mathbf{w}_{j,r}, \boldsymbol{\xi}_i \rangle$.

Let $\Delta F_{j,i} = F_j(\mathbf{W}^{(t+1)}, \mathbf{x}_i) - F_j(\mathbf{W}^{(t)}, \mathbf{x}_i)$, we can get the following Taylor expansion on the loss function $L_i(\mathbf{W}^{(t+1)})$,

$$L_i(\mathbf{W}^{(t+1)}) - L_i(\mathbf{W}^{(t)}) \leq \sum_j \frac{\partial L_i(\mathbf{W}^{(t)})}{\partial F_j(\mathbf{W}^{(t)}, \mathbf{x}_i)} \cdot \Delta F_{j,i} + \sum_j (\Delta F_{j,i})^2. \tag{B.39}$$

In particular, by Lemma B.12, we know that $\langle \mathbf{w}_{j,r}^{(t)}, y_i \mathbf{v} \rangle \leq \widetilde{\Theta}(1)$ and $\langle \mathbf{w}_{j,r}^{(t)}, \boldsymbol{\xi}_i \rangle \leq \widetilde{\Theta}(\sigma_0) \leq \widetilde{\Theta}(1)$. Then similar to (B.21), we can apply first-order Taylor expansion to $F_j(\mathbf{W}^{(t+1)}, \mathbf{x}_i)$, which requires to characterize the second-order error of the Taylor expansions on $\sigma(\langle \mathbf{w}_{j,r}^{(t+1)}, y_i \mathbf{v} \rangle)$ and $\sigma(\langle \mathbf{w}_{j,r}^{(t+1)}, \boldsymbol{\xi}_i \rangle)$,

$$\left| \sigma(\langle \mathbf{w}_{j,r}^{(t+1)}, y_i \mathbf{v} \rangle) - \sigma(\langle \mathbf{w}_{j,r}^{(t)}, y_i \mathbf{v} \rangle) - \langle \nabla_{\mathbf{w}_{j,r}} \sigma(\langle \mathbf{w}_{j,r}^{(t)}, y_i \mathbf{v} \rangle), \mathbf{w}_{j,r}^{(t+1)} - \mathbf{w}_{j,r}^{(t)} \rangle \right|$$
$$\leq \widetilde{\Theta}\big( \|\mathbf{w}_{j,r}^{(t+1)} - \mathbf{w}_{j,r}^{(t)}\|_2^2 \big) = \widetilde{\Theta}(\eta^2 \|\nabla_{\mathbf{w}_{j,r}} L(\mathbf{W}^{(t)})\|_2^2),$$
$$\left| \sigma(\langle \mathbf{w}_{j,r}^{(t+1)}, \boldsymbol{\xi}_i \rangle) - \sigma(\langle \mathbf{w}_{j,r}^{(t)}, \boldsymbol{\xi}_i \rangle) - \langle \nabla_{\mathbf{w}_{j,r}} \sigma(\langle \mathbf{w}_{j,r}^{(t)}, \boldsymbol{\xi}_i \rangle), \mathbf{w}_{j,r}^{(t+1)} - \mathbf{w}_{j,r}^{(t)} \rangle \right|$$
$$\leq \widetilde{\Theta}\big( \|\mathbf{w}_{j,r}^{(t+1)} - \mathbf{w}_{j,r}^{(t)}\|_2^2 \big) = \widetilde{\Theta}(\eta^2 \|\nabla_{\mathbf{w}_{j,r}} L(\mathbf{W}^{(t)})\|_2^2). \tag{B.40}$$

Then combining the above bounds for every $r \in [m]$, we can get the following bound for $\Delta F_{j,i}$

$$\left| \Delta F_{j,i} - \langle \nabla_{\mathbf{W}} F_j(\mathbf{W}^{(t)}, \mathbf{x}_i), \mathbf{W}^{(t+1)} - \mathbf{W}^{(t)} \rangle \right| \leq \widetilde{\Theta}\bigg( \eta^2 \sum_{r \in [m]} \|\nabla_{\mathbf{w}_{j,r}} L(\mathbf{W}^{(t)})\|_2^2 \bigg)$$
$$= \widetilde{\Theta}\big( \eta^2 \|\nabla L(\mathbf{W}^{(t)})\|_F^2 \big). \tag{B.41}$$

Moreover, since $\langle \mathbf{w}_{j,r}^{(t)}, y_i \mathbf{v} \rangle \leq \widetilde{\Theta}(1)$ and $\langle \mathbf{w}_{j,r}^{(t)}, \boldsymbol{\xi}_i \rangle \leq \widetilde{\Theta}(1)$ and $\sigma(\cdot)$ is convex, then we have

$$|\sigma(\langle \mathbf{w}_{j,r}^{(t+1)}, y_i \mathbf{v} \rangle) - \sigma(\langle \mathbf{w}_{j,r}^{(t)}, y_i \mathbf{v} \rangle)| \leq \max\big\{ |\sigma'(\langle \mathbf{w}_{j,r}^{(t+1)}, y_i \mathbf{v} \rangle)|, |\sigma'(\langle \mathbf{w}_{j,r}^{(t)}, y_i \mathbf{v} \rangle)| \big\} \cdot |\langle \mathbf{v}, \mathbf{w}_{j,r}^{(t+1)} - \mathbf{w}_{j,r}^{(t)} \rangle|$$
$$\leq \widetilde{\Theta}\big( \|\mathbf{w}_{j,r}^{(t+1)} - \mathbf{w}_{j,r}^{(t)}\|_2 \big).$$

Similarly we also have

$$|\sigma(\langle \mathbf{w}_{j,r}^{(t+1)}, \boldsymbol{\xi}_i \rangle) - \sigma(\langle \mathbf{w}_{j,r}^{(t)}, \boldsymbol{\xi}_i \rangle)| \leq \widetilde{\Theta}\big( \|\mathbf{w}_{j,r}^{(t+1)} - \mathbf{w}_{j,r}^{(t)}\|_2 \big).$$

Combining the above inequalities for every $r \in [m]$, we have

$$\left| \Delta F_{j,i} \right|^2 \leq \widetilde{\Theta}\bigg( \bigg[ \sum_{r \in [m]} \|\mathbf{w}_{j,r}^{(t+1)} - \mathbf{w}_{j,r}^{(t)}\|_2 \bigg]^2 \bigg) \leq \widetilde{\Theta}\big( m\eta^2 \|\nabla L(\mathbf{W}^{(t)})\|_F^2 \big) = \widetilde{\Theta}\big( \eta^2 \|\nabla L(\mathbf{W}^{(t)})\|_F^2 \big). \tag{B.42}$$

Now we can plug (B.41) and (B.42) into (B.39), which gives

$$L_i(\mathbf{W}^{(t+1)}) - L_i(\mathbf{W}^{(t)}) \leq \sum_j \frac{\partial L_i(\mathbf{W}^{(t)})}{\partial F_j(\mathbf{W}^{(t)}, \mathbf{x}_i)} \cdot \Delta F_{j,i} + \sum_j (\Delta F_{j,i})^2$$
$$= \langle \nabla L_i(\mathbf{W}^{(t)}), \mathbf{W}^{(t+1)} - \mathbf{W}^{(t)} \rangle + \widetilde{\Theta}(\eta^2 \|\nabla L(\mathbf{W}^{(t)})\|_F^2). \tag{B.43}$$

Taking sum over $i \in [n]$ and applying the smoothness property of the regularization function $\lambda \|\mathbf{W}\|_F^2$, we can get

$$L(\mathbf{W}^{(t+1)}) - L(\mathbf{W}^{(t)}) = \frac{1}{n} \sum_{i=1}^{n} \big[ L_i(\mathbf{W}^{(t+1)}) - L_i(\mathbf{W}^{(t)}) \big] + \lambda \big( \|\mathbf{W}^{(t+1)}\|_F^2 - \|\mathbf{W}^{(t)}\|_F^2 \big)$$
$$\leq \langle \nabla L(\mathbf{W}^{(t)}), \mathbf{W}^{(t+1)} - \mathbf{W}^{(t)} \rangle + \widetilde{\Theta}(\eta^2 \|\nabla L(\mathbf{W}^{(t)})\|_F^2)$$
$$= -\big( \eta - \widetilde{\Theta}(\eta^2) \big) \cdot \|\nabla L(\mathbf{W}^{(t)})\|_F^2$$
$$\leq -\frac{\eta}{2} \|\nabla L(\mathbf{W}^{(t)})\|_F^2,$$

where the last inequality is due to our choice of step size $\eta = o(1)$ so that gives $\eta - \widetilde{\Theta}(\eta^2) \geq \eta/2$. This completes the proof. □

**Lemma B.14** (Generalization Performance of GD). Let

$$\mathbf{W}^* = \arg\min_{\{\mathbf{W}^{(1)},\ldots,\mathbf{W}^{(T)}\}} \|\nabla L(\mathbf{W}^{(t)})\|_F.$$

Then for all training data, we have

$$\frac{1}{n}\sum_{i=1}^n \mathbb{1}\left[F_{y_i}(\mathbf{W}^*, \mathbf{x}_i) \le F_{-y_i}(\mathbf{W}^*, \mathbf{x}_i)\right] = 0.$$

Moreover, in terms of the test data $(\mathbf{x}, y) \sim \mathcal{D}$, we have

$$\mathbb{P}_{(\mathbf{x},y)\sim\mathcal{D}}\left[F_y(\mathbf{W}^*, \mathbf{x}) \le F_{-y}(\mathbf{W}^*, \mathbf{x})\right] = o(1).$$

*Proof.* By Lemma B.12 it is clear that all training data can be correctly classified so that the training error is zero. Besides, for test data $(\mathbf{x}, y)$ with $\mathbf{x} = [y\mathbf{v}^\top, \boldsymbol{\xi}^\top]^\top$, it is clear that with high probability $\langle \mathbf{w}_{y,r}^*, y\mathbf{v} \rangle = \widetilde{\Theta}(1)$ and $[\langle \mathbf{w}_{y,r}^*, \boldsymbol{\xi} \rangle]_+ \le \widetilde{O}(\sigma_0)$, then

$$F_y(\mathbf{W}^*, \mathbf{x}) = \sum_{r=1}^m \left[\sigma(\langle \mathbf{w}_{y,r}^*, y\mathbf{v} \rangle) + \sigma(\langle \mathbf{w}_{y,r}^*, \boldsymbol{\xi} \rangle)\right] \ge \widetilde{\Omega}(1).$$

If $j = -y$, we have with probability at least $1 - 1/\text{poly}(n)$, $\langle \mathbf{w}_{-y,r}^*, y\mathbf{v} \rangle \le 0$ and $[\mathbf{w}_{-y,r}^*, \boldsymbol{\xi} \rangle]_+ \le \widetilde{O}(\alpha)$, which leads to

$$F_{-y}(\mathbf{W}^*, \mathbf{x}) = \sum_{r=1}^m \left[\sigma(\langle \mathbf{w}_{-y,r}^*, y\mathbf{v} \rangle) + \sigma(\langle \mathbf{w}_{-y,r}^*, \boldsymbol{\xi} \rangle)\right] \le \widetilde{O}(m\alpha^q) = \widetilde{O}(\alpha^q) = o(1).$$

This implies that GD can also achieve nearly at most $1/\text{poly}(n)$ test error. This completes the proof. $\qquad\square$

## C  PROOF OF THEOREM 4.2: CONVEX CASE

**Theorem C.1** (Convex setting, restated). Assume the model is over-parameterized. Then for any convex and smooth training objective with positive regularization parameter $\lambda$, suppose we run **Adam** and **gradient descent** for $T = \frac{\text{poly}(n)}{\eta}$ iterations, then with probability at least $1 - n^{-1}$, the obtained parameters $\mathbf{W}_{\text{Adam}}^*$ and $\mathbf{W}_{\text{GD}}^*$ satisfy that $\|\nabla L(\mathbf{W}_{\text{Adam}}^*)\|_1 \le \frac{1}{T\eta}$ and $\|\nabla L(\mathbf{W}_{\text{Adam}}^*)\|_2^2 \le \frac{1}{T\eta}$ respectively. Moreover, it holds that:

- Training errors are both zero:

$$\frac{1}{n}\sum_{i=1}^n \mathbb{1}\left[\text{sgn}\big(F(\mathbf{W}_{\text{Adam}}^*, \mathbf{x}_i)\big) \ne y_i\right] = \frac{1}{n}\sum_{i=1}^n \mathbb{1}\left[\text{sgn}\big(F(\mathbf{W}_{\text{GD}}^*, \mathbf{x}_i)\big) \ne y_i\right] = 0.$$

- Test errors are nearly the same:

$$\mathbb{P}_{(\mathbf{x},y)\sim\mathcal{D}}\left[\text{sgn}\big(F(\mathbf{W}_{\text{Adam}}^*, \mathbf{x}_i)\big) \ne y\right] = \mathbb{P}_{(\mathbf{x},y)\sim\mathcal{D}}\left[\text{sgn}\big(F(\mathbf{W}_{\text{GD}}^*, \mathbf{x})\big) \ne y\right] \pm o(1).$$

*Proof.* The proof is straightforward by applying the same proof technique used for Lemmas B.6 and B.13, where we only need to use the smoothness property of the loss function. Then it is clear that both Adam and GD can provably find a point with sufficiently small gradient. Note that the training objective becomes strongly convex when adding weight decay regularization, implying that the entire training objective only has one stationary point, i.e., point with sufficiently small gradient. This further imply that the points found by Adam and GD must be exactly same and thus GD and Adam must have nearly same training and test performance.

Besides, note that the problem is also sufficiently over-parameterized, thus with proper regularization (feasibly small), we can still guarantee zero training errors. $\qquad\square$

