# OpenReview forum: "Understanding the Generalization of Adam in Learning Neural Networks with Proper Regularization"
_ICLR.cc/2022/Conference — ICLR 2022 Submitted_

### Official Review · Reviewer_yf6t · 2021-11-02

**Correctness:** 4
**Technical Novelty And Significance:** 4
**Empirical Novelty And Significance:** Not applicable
**Recommendation:** 5
**Confidence:** 4

**Details Of Ethics Concerns:**

no concerns

**Main Review:**

Strength:
- experimental verification helps believe the result is true, as the proof is complex.
- a simplified version of the result is presented assuming SignGD is used instead of Adam and can be intuitively understood.
- food for thought paper on the limitations of Adam even if on a very specific example.

Weaknesses:
- not really a two layer network because the second layer is not learnable. If it were learnable, then setting a negative weight on the second patch would remove the issue that the dimension with the answer has different signs in the two patches.
- The behavior only exists because q>1 (amplification of the gradient from the dot product with the noise in the second patch). Otherwise, the gradient from the first patch for the coordinate 1 would always win.
- Behavior only exist for d > n^4.
- The level of sparsity is quite strong ($\sqrt{d} / n^2$), which seems a bit unlikely for a neural network (the citation (Papyan et al. 2017) does not just really justify such as assumption).
- The result is for batch GD and Adam. Notice that for the stochastic version, we wouldn't have such a strong difference between GD and Adam, because most of the time the gradient would be zero for the noise coordinates, and when they are seen, Adam would rescale by 1/sqrt(n) rather than 1/n with the batch version.


Questions:
- Each noise coordinate is present only once. So its gradient will be of the order of 1/n. Adam will rescale the gradient by a factor of n. What if you would do GD but with a learning rate n times bigger for the sparse features ?

Minor remarks:
- Lemma numbering is not aligned between main paper and appendix, while the Lemmas are the same. Either use same numbering or mention the Lemma number in the appendix with the proof.
- mention that you assumme d>>n^4, this is not expicitely the case.
- definition of Adam is missing the initialization bias correction, mention you drop it for simplicity.

update: after discussing with the authors on the activation function, I believe the authors reply on the smooth ReLU, while technically correct, is unrealistic. In particular, the authors rely on the scaling down of the pre-activation outputs to a region where the smooth ReLU is polynomial, in order to obtain again the amplification that allows the bit flipping. Neural network initialization is specifically made so as to avoid such small pre-activation outputs, and I believe this makes this theoretical example too far off current practices to be considered an illustration of the generalization issues faced by Adam.

**Summary Of The Paper:**

This paper propose a non convex optimization problem where the batch version of Adam has worse generalization than GD. Both methods achieve 0 training loss asymptotically but GD has a zero loss on the test set, while Adam has a large loss.

The paper studies a simplified CNN model with only two patches. The two layers are multiplied with the same weight matrix, fed into the activation function and then summed. A cross entropy loss is used for the classification. The activation function must be of the form $\max(0, x^q)$ for $q \geq 3$. Note that the second layer weights are not learnable and fixed to 1.

The dataset is built as follow: the label is y in {-1, 1}. One of the patch is just $x_1 = [y, 0, 0, ...]$. The second patch is $x_2 = [-\alpha y, \text{sparse gaussian noise}]$ ($0 <\alpha < 1$). The sparsity level is so high that no two examples have the same features.

The idea is that Adam will use a much larger effective learning rate for the sparse noise features than SGD, and grow the coordinates quickly. At some point the dot product with the noise in $x_2$ will be larger than the dot product with $x_1$. Because the activation function is of the form $x^q$ with $q \geq3$, this means the gradient from $x_2$ will overtake the one from $x_1$ and the weight for the first dimension will be negative (while a positive weight is the best thing to classify from $x_1$).
So Adam will focus on the weaker signal from $x_2$ (remember $\alpha < 1$).

To summarize further, one patch has the right answer, the second path has the right answer (but with opposite sign) + noise. Adam will focus on the second patch which has lower signal, while GD will focus on the first patch and ignore the noise.

The authors provide experimental verification of this fact.


**Summary Of The Review:**

Interesting example where Adam generalizes worse than GD. The assumptions are very far from a real network, and removing any of them make this example incorrect: second layer is not learnable, very high level of sparsity, and activation function has an amplification behavior.

As far as I can say, claims are correct but the proof is quite hard to verify.

To summarize, I'm recommending weak acceptance, because the example is interesting but the assumptions are unrealistic.

---

> ### Author Response · Authors · 2021-11-18
> **Response to Reviewer yf6t (2/2)**
>
> ---
> **Q:** "Each noise coordinate is present only once. So its gradient will be of the order of 1/n. Adam will rescale the gradient by a factor of n. What if you would do GD but with a learning rate n times bigger for the sparse features?"
>
> **A:** First, as we have mentioned previously, it would be difficult to only apply a large learning rate for the feature patch since we do not know which patch contains features for all training data. Second, we have already shown that GD, with a standard learning rate, can well learn the feature rather than the noise, so it is evident that increasing the learning rate for the feature patch will further help learn the feature.
>
> ---
>
> **Q:** Minor remarks
>
> **A:** We will fix those issues accordingly.

---

> ### Author Response · Authors · 2021-11-18
> **Response to Reviewer yf6t (1/2)**
>
> ---
>
> Thank you for your positive and valuable comments.
>
> **Q:** "not really a two layer network because the second layer is not learnable. "
>
> **A:** Thanks for pointing this out. We would like to clarify that our results can be directly extended to the case where the second layer is learnable and the results will remain similar. The reason behind this is that the difference between Adam and GD comes from the learning of the hidden layer, while the second layer weights will not affect which patch is more likely to be learned by different optimization algorithms. Consequently, we can still get the result that GD tends to learn the feature patch while Adam tends to learn the noise patch.
>
> ---
>
> **Q:** "If it were learnable, then setting a negative weight on the second patch would remove the issue that the dimension with the answer has different signs in the two patches."
>
> **A:** That’s a great point, it is true that if we can assign negative weight on the noise patch, then the model will always learn the right direction. However, we cannot do so since the learner does not know which of the two data patches is the feature patch or the noise patch (we have stated this in Definition 3.1 and will highlight this in the description of CNN model). For instance, for one data point, the first patch could be a feature patch while for the other data point the first patch could be a noise patch. Our understanding is that if the second layer is learnable, the algorithm may prefer to pick some neurons with nice initial weights (for the second layer) since that neuron could produce large gradients.
>
> ---
> **Q:** "The behavior only exists because q>1 (amplification of the gradient from the dot product with the noise in the second patch). Otherwise, the gradient from the first patch for the coordinate 1 would always win."
>
> **A:** It is true that our analysis requires that $q>1$ (at least in the range [0, 1]) so that the feature learning in adam can be flipped.
> Our analysis also applies when the activation function is a smoothed-version of ReLU function (i.e., the activation function used in [Allen-Zhu and Li, 2020c], which behaves as a degree-$q$ polynomial in the range [0, O(1/polylog(n))] and is linear afterward.). If we assume the underlying input data distribution is Gaussian, then we can also deal with ReLU activation function (p=1) (See e.g., [Li et al., 2020]).
>
> ---
> **Q:** Conditions on the dimension and sparsity
>
> **A:** We agree that the conditions on the dimension and sparsity in our paper are not exactly aligned with the practice and they are made for the ease of theoretical analysis. We believe these conditions can be potentially weakened by using a sharper analysis on feature learning and noise memorization. The main goal of this paper is to argue that the nonconvex landscape is critical in NN analysis by demonstrating the separation between Adam and GD (with proper regularization) on some nontrivial distributions, which cannot be explained by the existing neural tangent kernel (NTK)-based analysis (which are built based on some local convexity property) or other analyses. This work is the first of its kind and improving the conditions on the problem is left as our future work. (Think about the first work on NTK, which relies on quite strong conditions on the neural network width. Since then, many follow up works have improved the conditions significantly)
>
> ---
> **Q:** "The result is for batch GD and Adam. Notice that for the stochastic version, we wouldn't have such a strong difference between GD and Adam, because most of the time the gradient would be zero for the noise coordinates, and when they are seen, Adam would rescale by 1/sqrt(n) rather than 1/n with the batch version."
>
> **A:** Thanks for pointing this out! It is true when using mini-batch stochastic gradients, Adam would be rescaled while GD would not. However, we would like to clarify that the difference between SGD and mini-batch Adam still exists (though the difference might be smaller). This is because, in our paper, the separation between Adam and GD is characterized by $\mathrm{poly}(d)$ factors: the speed of feature learning in GD and Adam, and the speed of noise memorization in GD are both in the order of $O(1)$ (in each step), while the speed of noise memorization in Adam is proportional to the number of nonzero entries, which is in the order of $\mathrm{poly}(d)$. In contrast, using mini-batch stochastic gradients can only reduce the difference by at most $O(n)$ factors, which is dominated by the $\mathrm{poly}(d)$ separation as long as the dimension $d$ is sufficiently large (overparameterized).

---

> > ### Comment · Reviewer_yf6t · 2021-11-24
> > **polynomial ReLU not needed**
> >
> > Thank you for your reply,
> > I am not sure about your reply on the polynomial ReLU. You do not give any strong arguments to back your claim. It still seems to me that without the amplification from the activation function you will not be able to replicate your example.
> >
> > Smoothed ReLU is only going to have a tiny portion where the slope of the function is not 0/1, and their seem to be no reason that the patch with only the evidence vector would be in that portion. If both the noisy and evidence patch are in the part with a slope of 1, then the evidence patch will always win.

---

> > > ### Author Response · Authors · 2021-11-25
> > > **Re: polynomial ReLU not needed**
> > >
> > > ---
> > > Thank you for your further questions! We answer them as follows.
> > >
> > > **Q:** I am not sure about your reply on the polynomial ReLU. You do not give any strong arguments to back your claim. It still seems to me that without the amplification from the activation function you will not be able to replicate your example.
> > >
> > > **A:** We are sorry if we did not make it clear. We mentioned in our previous reply that $q>1$ is required to get the theoretical results. You are right that the amplification of the polynomial ReLU is important and necessary, at least for the proof of Adam.
> > >
> > > ---
> > > **Q:** Smoothed ReLU is only going to have a tiny portion where the slope of the function is not 0/1, and their seem to be no reason that the patch with only the evidence vector would be in that portion. If both the noisy and evidence patch are in the part with a slope of 1, then the evidence patch will always win.
> > >
> > > **A:** For smoothed ReLU, we agree that if both noisy and evidence patches are in the region with slope=1, then the evidence/feature patch will always win. However, this is not the case for Adam, where only the noisy patch can reach the region with slope=1, while the feature patch cannot. The reason is as follows:
> > > First, in order to remove possible confusion, we remark that we consider the smoothed ReLU activation in the following form:
> > > \begin{align}
> > > \sigma(x) = 0 \mbox{ if } x < 0;
> > > \end{align}
> > > $$
> > > \sigma(x) = \frac{x^q}{q\rho^{q-1}}  \mbox{ if }  x \in[0, \rho];
> > > $$
> > > $$
> > >  \sigma(x) = x - (1-1/q) \rho  \mbox{ if } x > \rho
> > > $$
> > > where $\rho = 1/\mathrm{polylog}(n)$.
> > >
> > > * According to the initialization for the model weights in our paper, the preactivation of both feature patch and noise patch will be in the order of $\tilde O(\sigma_0) = o(1)$, which lies in the “polynomial region” (i.e., $(0, 1/\mathrm{polylog}(n)]$). So the initial training behavior will be almost the same as that of using the polynomial ReLU activation function.
> > >
> > > * What happens next is that after a certain number of iterations, due to the amplification of the activation function in such a “polynomial region” and the feature noise ($-\alpha \mathbf{v}$) in the noise patch, the feature learning will be flipped (See Lemma 5.3 and its proof on page 19-20) and the inner product between the weight and the feature vector will finally be negative. Note that now all the preactivations still remain in the “polynomial region”.
> > >
> > > * In the remaining training stages, the preactivation of the noise patch will reach the “linear region” (the region with slope 1), while the preactivation of the feature patch still remains in the “polynomial region” or even negative.  This will continue to the convergence where the converging point has $\tilde O(1)$ preactivation for noise patch and $-o(1)$ preactivation for the feature patch, so the claim in our paper still holds.

---

> > > > ### Comment · Reviewer_yf6t · 2021-11-26
> > > > **reply**
> > > >
> > > > Thank you for your reply, I see the reasoning, although it is made artificial by the fact that real neural network activations at initialization are closer to O(1) than o(1). Basically you are scaling down the outputs sufficiently so that the "smoothed" part of the activation becomes the entire activation function for the for seeable future, and so you recover the polynomial activation.

---

> > > > > ### Author Response · Authors · 2021-11-27
> > > > > **Response to the concerns about the polynomial activation**
> > > > >
> > > > > We thanks the reviewer for expressing the concerns, we would like to clarify:
> > > > >
> > > > > (1). The reviewer might have misunderstood our previous reply.
> > > > >
> > > > > "Basically you are scaling down the outputs sufficiently so that the "smoothed" part of the activation becomes the entire activation function for the for seeable future"
> > > > >
> > > > > As we were explaining, **our proof technique and result generalizes** to the case of a  " smoothed ReLU where the smoothing region is o(1) small", as long as the initialization satisfies that the activation is initially in the smoothed region. We were pointing out that the noise and the features can indeed grow to the linear region of the smoothed ReLU, but our result can still be proved.
> > > > >
> > > > > Moreover, in practice, smoothed ReLU such as SoftPlus or GELU do have a non-linear regime of $\Omega(1)$ large. To keep result simple, we choose to represent the simplified version of smoothed ReLU with only the non-linear regime, but our result indeed extends to the case where the activation can reach linear regime during the training.
> > > > >
> > > > >
> > > > >
> > > > >
> > > > > (2). **The choice of polynomial activation is very conventional in theory of deep learning**, as a proxy for studying the behavior of neural networks with ReLU activations, see [1], [2], [3] etc. and its also known empirically that polynomial activation works comparably as ReLU activation in real neural networks [4].
> > > > >
> > > > > [1] Li, Y., et al. Algorithmic regularization in over-parameterized matrix sensing and neural networks with quadratic activations. 2018.
> > > > >
> > > > > [2] Woodworth, B., et al. Kernel and rich regimes in overparametrized models. 2020.
> > > > >
> > > > > [3] Haochen, J., Shape Matters: Understanding the Implicit Bias of the Noise Covariance
> > > > >
> > > > > [4] Allen-Zhu, Z., et al. Backward feature correction: How deep learning performs deep learning. 2020.

---

### Official Review · Reviewer_ahKf · 2021-11-02

**Correctness:** 2
**Technical Novelty And Significance:** 3
**Empirical Novelty And Significance:** 2
**Recommendation:** 3
**Confidence:** 4

**Main Review:**

Using artificial datasets to simplify the analysis of complex optimization dynamics seems to be a highly promising approach. It has the potential to provide a theoretical yet intuitive understanding of the interactions between data, models, and optimizers. While I believe this paper has made some progress in this direction, many of its assumptions are too contrived.

Firstly, the same dimensions of different patches of the constructed data are reserved for either feature or feature noise, and the norm of the noise vector is specifically chosen to be much smaller than that of the feature vector, both of which are not well motivated. It would be interesting to see if some of the results still hold with weaker assumptions. Secondly, it seems the convolution part of the two-layer model is only introduced to impose weight sharing, such that the feature patch and the noise patch have to compete for the same set of weights, which is not necessarily the case in practice. Furthermore, the choice of truncated polynomial activation function is unconventional, and the second layer of the model simply sums up all the activations of the first layer without learned weights. Thirdly, in practice, the generalization gap is mostly observed between Adam and SGD rather than GD, and it is well known that SGD usually generalizes better than GD; therefore making directly comparisons between Adam and SGD would be more convincing.

**Summary Of The Paper:**

This paper aims to explain why there is a generalization gap between Adam and gradient descent in learning neural networks, even with proper regularization. To this end, the authors construct an artificial dataset in attempt to capture the basic properties of real image datasets that can lead to the generalization gap. Based on this dataset, the authors prove that Adam is more likely to fit noise in data, while GD tends to fit real features. The theoretical results are verified by experiments on the proposed dataset.

**Summary Of The Review:**

This paper presents a promising approach to explaining the generalization gap between Adam and gradient descent, but many of the assumptions on its data, model, and optimizers are far from realistic.

---

> ### Author Response · Authors · 2021-11-18
> **Response to Reviewer ahKf**
>
> ---
> Thank you for your constructive comments.
>
> **Q:** "Firstly, the same dimensions of different patches of the constructed data are reserved for either feature or feature noise, and the norm of the noise vector is specifically chosen to be much smaller than that of the feature vector, both of which are not well motivated."
>
> **A:** We would like to point out that there is a misunderstanding regarding our data distribution: (1) we indeed decompose the data into feature patch and noise patch, but the noise patch contains both feature noise and random noise; (2) the norm of noise vector is not much smaller than feature vector (only the feature noise is smaller otherwise the feature noise will become the feature).  In addition, within each layer of CNNs, the filter size is typically the same, and this is why we assume different patches have the same dimension.
>
> ---
>
> **Q:**  "it seems the convolution part of the two-layer model is only introduced to impose weight sharing, such that the feature patch and the noise patch have to compete for the same set of weights, which is not necessarily the case in practice"
>
> **A:** Weight sharing is standard in CNN since the same filter will be operated on different patches. Our theoretical analysis can also handle the case of using different weights for feature and noise patches, and the analysis is even simpler since we no longer need to take care of the interaction between feature learning and noise memorization. So weight sharing is actually a strength rather than a weakness of our analysis.
>
> ---
>
> **Q:** "the choice of truncated polynomial activation function is unconventional, and the second layer of the model simply sums up all the activations of the first layer without learned weights."
>
> **A:** The use of the polynomial ReLU activation function is for simplifying our analysis. It can be replaced by a smoothed ReLU activation function (e.g., the activation function used in [Allen-Zhu and Li, 2020c]). If we assume the underlying input data distribution is Gaussian, we can also deal with ReLU activation function [Li et al., 2020].
>
> The choice of polynomial activation is **conventional** in theory of deep learning, see [1], [2], [3] etc. and its also known empirically that polynomial activation works comparably as ReLU activation in real neural networks [4].
>
> [1] Li, Y., et al. Algorithmic regularization in over-parameterized matrix sensing and neural networks with quadratic activations. 2018.
>
> [2] Woodworth, B., et al. Kernel and rich regimes in overparametrized models. 2020.
>
> [3] Haochen, J., Shape Matters: Understanding the Implicit Bias of the Noise Covariance
>
> [4] Allen-Zhu, Z., et al. Backward feature correction: How deep learning performs deep learning. 2020.
>
> We would also like to emphasize that adding learnable weights in the second layer will not affect our result. In fact, the separation between Adam and GD is due to their different learning behaviors on the noise and features, which happen in the hidden layer. In our paper, we consider summing up activations for ease of analysis.  We will add the discussion about adding a learnable second layer in the revision.
>
>
> ---
>
> **Q:** "the generalization gap is mostly observed between Adam and SGD rather than GD, and it is well known that SGD usually generalizes better than GD; therefore making directly comparisons between Adam and SGD would be more convincing."
>
> **A:** We would like to first emphasize that both Adam and GD considered in our paper are using full-batch gradients so that the comparison is fair. Second, our results show that the separation between Adam and GD is mainly due to their different usage of gradients that lead to different feature-noise learning patterns, which can be extended to stochastic gradients. In fact, using stochastic gradients will not affect feature learning since the feature vector is assumed to be contained in all training data. For SGD, the noise memorization will be largely similar to that of GD since the gradients (full-batch ones or mini-batch ones) are always the linear combination of data vectors and the learning process of noise vectors will be similar. For mini-batch Adam, the update in each iteration can be approximated by signed stochastic gradients and we can also show that the algorithm tends to memorize the noise patch rather than the feature patch since the noise vector is much denser. We will add this discussion in the revision.

---

> > ### Comment · Reviewer_ahKf · 2021-11-27
> > **Assumptions are not well-motivated**
> >
> > Thanks for the response.
> >
> > I understand the difference between feature noise and random noise. What I don't understand is why the variance of random noise has to decrease as the number of samples (n) increases. More importantly, there is no particular reason that the spatial distribution of feature and feature noise should be perfectly aligned with the patch division as assumed in the paper. Additionally, this issue is made worse by the fixed second-layer weights. If the weights were learnable, the weights of \sigma(w,x_1) and \sigma(w,x_2) would likely have different signs due to the feature noise, contrary to the assumption.
> >
> > Therefore, I'm not convinced that the assumptions mentioned above are well-motivated. I suggest using weaker assumptions if possible, as long as some generalization gap between Adam and GD (or SGD) can be proved, which should be easier and more practically relevant than making Adam completely fail to generalize.

---

> > > ### Author Response · Authors · 2021-11-28
> > > **Thanks for your further comment**
> > >
> > > Thank you for your reply. We answer your additional comments and questions as follows.
> > >
> > > **Q:** I  understand the difference between feature noise and random noise. What I don't understand is why the variance of random noise has to decrease as the number of samples (n) increases.
> > >
> > > **A:** The scaling of the variance of the noise in the sample size $n$ is not essential. Due to the close relation between the dimension $d$ and sample size $n$, we can translate the dependence of the noise variance from $n$ to $d$. Then as long as $d$ is fixed and $d>Cn^4$ for some constant $C$, the noise variance won’t change (e.g., decrease) as $n$ increases.
> > >
> > > ---
> > > **Q:** More importantly, there is no particular reason that the spatial distribution of feature and feature noise should be perfectly aligned with the patch division as assumed in the paper.
> > >
> > > **A:** The perfect alignment of feature and feature noise with the patch division is not essential. If a patch contains both feature and feature noise (or random noise), then one of them will be the dominating factor that drives the learning of CNNs and our theoretical analysis can still be applied. In other words, our current analysis can be extended to deal with non-perfectly aligned features and feature noise. The perfect alignment assumption is solely introduced to simplify the analysis.
> > >
> > > ---
> > >
> > > **Q:** Additionally, this issue is made worse by the fixed second-layer weights. If the weights were learnable, the weights of $\sigma(w,x_1)$ and $\sigma(w,x_2)$ would likely have different signs due to the feature noise, contrary to the assumption.
> > >
> > > **A:** We think the reviewer might misunderstand the setting. Note that the index/ordering of the feature patch or noise patch is not fixed. For instance, for one data point, the first patch could be the feature patch and the second one is the noise patch,  while for the other data point the first patch could be a noise patch and the second one is the feature patch (See Definition 3.1). Therefore, we do not see why the weights of $\sigma(w,x_1)$ and $\sigma(w,x_2)$ would likely have different signs since both $x_1$ and $x_2$ can be feature/noise patches with equal probability. Also, as we have explained in our response, our result can be extended to deal with learnable second-layer weights.
> > >
> > > ---
> > >
> > > **Q:** Therefore, I'm not convinced that the assumptions mentioned above are well-motivated. I suggest using weaker assumptions if possible, as long as some generalization gap between Adam and GD (or SGD) can be proved, which should be easier and more practically relevant than making Adam completely fail to generalize.
> > >
> > > **A:** We believe our assumptions are not strong. In fact, compared with many deep learning theory papers that assume the networks are linear or deep linear, we considered two-layer CNNs, which are more realistic and substantially closer to the practice of deep learning.
> > >
> > > Compared with NTK-type analysis, our analysis does not require ADAM/GD to stay close to the initialization, and can clearly demonstrate the feature learning and noise memorization mechanism during deep learning. Our analysis can for sure be generalized to settings with weaker assumptions, but that may sacrifice the readability of the papers.
> > >
> > > In addition, compared with the seminal works (Wilson et al., 2017, Reddi et al., 2018) that study the different behaviors between Adam and GD by considering some handcrafted optimization problems, our setting is more general and our assumptions are definitely no stronger than those made in these papers.

---

### Official Review · Reviewer_9CZz · 2021-11-03

**Correctness:** 3
**Technical Novelty And Significance:** 3
**Empirical Novelty And Significance:** 3
**Recommendation:** 6
**Confidence:** 3

**Main Review:**

Strengths:
1. The proposed analysis of the comparison between GD and Adam is, as far as I am concerned, novel.
2. The results look interesting and promising, the dataset proposed by the authors is interesting and could be useful in other analyses as well.


Weakness:
Although I am not an expert on the NN analysis literature, I happen to be familiar with some of the papers the authors have mentioned, especially those related to NTK. I have to admit that the settings in this paper are so novel that it looks a bit strange to me. Although I understand that it is hard to analyze neural networks, especially when the authors are trying to compare GD with Adam, which is notoriously known to be unnecessarily fancy, I still have the following questions/concerns.

1. First of all, the dataset looks interesting to me, but it seems that it is particularly designed for Adam on purpose. That is to say, it is already known to the public that Adam has convergence issues (Reddi et al, 2018), then it may not be too hard to construct such a dataset where Adam performs badly. Also, such a specific construction process raises the question of whether it is also possible to construct a case where GD performs pretty badly but Adam performs fairly well. I would sincerely encourage the authors to give more intuitions about their dataset, and maybe relax the assumptions (at least for GD)

2. Second, the neural network looks a little strange to me. The authors are using a truncated polynomial activation function, instead of ReLU. Is this on purpose? I don't see such kinds of activation functions very often in the literature. Is it possible to replace it with, say, sigmoid, tanh, or maybe ReLU so that the analysis is more similar to the real training process? I do not see where the technical difficulty is. It would be helpful if the authors can explain it a little.

3. If I understand it correctly, all the algorithms in this paper use full-batch gradients as their updates, instead of mini-batch gradients. I wonder whether it is possible to generalize the results to the stochastic mini-batch version.


Sashank J Reddi, Satyen Kale, and Sanjiv Kumar. On the convergence of adam and beyond. In International Conference on Learning Representations, 2018.



**Summary Of The Paper:**

This paper proposes a new perspective to explain the difference in generalization ability between Adam and SGD. The main contribution of this paper is that it analyzes the behavior of Adam and SGD training on a two-layer network and a very special dataset, and the authors claim that Adam generalizes poorly while SGD generalizes well. The authors also provide analysis for convex functions and some experimental results to validate their claims.

**Summary Of The Review:**

Although I find this paper to be novel and interesting, I have to admit that some of the settings in the paper look strange to me. However, I would like to encourage such explorations and vote for acceptance because the authors are not using NTK, and I could imagine how difficult it is when trying to analyze Adam on neural networks. However, I would also like to encourage the authors to further generalize and improve their analysis.

---

> ### Author Response · Authors · 2021-11-18
> **Response to Review 9CZz**
>
>
> ---
> Thank you for your positive comments.
>
> **Q:** Comparison with Reddi et al. 2018
>
> **A:** Thanks for pointing out this paper, we will add a discussion on this in the revision. We would also like to clarify that our focus is different from theirs: (1) in our setting, both Adam and SGD can attain global convergence, i.e., finding solutions that lead to zero training error, so there is no convergence issue for either SGD or Adam; (2) we prove the separation between SGD and Adam in terms of the generalization error, which is not considered in Reddi et al, 2018.
>
> ---
> **Q:** More intuitions about the data distribution.
>
> **A:** Our construction of data distribution is not particularly designed for Adam, but in fact, comes from our understanding of the image data. Specifically, the image data, consisting of many pixels, can be decomposed into feature components (which are useful for the classification task) and noise components (which are useless or even harmful for the classification task). Moreover, the number of feature pixels is typically much smaller than that of noise pixels. Then based on the fact that CNN filters are applied to different patches of the image, our paper considers a simplified image data distribution: each data contains a feature patch and a noise patch, where the feature is sparse and noise is dense. As mentioned in the second paragraph after Definition 3.1. the setting considered in our paper is a simplified case and our theory and analysis can be generalized to a more general data distribution that consists of multiple features and multiple image patches, and the goal will be studying which features can be learned by SGD but cannot be learned by Adam, based on their strength and sparsity.
>
> ---
> **Q:** "Is it possible to replace polynomial ReLU by other activation functions"
>
> **A:** The use of the polynomial ReLU activation function is for simplifying our analysis. It can be replaced by a smoothed ReLU activation function (e.g., the activation function used in [Allen-Zhu and Li, 2020c]). If we assume the underlying input data distribution is Gaussian, we can also deal with ReLU activation function [Li et al., 2020].
>
> ---
> **Q:** "all the algorithms in this paper use full-batch gradients as their updates, instead of mini-batch gradients."
>
> **A:** Our results can be easily extended to the case of using mini-batch stochastic gradients and the results will not change. Basically, using stochastic gradients will not affect feature learning since the feature vector is assumed to be contained in all training data. For SGD, the noise memorization will also be largely similar to that of GD since the gradients (full-batch ones or mini-batch ones) are always the linear combination of data vectors and the learning process of noise vectors will be similar as well. For mini-batch Adam, the update in each iteration can be approximated by signed stochastic gradients and we can also show that the algorithm tends to memorize the noise patch rather than the feature patch since the noise vector is much denser. We will comment on this in the revision.

---

> > ### Comment · Reviewer_9CZz · 2021-11-26
> > **Thanks for the clarification**
> >
> > I thank the authors for their response and their reply to my questions. I now understand the difference between the Reddi 2018 paper and the setting in this one. Also, it would be nice if the authors can improve the results to mini-batch instead of just full-batch updates. I hope you can update them in your camera ready.
> >
> > I am still not so convinced of the polynomial ReLU activation and as pointed out by some other reviewers, this activation seems to be designed particularly for the analysis in this paper. Again, I would not say that using a special activation is the reason to reject this paper, and I would not change my initial rating because I believe there are merits in this paper that are worth encouraging.

---

> > > ### Author Response · Authors · 2021-11-28
> > > **Thank you**
> > >
> > > Thanks for your positive feedback. We will surely update them in the camera ready.

---

### Official Review · Reviewer_6gZd · 2021-11-07

**Correctness:** 4
**Technical Novelty And Significance:** 3
**Empirical Novelty And Significance:** 3
**Recommendation:** 6
**Confidence:** 3

**Main Review:**

The paper makes interesting observations about generalization properties of GD vs ADAM on a few settings.

Strengths
- Global convergence results for full batch ADAM and GD in non-convex setting when proper regularization is applied.
- Both Thrm 4.1 and 4.2 explores interesting differences/non-differences of ADAM and GD with respect to their generalization abilities.
- Based on a carefully constructed data distribution, the paper makes the argument that the inferior generalization performance of ADAM is closely tied to non-convex landscape of deep learning.

Weakness
- While all their observations are interesting in their own right, to me, the main weakness of the paper is its limited scope (2 layer conv net, full batch ADAM instead of online setting), some of which the authors also acknowledge in Section 7.
- Since ADAM performs *better* than SGD in when used with transformer/lstms in language setting, a more complete analysis would be to also consider a data distribution/model where that can be explained as well.

**Summary Of The Paper:**

This paper studies, in more detail than predecessors, the generalization gap between Adam and gradient descent for image-like datasets.

**Summary Of The Review:**

This paper makes interesting and novel observations about generalization gap between adam and GD. Their observations are somewhat limited in scope.

---

> ### Author Response · Authors · 2021-11-18
> **Response to Reviewer 6gZd**
>
> ---
> Thank you for your positive comments.
>
> **Q:**  "the main weakness of the paper is its limited scope."
>
> **A:** Many existing works on the analysis of deep learning assume that the neural network is sufficiently wide so that the training loss is locally almost convex (e.g., works in the so-called Neural Tangent Kernel (NTK) regime). Therefore, when using weight decay, these results will indicate that SGD and Adam will converge to the solutions with the same generalization performance, which is not consistent with the observation in practice. In our paper, we develop new theoretical analysis and demonstrate the separation between SGD and Adam on a simplified image classification task (two-layer CNN, full-batch Adam, etc.). Our analysis is beyond the NTK regime. We believe that without fully understanding the learning patterns on simple models, it is difficult to explore more complicated problems (e.g., multi-layer). Moreover, our analysis can be at least extended to mini-batch Adam/SGD, since using mini-batch stochastic gradients do not affect the current analysis for feature learning and noise memorization a lot (their convergence rates could be slightly different, but the separation between Adam and SGD still holds). For multi-layer NN cases, the high-level idea can still be applied, where we will not only look at the input data but also consider the output of each intermediate layer as “input”. However, the analysis will be much more involved and we will leave it as a future work.
>
> ---
> **Q:**  "Since ADAM performs better than SGD when used with transformer/lstms in language setting, a more complete analysis would be to also consider a data distribution/model where that can be explained as well."
>
> **A:** Thanks for your suggestion. It is true that Adam performs better than SGD in solving many natural language processing problems, which could stem from either the network structure or data distribution (they are largely different from image classification tasks). We have also mentioned this in Section 7 and will explore it in our future work.

---

### Comment · Area_Chair_4W7L · 2021-11-17
**Relation to Wilson**

It would be great if the authors can further relate their results to those of Wilson et al. In the Wilson paper, it was shown that Adam can be inferior to GD in terms of test error for certain over-parameterized cases. This is also a key conclusion of the current paper, so I think more discussion of the relation between the two results is appropriate.
In this context, the current paper mentions theorem 4.2 as a reason why non-convexity is the main driver of Adam/SGD difference. But I don't find this completely convincing. It seems like the theorem "just" says that if one adds regularization then overparameterization is "resolved" in a particular way, thus over-riding the specific implicit biases of Adam and SGD.
Also, it is not clear what will happen if regularization is added to the non-convex objective of Theorem 4.1. Presumably with sufficient regularization, this will also force a unique solution and counter the differences between Adam and SGD.
Please address these issues to clarify the contribution of the submission.

---

> ### Author Response · Authors · 2021-11-18
> **Thank you for your comments and suggestions.**
>
> ---
>
> **Q:** "It would be great if the authors can further relate their results to those of Wilson et al. In the Wilson paper, it was shown that Adam can be inferior to GD in terms of test error for certain over-parameterized cases."
>
> **A:** Thanks for pointing this out. We will add a detailed discussion in the revision.
>
> ---
> **Q:** "In this context, the current paper mentions theorem 4.2 as a reason why non-convexity is the main driver of Adam/SGD difference. But I don't find this completely convincing. It seems like the theorem "just" says that if one adds regularization then overparameterization is "resolved" in a particular way, thus over-riding the specific implicit biases of Adam and SGD."
>
> **A:** It is true that without regularization, for convex problems (such as the overparamterized linear regression problem considered in Wilson et al.), Adam and SGD will exhibit different implicit biases and find global solutions with significantly different generalization performances. However, this difference can be resolved by adding (a small) regularization to the training loss, and consequently, the training loss becomes strongly convex, and Adam and SGD will output the same solution with good generalization performance. In contrast, this is not true in nonconvex optimization (the problem considered in our paper). In particular, after adding a regularization to the nonconvex empirical loss function, its landscape is still largely nonconvex and contains many global solutions (that give zero training error) that generalize differently. Our results show that, unlike the convex optimization problem, adding regularization cannot make Adam and SGD converge to the same global solution (instead, their solutions will lie in different basins) when the training loss function is nonconvex. This is the main message conveyed by our paper and we believe that the different performance of Adam and SGD is fundamentally tied to the non-convexity of the training loss function.
>
> ---
> **Q:** "Also, it is not clear what will happen if regularization is added to the non-convex objective of Theorem 4.1. Presumably with sufficient regularization, this will also force a unique solution and counter the differences between Adam and SGD. "
>
> **A:** In fact, we have added regularization to the non-convex objective in Theorem 4.1 (we apologize that we did not make it clear in the statement of Theorem 4.1) and proved that with regularization, Adam and SGD will converge to different solutions with different generalization performance.

---

### Author Response · Authors · 2021-11-23
**We have uploaded the revision**

We thank all reviewers and area chair for their insightful comments and suggestions! We have uploaded a revision of our paper to reflect our response to the comments. The revised parts are highlighted in purple color. Please let us know if you have any further comments and suggestions.

---

### Decision · Program_Chairs · 2022-01-20

**Decision:**

Reject

**Comment:**

The paper considers the difference between GD and ADAM in terms of implicit bias. It considers a specific distribution and architecture where the two algorithms converge to different solutions while perfectly fitting the training data. The authors highlight the fact that this happens while adding regularization, which does not happen in the linear case.
The reviewers found some of the insights and analysis interesting. However, they also had reservations about the impact of the results given that it is known that GD and ADAM have different implicit biases, and that the distribution appears specifically crafted towards  showing this effect for the architecture studied.
In future versions, the authors are encouraged to better motivate the chosen distribution, use more standard neural architecture (e.g., standard relu), and provide more explanation about the role of regularization in their result.